# Light and matter co-confined multi-photon lithography

Lingling Guan[1,5], Chun Cao [2,3,5] ✉, Xi Liu[1,5], Qiulan Liu[1], Yiwei Qiu[1], Xiaobing Wang[1], Zhenyao Yang[1], Huiying Lai[1], Qiuyuan Sun[1], Chenliang Ding[1], Dazhao Zhu[1], Cuifang Kuang [2,4] ✉ & Xu Liu [2,4] ✉

Mask-free multi-photon lithography enables the fabrication of arbitrary nanostructures low cost and more accessible than conventional lithography. A major challenge for multi-photon lithography is to achieve ultra-high precision and desirable lateral resolution due to the inevitable optical diffraction barrier and proximity effect. Here, we show a strategy, light and matter co-confined multi-photon lithography, to overcome the issues via combining photo-inhibition and chemical quenchers. We deeply explore the quenching mechanism and photoinhibition mechanism for light and matter co-confined multiphoton lithography. Besides, mathematical modeling helps us better understand that the synergy of quencher and photo-inhibition can gain a narrowest distribution of free radicals. By using light and matter co-confined multiphoton lithography, we gain a 30 nm critical dimension and 100 nm lateral resolution, which further decrease the gap with conventional lithography.

Modern science and industry have increasingly strong demands for high-resolution nanolithography. Multi-photon lithography (MPL) with advantages in mask-free, high-precision, and arbitrary 3D architectures[1–3], has been applied to versatile fields[4–6]. However, it seems impossible for MPL to realize comparable critical dimension (CD, the minimum achievable linewidth of an isolated line) and lateral resolution (LR, the minimum grating period between two separated lines) achieved by e-beam lithography (EBL)[7] and extreme ultraviolet lithography (EUV)[8] due to optical diffraction barrier[9] and proximity effect (simultaneous writing of structures in close spatial proximity generates fabrication artifacts and limits the accessible structure resolution[10]). Specifically, the optical diffraction barrier is mainly caused by the long wavelength of the excitation laser, while the proximity effect is generally caused by the diffusion and accumulation of free radicals[11].

As a result, scientific pioneers have carried out extensive studies to improve the CD and LR of MPL. Reducing the wavelength of the excitation laser is an effective method[12]. Atsushi Taguchi et al. used a femtosecond laser with a 400 nm wavelength for MPL[13]. The achieved CD (80 nm) and LR (160 nm) are better than those previously reported for NIR excitation[14] or 520 nm excitation[15]. However, both obtaining shorter-wavelength femtosecond lasers and developing applicable multiphoton photoresists are daunting tasks. Therefore, light-confined multiphoton lithography (LC-MPL), inspired by stimulated emission depletion (STED) microscopy, was born in 2009[16], which can somewhat overcome the diffraction barrier via a shaped inhibition beam, although the underlying mechanisms remain more controversial[17]. It is possible to create a remarkable 36 nm CD ($\lambda$/14.8) on the sub-diffraction scale with a 140 nm LR by LC-MPL made by Minfei He et al.[18]. Unfortunately, due to the inevitable power fluctuation and focus drift in the system, the line-edge-roughness of the CD they gained is not ideal, which is intolerable for further pattern transfer. More importantly, the LR in Minfei He et al.'s work is still limited by the proximity effect.

Another alternative way to improve the performance of MPL is to develop multiphoton photoresists, which are usually composed of photo-initiators, resins, and solvents[19,20]. For decades, although a diversity of photo-initiators[21] and photoresists[22] with good

[1]Research Center for Intelligent Chips and Devices, Zhejiang Lab, 311121 Hangzhou, China. [2]State Key Laboratory of Extreme Photonics and Instrumentation, College of Optical Science and Engineering, Zhejiang University, 310027 Hangzhou, China. [3]School of Mechanical Engineering, Hangzhou Dianzi University, 310018 Hangzhou, China. [4]ZJU-Hangzhou Global Scientific and Technological Innovation Center, 311200 Hangzhou, China. [5]These authors contributed equally: Lingling Guan, Chun Cao, Xi Liu. ✉e-mail: caochun@iccas.ac.cn; cfkuang@zju.edu.cn; liuxu@zju.edu.cn

photosensitivity have been designed for MPL, most of them have ignored the key issue of proximity effect. In 2006, Sang Hu Park et al. proposed a matter-confined lithography (MC-MPL) by introducing 2,6-di-tert-butyl-4-methylphenol (DBMP) into photoresists as radical quenchers to suppress the diffusion of radicals, thereby obtained a CD of 96 nm[23], but LR was not mentioned. Subsequently, many other quenchers, such as 4-methoxyphenol (MEHQ)[24], 2-(dimethylamino) ethyl methacrylate (DMAE-MA)[11], butylated hydroxytoluene (BHT)[25] and 2,2,6,6-tetramethyl-4-piperidyl-1-oxyl (TEMPO)[26] were also used for MC-MPL. Recently, the quencher, bis(2,2,6,6tetramethyl-4-piperidyl-1-oxyl)sebacate (BTPOS), was reported for 3D nano-printing, and a resolution below 150 nm was achieved, which surpasses previous best values obtained by MPL[27]. BTPOS contains two linked 2,2,6,6-tetramethyl-4-piperidyl-1-oxyl (TEMPO) moieties, which can suppress the undesired polymerization by rapidly reacting with free radicals to form a non-reactive adduct[28]. Notably, if the quencher can be consumed rapidly in the exposed area, it will lead to a continuous diffusion of the quencher molecules from other non-depleted areas, which will further suppress the proximity effect[11,29]. In addition, a heat-shrinking method is also effective in improving CD and LR, e.g. photonic crystals with CD = 100 nm, LR = 350 nm were obtained by Y. Liu et al.[30]. However, this method cannot improve the resolution of patterns on the substrate, and the shrinkage anisotropy is difficult to resolve.

Interestingly, although the above implementations of LC-MPL and MC-MPL are very different, they produce essentially the same results, that is, the suppression of free radicals at the edges of the exposed area. However, achieving stable sub-50 nm linewidth and 100 nm resolution simultaneously remains a great challenge, even using LC-MPL or MC-MPL, and this has great implications for MPL technology towards optoelectronics and integrated circuits.

Therefore, light and matter co-confined multiphoton lithography (LMC-MPL), which combines photo-inhibition and quenchers, is presented in this work. The matter-confining capability of various quenchers, which refers to their ability to inhibit polymerization in the tails of the focused laser beam by chemical and physical quenching processes and thus confine the polymerization zone into a smaller space, was first investigated. The optimal quencher was used for exploring the matter confined mechanism and the light-confining mechanism. To demonstrate the synergistic effect of light-confined and matter-confined, the specific performance of MPL and MC-MPL, LC-MPL, and LMC-MPL were explored successively. After that, the linewidth and lateral resolution obtained by the four lithography strategies, MPL, LC-MPL, MC-MPL, and LMC-MPL, were studied and compared in detail. Remarkably, theoretical simulations were proposed to clearly reveal the effects of quenchers and photo-inhibition. The synergistic strategy proposed in this work can not only obtain ultra-high precision (CD = 30 nm) and resolution (LR = 100 nm) but also have excellent 3D manufacturing and pattern transfer ability.

## Results

### Quenchers optimization and the advantages of MC-MPL

The schematic diagram of MC-MPL is illustrated in Fig. 1a, and a series of photoresists containing the same resins/initiator (Fig. 1b) and various quenchers (Fig. 1c) were prepared (Supplementary Table 1) to optimize the matter confining capability for MC-MPL. By collecting the threshold power ($P_{th}$) of these photoresists at different writing speeds, the ability of each quencher to inhibit polymerization can be compared. As a result, all the quenchers can hinder the polymerization reaction, causing an increase of $P_{th}$ (Supplementary Fig. 1a), and Pr2 exhibits the highest $P_{th}$. The lithography results show that the linewidths of the photoresists are narrowed once the quencher is introduced at the same processing parameters (Fig. 1d and Supplementary Fig. 1b). Obviously, Pr2 exhibits the best matter confining capability, which substantially compressed the lateral linewidth of Pr1 from 364 to 158 nm, proving that Q-1 should be the most efficient quencher.

Figure 1d shows that the inhibition capability of quenchers is inversely proportional to the relative molecular weight of the quenchers. An increase in molecular weight might lead to a decrease in the active site freedom and molecular diffusion coefficient[31,32], thus causing inefficient kinetic processes of quenching. In addition, we have also investigated and compared the smallest linewidth of the photoresists with different quenchers. As shown in Supplementary Fig. 2, Pr2 also exhibits the smallest linewidth. TEMPO also performs the best compared with other different types of quenchers, 4-Methoxyphenol (MEHQ) and 2-(Dimethylamino)ethyl methacrylate (DMAE-MA) in Supplementary Fig. 3. All these demonstrate that TEMPO has the best matter confining capability among these quenchers. Thus, only Q-1 (TEMPO) was employed as a chemical quencher hereinafter.

To explore the advantages of MC-MPL, We first fabricated a miniature 3D model of *Nezha*, a figure of Chinese legend, using Pr1 and Pr2 (Fig. 1e). According to the original morphology, the hair lines of *Nezha* made by Pr2 are more clearly visible than that made by Pr1, which demonstrates that MC-MPL has a better ability to fabricate fine structures than MPL. Then, a threshold test (Supplementary Fig. 4) was adopted to figure out the effect of the quencher on the processing window and the nonlinearity absorption exponent (*N*) of MC-MPL. The TEMPO in Pr2 not only makes $P_{th}$ of MC-MPL higher than MPL (Fig. 1f) but also raises the damage power of MC-MPL (Supplementary Fig. 4). The simultaneous increase in $P_{th}$ and damage power results in little impact on the width of the processing window (Supplementary Fig. 4), especially when writing at low speed. In addition, the introduction of TEMPO also exhibits little influence on the values of *N*, which are well fitted to 2.90 and 2.84 for MPL and MC-MPL, respectively (Fig. 1g), in good agreement with previous report (*N* = 3)[33]. Meanwhile, we find the energy level ($S_{52}$, 7.09 eV) matching the three-photon absorption by density functional theory (DFT) calculation (Fig. 1h), i.e., it can be assumed that the three-photon absorption occurs in MC-MPL and MPL.

Afterward, we investigated the CD achieved by MPL and MC-MPL to verify the effectiveness of TEMPO in reducing linewidth. Obviously, the linewidth reduces as laser power decreases for both MPL and MC-MPL (Fig. 1i). In detail, the linewidth fabricated via MPL decreases from 215 to 55 nm (image A in Fig. 1i) when laser power is reduced from 1.24 to 0.58 mW, while that of MC-MPL decreases from 188 to 44 nm (image B in Fig. 1i) with the power reducing from 1.77 to 1.10 mW. In other words, MC-MPL can more efficiently change the linewidth by adjusting the laser power in small increments and meanwhile realize a CD of 44 nm. Overall, quenchers are a double-edged sword for MC-MPL. Although TEMPO will reduce the photosensitivity, it allows for higher 2D/3D lithography precision for MC-MPL.

### Matter confining mechanism

So far, it is not clear how the TEMPO inhibits polymerization, although this is very critical for MC-MPL. In order to understand the matter-confining mechanism in MC-MPL, it is first necessary to understand how photopolymerization occurs. It is generally believed that the photoinitiation process of DETC used in MC-MPL undergoes three steps (Fig. 2a): multiphoton absorption (MPA, $S_0 \rightarrow S_1$), inter-system crossing (ISC, $S_1 \rightarrow T_1$), and finally generating free radicals at $T_1$ state[34]. The $T_1$-state DETC will form exciplex species through fast electron transfer, followed by proton transfer to produce a C-centered radical and an N-centered radical to initiate polymerization[35]. The generation of the above free radicals is confirmed by electron spin resonance (ESR) spectroscopy in Fig. 2b.

So, blocking any of the above links can inhibit polymerization. Based on the following experiments, we believe that there are three quenching paths to inhibit polymerization (Fig. 2a). Quenching path 1: static quenching between TEMPO and the ground-state DETC. The quenching process of DETC by TEMPO can be understood by the well-known Stern–Volmer equation[36]. By testing the fluorescence emission

(Fig. 2d) of the photoresists with different concentrations of TEMPO, Stern–Volmer plots of fluorescence intensity are given in Fig. 2e. A linear Stern–Volmer relationship may be observed if either a static (ground state interaction) or dynamic (excited state interaction) quenching paths is dominant[36,37]. In Fig. 2e, the Stern–Volmer plot shows a linear relationship at low concentrations (<20 mM) and a non-linear relationship at high concentrations. Therefore, one kind of quenching path dominates at low concentrations, while the non-linear plot at higher concentrations in Fig. 2e proves the co-existence of dynamic and static quenching[36]. The peak intensity (ESR spectroscopy, Fig. 2c) and the number of spins (Supplementary Fig. 5a) of TEMPO is significantly reduced when DETC is introduced (before UV), which

demonstrates that there is indeed an interaction between ground state DETC and TEMPO. Interestingly, the peak intensity and the number of spins for TEMPO + DETC increases under UV irradiation, and it will gradually recover as the UV lamp is turned off. Temperature changes due to the switching of UV lamps may also cause changes in ESR intensity, especially on longer time scales (up to 60 min). Nevertheless, the ESR signal of TEMPO remains unchanged before and after UV irradiation (Fig. 2c), which illustrates that whether it is the light itself or the temperature changes caused by the light will not directly affect the number of spins for TEMPO (Supplementary Fig. 5a). So, it is probably that temperature changes caused by UV irradiation may also have an impact on the interaction between ground state DETC and TEMPO.

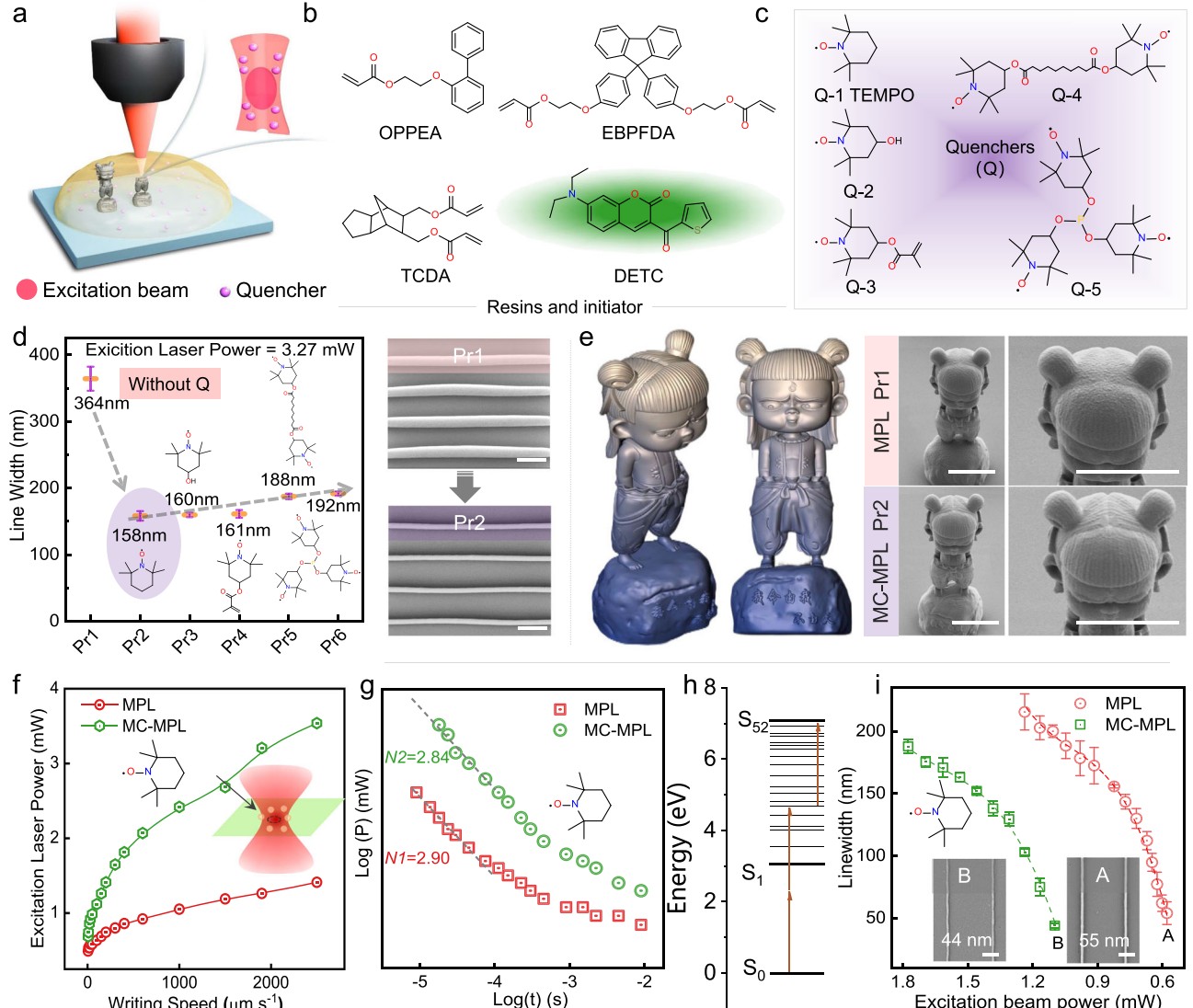

**Fig. 1 | Performance of different quenchers and comparison of MPL and MC-MPL. a** Schematic diagram of MC-MPL using a microscope objective lens and an excitation beam. **b** The chemical structures of resins and initiators. The selection of the mixed monomer here is to obtain a refractive index ($n$ = 1.518) that matches the objective lens[19]. **c** The chemical structures of different quenchers (Q): Q-1 (2,2,6,6-tetramethylpiperidinooxy, TEMPO), Q-2 (4-Hydroxy-2,2,6,6-tetramethyl-piperidinooxy), Q-3 (TEMPO methacrylate), Q-4 (Bis(2,2,6,6-tetramethyl-4-piperidyl-1-oxyl)Sebacate), Q-5 (Tri-(4-hydroxy-TEMPO)phosphite). **d** Linewidth of Pr1 (the photoresist without quencher), Pr2 (with Q-1), Pr3 (with Q-2), Pr4 (with Q-3), Pr5 (with Q-4), and Pr6 (with Q-5) at the same processing parameters (excitation beam power: $P_{ex}$ = 3.27 mW, writing speed: $v$ = 500 μm s$^{-1}$). The corresponding SEM images of Pr1 and Pr2 (error bars represent mean ± SD for four independent measurements). Scale bar: 1 μm. **e** The original 3D model and fabricated SEM

morphology of *Nezha* (40 μm in height) using Pr1 and Pr2. $P_{ex}$ = 5.7 mW and $v$ = 10 mm s$^{-1}$. Scale bar: 10 μm. **f** The threshold power ($P_{th}$) curves of MPL and MC-MPL at different writing speeds. $P_{th}$ is defined as the minimum laser power that will allow the photoresist to be retained after development. **g** The double-logarithmic representation of the excitation laser power ($P$) versus the exposure time ($t$) for MPL and MC-MPL. The dashed curve is a linear fit to the data at a short exposure time, and nonlinearity absorption exponent $N$ is the slope of the linear fit line. **h** The calculated Jablonski diagram (single state) of DETC and energy level transition processes occurring in three-photon absorption. **i** The curve of linewidth versus excitation beam power for MPL and MC-MPL and the error bars represent mean ± SD for five independent measurements. The inserted SEM images (A and B) are the minimum CD of MPL and MC-MPL corresponding to points A and B. Scale bar: 400 nm.

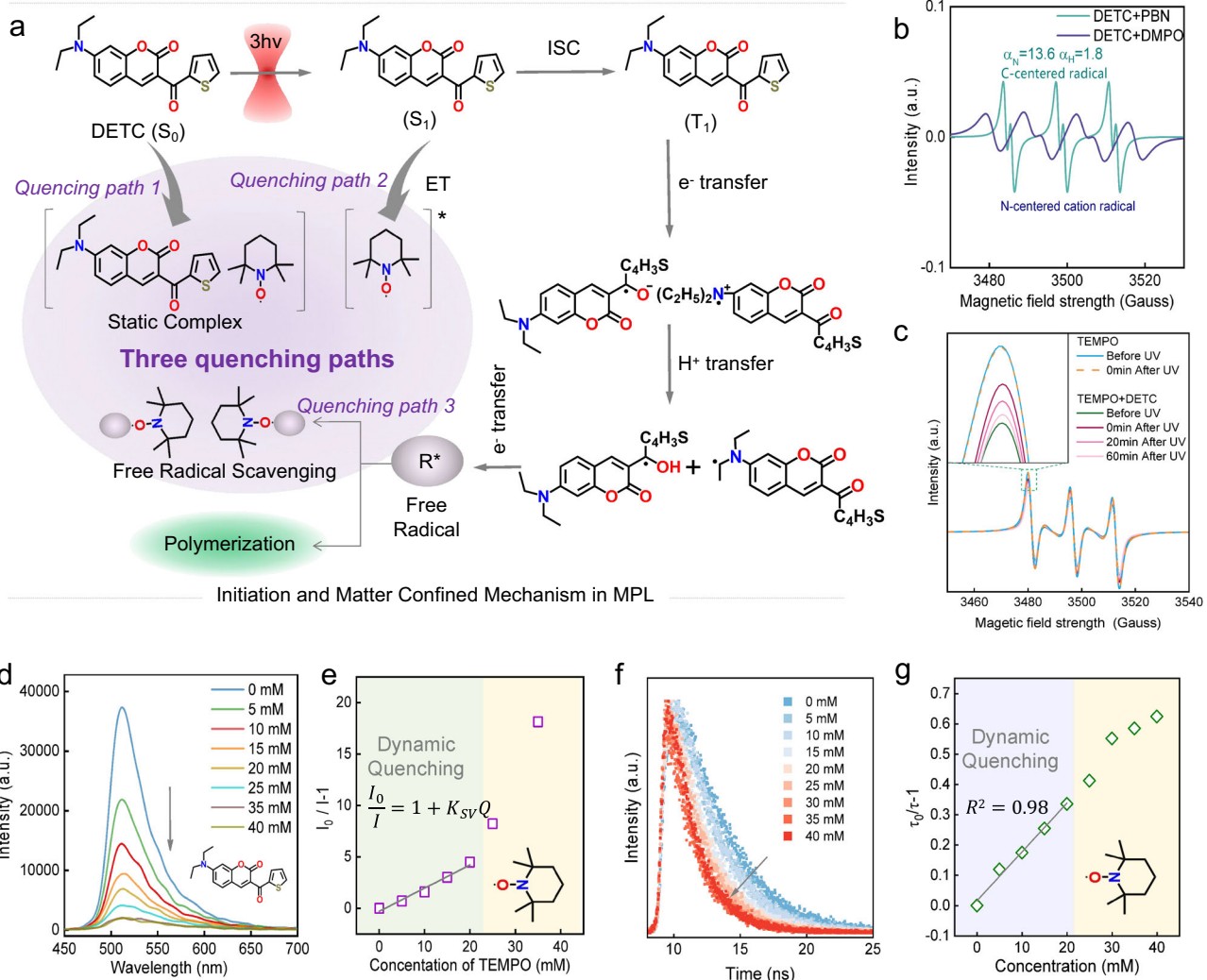

**Fig. 2 | Quenching mechanism of TEMPO. a** Initiation and matter confining mechanism in MC-MPL. ET energy transfer, ISC inter-system crossing. **b** ESR spectra of DETC in dichloromethane after 10 min irradiation by a 365 nm light, in which N-tert-Butyl-α-phenylnitrone (PBN) and 5,5-dimethyl-1-pyrroline N-oxide (DMPO) are employed as trapping agents. The hyperfine coupling constants of the PBN-based radical adduct are $\alpha_N = 13.6$ G and $\alpha_H = 1.8$ G, suggesting the generation of C-centered radicals. The ESR spectra of the DMPO-based radical adduct imply the formation of N-centered cation radicals. **c** ESR spectra of TEMPO (0.2 mM) and TEMPO (0.2 mM) + DETC (1 mM) in dichloromethane before UV irradiation and after UV irradiation (placed in the dark for 0, 20 or 60 min).

The irradiation duration is 10 min by a 365 nm light. **d** The fluorescence emission spectra ($\lambda_{ex} = 263$ nm) of the photoresists (DETC) with different concentrations (0–40 mM) of TEMPO. The concentration of TEMPO in Pr2 is about 40 mM. **e** Typical Stern–Volmer plots, $\frac{I_0}{I} = 1 + K_{SV}Q$, for the quenching process of DETC by TEMPO in the photoresist, where $I_0$ and $I$ is the fluorescence intensity in the presence and absence of quenchers, $Q$ is the concentration of quenchers. **f** Fluorescence decays spectra of the photoresists (DETC) with different concentrations (0–40 mM) of TEMPO. **g** The Stern–Volmer plots of fluorescence lifetime for the quenching process of DETC by TEMPO in photoresist, where $\tau_0$ and $\tau$ is the fluorescence lifetime in the presence and absence of quenchers.

Under UV irradiation, temperature rises, and the ESR signal of TEMPO + DETC increases. After turning off the UV lamp, the temperature drops, and the signal recovers. Additionally, a redox hypothesis seems to give a better explanation. DETC is photosensitive and is likely to undergo intermolecular electron transfer with TEMPO after absorbing photons, causing a spin reactivation of TEMPO[38] and thereby leading to a change of ESR signal after UV irradiation. Quenching path 2: dynamic quenching by energy transfer from $S_1$-state DETC to TEMPO. The greatly reduced lifetime (Fig. 2f) when the TEMPO is present demonstrates that TEMPO will interact with the $S_1$-state DETC. This will inevitably prevent the ISC process and suppress the generation of radicals. In addition, the linear change of the lifetime indicates that dynamic quenching dominates at lower concentrations of TEMPO (Fig. 2g). In the case of higher concentrations of TEMPO, the two processes (static and dynamic quenching) may be competitive, which results in a nonlinear relationship of lifetime and fluorescence[36].

The combination of static and dynamic quenching almost completely quenched the fluorescence of Pr2 (Supplementary Fig. 5b and Fig. 2d). Quenching path 3: active free radical scavenging by nitroxide radicals in TEMPO. It is known that TEMPO is an electron acceptor and can react rapidly with active radicals generated by DETC, forming a non-reactive adduct[28]. Theoretically, when the UV light is turned on, the free radicals produced by the excited state DETC also deplete the TEMPO via quenching path 3 to reduce the ESR signal. Therefore, quenching path 1 and quenching path 3 have opposite effects on the ESR signal of the TEMPO, i.e., they are a competitive pair. Since the quantum efficiency of radical generation by excited DETC is less than 1, the effect of static quenching on the ESR signal of TEMPO should dominate, resulting in the experimental phenomenon in Fig. 2c. Moreover, Supplementary Fig. 5c shows that TEMPO has very weak absorption compared with DETC (350–550 nm) and has little impact on the absorption peak of DETC, which proves the quenching effect of

TEMPO does not come from the competitive absorption of the excitation and inhibition light. In summary, the above three quenching paths allow TEMPO excellent matter confining capability in MC-MPL. All the above conclusions are based on ex-situ tests. Therefore, it is undeniable that they have limited validity in exploring the mechanism for in-situ MPL.

## The advantages of LMC-MPL and light confining mechanism

The schematic diagram of LMC-MPL is illustrated in Fig. 3a, and a doughnut-shaped inhibition beam was introduced into the MPL system (Supplementary Fig. 6). Both Pr1 and Pr2 exhibit efficient light confining capability, which refers to the ability of the inhibition beam to hinder polymerization at the tail of the focal beam via STED process, and thus confine the polymerization to a smaller zone. The written line can be completely erased once the inhibition beam is turned on (Fig. 3b), indicating that the introduction of TEMPO will not lead to the loss of the light-confining capability. The manufacturing capability for woodpile structure by MPL, MC-MPL, and LMC-MPL were verified and compared in Fig. 3c and d. Compared with MPL (voxel: $a = 260$ nm and $b = 806$ nm) and MC-MPL (voxel: $a = 78$ nm and $b = 250$ nm), LMC-MPL can obtain the finest (voxel: $a = 41$ nm and $b = 148$ nm) woodpile structure under the fixed excitation laser power, where the voxel is the size of an isolated fabricated structure (analogous to a pixel−the 2D picture element). Moreover, both at a fixed excitation laser power and at the respective threshold excitation laser power, LMC-MPL can gain the highest resolution (200 nm, Supplementary Fig. 7) of woodpile structures among the three strategies. A 3D *Nezha* with 40 μm height (Supplementary Fig. 8a) was fabricated by LMC-MPL using Pr2 as well. Compared to MPL and MC-MPL in Fig. 1e, the details of *Nezha*'s hair, eyes, and nose become clearer, presenting a more vivid *Nezha*. Besides, when *Nezha* is further reduced in size to 30 and 20 μm (Supplementary Fig. 8b), many details of the morphology cannot be clearly fabricated by MPL or MC-MPL, while LMC-MPL can still do so. The same conclusion can also be gained for a square pyramid structure (Supplementary Fig. 9). All these results demonstrate that LMC-MPL can fabricate fine 2.5D or 3D structures.

Next, we explored the achievable CD of LC-MPL (Pr1) and LMC-MPL (Pr2) by a typical test array in Fig. 3e. With different fixed excitation beam power ($P_{ex}$), the curves of linewidth versus inhibition beam power ($P_{in}$) in Fig. 3f were obtained by collecting the linewidths in Supplementary Fig. 10. Both LMC-MPL and LC-MPL show the same trend that the linewidth reduces first (decreasing part) and increases subsequently (increasing part) when $P_{in}$ increases. The decreasing part owing to light confining capability, while the increasing part should be attributed to an enhanced single-photon absorption (SPA) of the inhibition beam, which will dominate over the light confining capability once $P_{in}$ reaches a certain value[39]. Therefore, there exists an optimal $P_{in}$ for LC-MPL (Pr1, 2.23 mW) and LMC-MPL (Pr2, 9.65 mW) to achieve CDs. A higher optimal $P_{in}$ of LMC-MPL could be attributed to the quenching path 2 in Pr2. Due to the presence of the TEMPO, almost all the $S_1$-state DETC that are excited via SPA of the inhibition beam will be quenched; thus, the generation of excessive free radical ($R^*$) can be hindered in LMC-MPL. Therefore, as $P_{in}$ increases from 0 mW to the optimal value, the linewidth can be narrowed from 59 to 39 nm (Point C in Fig. 3f, and g) by LC-MPL and from 55 to 30 nm (Point D in Fig. 3f and g) by LMC-MPL. In a word, with the synergistic advantage of the matter-confining capability and light-confining capability, LMC-MPL exhibits the smallest CD of 30 nm.

Nevertheless, the light confining mechanism remains more controversial[17], especially in the more complex LMC-MPL, which has never been explored. Before discussing the light confining mechanism, the photophysical transitions of DETC during photo-excitation must first be clarified. According to the previously reported photo-excitation theory[39,40], free radicals are merely formed from ISC ($S_1 \to T_1$). In other words, once $S_1$-state DETC is quenched completely by TEMPO (quenching path 2 in Fig. 2a), initiation will be totally blocked, and thereby Pr2 should lose the polymerization capability, but it is not. The above results show that even if the concentration of TEMPO increases to almost completely quench the fluorescence of DETC ($S_1$-state DETC), initiation and polymerization in Pr2 still occur (Fig. 3b–e). So, we proposed an assumption that the $S_n$-state DETC has two paths to go after MPA (Fig. 3i). One is to transition from $S_n$ to $T_n$ state (ISC, $S_n \to T_n$) and generate free radicals at $T_n$ state, and the other is to return to $S_1$ state through vibrational relaxation (VR), then transition to $T_1$ state (ISC, $S_1 \to T_1$), and finally generate free radicals at $T_1$ state. In-depth, the proportion of free radicals generated from the two paths are proved to be 62% (ISC, $S_1 \to T_1$) and 38% (ISC, $S_n \to T_n$), respectively (Supplementary Section A). Through DFT calculations, we have found the energy levels (Supplementary Fig. 11) that match these two paths, thus confirming the assumption. A similar assumption has been proposed by Martin Wegener et al. before[34]. The only difference is he believed polymerization is initiated via two competing pathways (three-photon absorption and four photon absorption), while we think there is only a three-photon absorption based on our experiments.

Currently, there are two typical light confining mechanisms for LC-MPL, namely, stimulated emission depletion (STED) and triplet state absorption (TSA)[9]. For STED, it is regarded that an $S_1$-state initiator will be brought back to the $S_0$ state by inhibition beam[39,40]. If this theory holds, the inhibition beam should lose its light-confining capability in Pr2 because the $S_1$-state DETC is completely quenched by the TEMPO before STED occurs. But in fact, Pr2 still maintains a good light confining capability in Fig. 3b. By contrast, TSA suggests that $T_1$-state DETC can be consumed by an inhibition beam through a process of $T_1 \to T_n$[41,42]. If the inhibition process is determined by TSA, the introduction of TEMPO should have no effect on the efficiency of light confining capability, and the slope (related to TSA-$K_{T1}$, Fig. 3h and i) of the curves in the decreasing part in Fig. 3f should remain constant for Pr1 and Pr2. However, the slope of Pr2 reduces significantly (Fig. 3f), which means that TEMPO weakens the light confining capability. Overall, neither the typical STED nor TSA seems reasonable for a light confining mechanism in the complex LMC-MPL.

Considering there are two photo-excitation paths, we proposed a two-step-STED light confining mechanism. As shown in Fig. 3i, by STED, the inhibition beam not only enables the transition from $S_1$ to $S_0$ (STED-$K_{S1}$) but also from $S_n$ to $S_1$ (STED-$K_{Sn}$). This two-step-STED light confining mechanism can be well matched with the experimental results in Fig. 3f. For example, the reason that Pr2 still has a good light confining capability, can be attributed to STED-$K_{Sn}$, even if STED-$K_{S1}$ is blocked by TEMPO. The synergistic effect of STED-$K_{S1}$ and STED-$K_{Sn}$ offers Pr1 a steeper slope ($K_{S1} + K_{Sn}$) in the decreasing part than that ($K_{Sn}$) of Pr2. To further verify this two-step-STED, we performed DFT calculations for the ground and excited states of DETC with single and triplet spin multiplicities (Supplementary Fig. 11). As a result, three-photon absorption (525 nm, 2.36 eV per photon) will excite ground-state electrons to jump to the energy level of $S_{52}$ (7.09 eV), and STED-$K_{Sn}$ is very likely to happen at $S_{16}$ because the energy difference between $S_{16}$ and $S_1$ ($\Delta E = 2.32$ eV) is very close to the energy of single inhibition photon (532 nm, 2.33 eV). Therefore, electrons at $S_{52}$ will quickly transition to $S_{16}$ (5.37 eV) via VR before STED-$K_{Sn}$. Meanwhile, $T_{20}$ (5.35 eV) allows a great possibility of ISC for $S_{16}$ (5.37 eV) to further generate radicals. This can be a perfect fit with the above proposed two photo-excitation paths and a two-step-STED light confining mechanism.

## CD, LR, and mathematical modeling

To understand more intuitively of the method, we explored the lithography performance (CD and LR) of MPL, LC-MPL, MC-MPL, and LMC-MPL with the same excitation and the respective optimal inhibition dosage. In order to better distinguish CD and LR, a schematic diagram was given in Supplementary Fig. 12. The laser power distribution and quencher distribution in the exposure area of the above

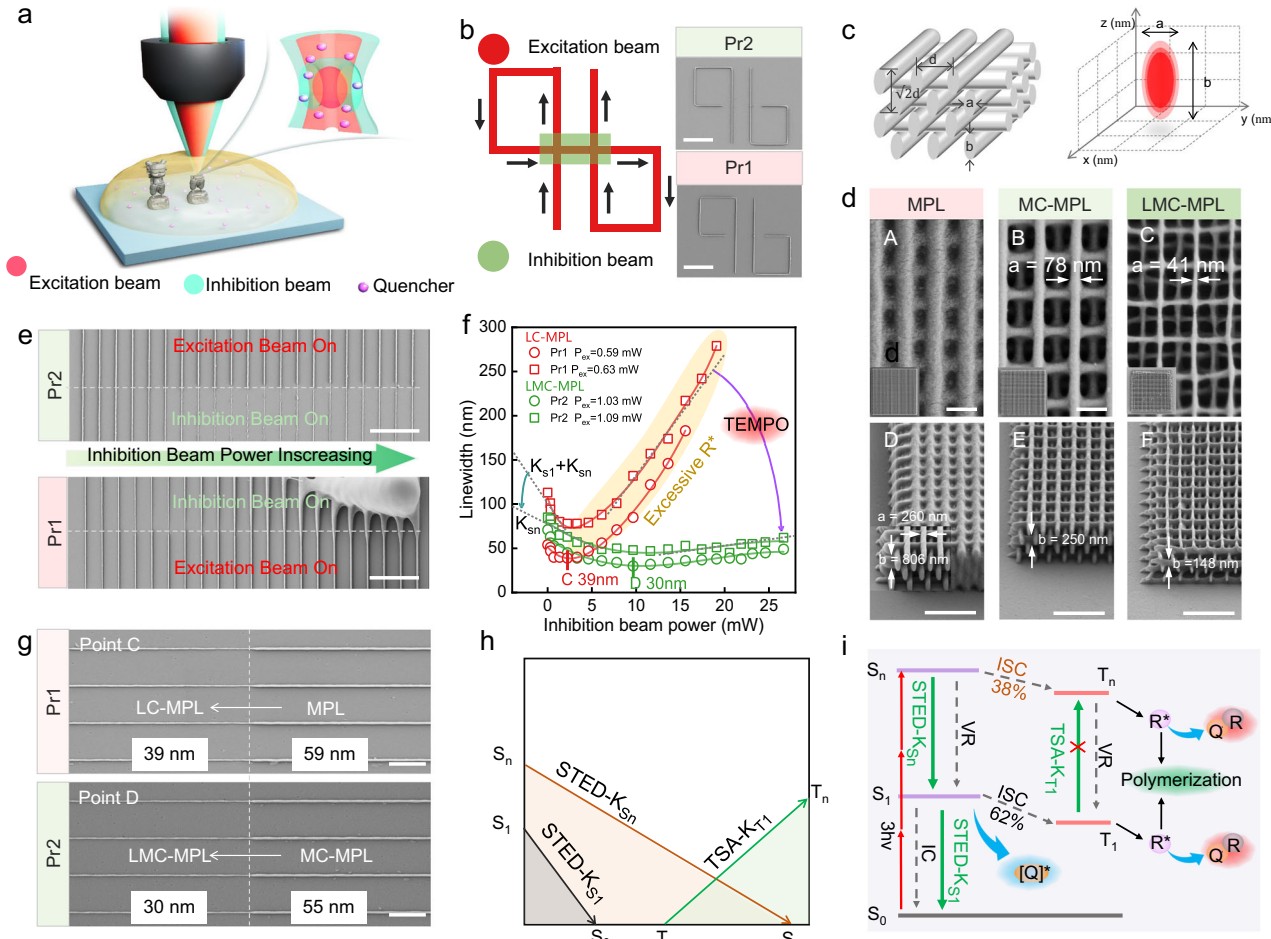

**Fig. 3 | The performance of LMC-MPL and light confining mechanism.**
**a** Schematic diagram of LMC-MPL. **b** Standard pattern for light confining capability tests, excitation beam is always on while inhibition beam is only on in the green area. Scale bar: 3 μm. **c** Left: schematic of a 3D woodpile structure with lateral period $d$ and axial period $\sqrt{2}d$. Right: Schematic diagram of voxels. **d** Woodpile structures ($d = 300$ nm) made by MPL, MC-MPL, and LMC-MPL ($P_{ex} = 1.12$ mW, $P_{in} = 9.65$ mW and $v = 50$ μm s$^{-1}$). A–C are taken perpendicular to the $xy$ plane (scale bar: 300 nm), while D–F are taken with the $xy$ plane tilted 45° (scale bar: 1 μm). With $d = 300$ nm, the lines of woodpile structure obtained by MPL cannot be separated. Therefore, the period is expanded to $d = 700$ nm (inset of D image) for MPL. **e** Light confining capability test patterns for Pr1 and Pr2. $P_{in}$ is gradually increasing from 0 to 26.65 mW, while $P_{ex}$ is fixed at 1.03 mW and 0.59 mW for Pr2 and Pr1, respectively. Scale bar: 3 μm. **f** The curves of the linewidth versus the intensity of inhibition beam for LC-MPL and LMC-MPL under different $P_{ex}$ ($K_{S1}$ and $K_{Sn}$ represent the respective inhibition coefficient). **g** The CD lines for LC-MPL (Pr1) and LMC-MPL (Pr2) correspond to points C and D in Fig. 3f. Scale bar: 1 μm. **h** Schematic diagram of different light confining paths. Inhibition path: STED-$K_{S1}$ (S$_1$ to S$_0$ path), STED-$K_{Sn}$ (S$_n$ to S$_1$ path), TSA-$K_{T1}$ (T$_1$ to T$_n$ path). $K_{T1}$ represents the inhibition coefficient of TSA-$K_{T1}$. **i** Energy level transitions for LMC-MPL (3hv three-photon absorption, IC internal conversion, VR vibrational relaxation, ISC inter-system crossing, Q quencher, [Q]*: excited quencher, R*: free radical, Q–R non-reactive adduct formed by free radicals and quencher).

four lithography strategies are shown in Fig. 4a, by which the advantages of LMC-MPL can be clearly presented. As a result, from 139 nm for MPL to 55 nm for MC-MPL, the average linewidth is narrowed by 60.4%, while that is only 13.7% for LC-MPL (120 nm), indicating that TEMPO seems to be more efficient than the inhibition beam (Fig. 4b). Even so, inhibition beam is essential to achieve an average linewidth reduction from 55 to 30 nm for LMC-MPL. Besides, a FWHM profile in Fig. 4b shows that LMC-MPL has a minimum CD of 29.2 nm. In terms of LR, 300 nm is almost the limit of MPL and LC-MPL (Fig. 4c and Supplementary Fig. 13a) in this case due to the combined negative effect of the optical diffraction barrier and proximity effect. Thanks to the elimination of the proximity effect by TEMPO, MC-MPL exhibits a better LR of 175 nm (Supplementary Fig. 13b), which has been further improved to a remarkable 100 nm in LMC-MPL (Fig. 4c). Furthermore, we also gave the CD (20 nm) and LR (80 nm) of suspended line for LMC-MPL in Supplementary Fig. 14.

Subsequently, we proposed a mathematical model to reveal the impact of the quencher and inhibition beam on CD and LR. In this

model, we only consider the termination of free radicals by quenching molecules and ignore the quenching effect of oxygen since interfering factors can be eliminated by controlling variables in our controlled experiment. In free radical photopolymerization systems, the distribution of free radicals determines the degree of curing in the exposed area. Once the free radicals in a tiny region allow the photoresist to cure to a specific threshold, the region can be retained after development. Thus, both CD and LR are essentially under the control of free radicals. For simplicity, we create a three-dimensional coordinate system (Supplementary Fig. 15a), and the concentration distribution of free radicals in the exposed area can be considered to follow the excitation beam power, i.e. Gaussian distribution, as shown in Eq. (1).

$$R(x) = R_0 \exp\left(-\frac{x^2}{2}\right) \qquad (1)$$

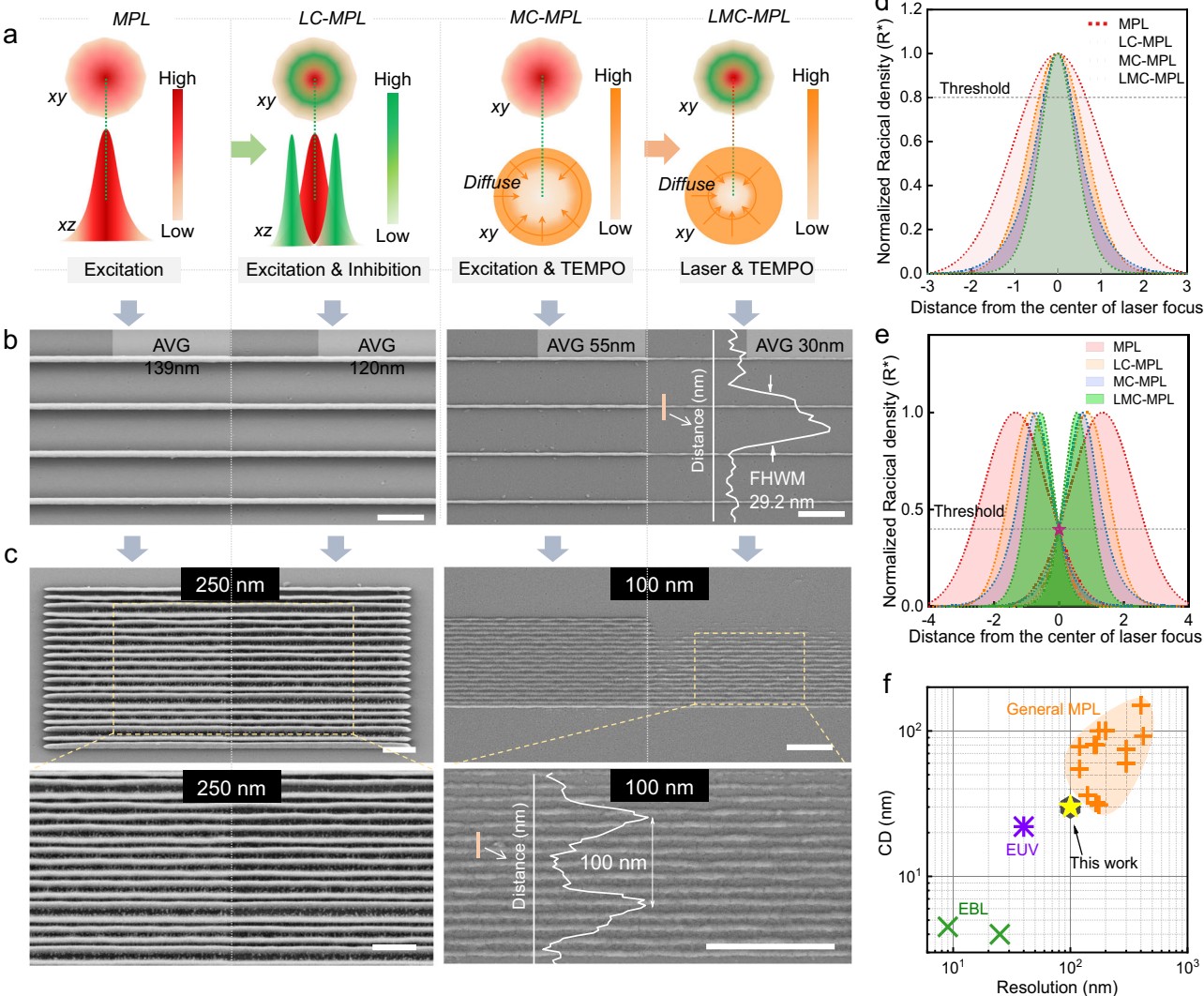

**Fig. 4 | Performance (linewidth and resolution) comparison and mathematical modeling. a** Schematic diagram of the laser power distribution and quencher distribution in the exposure area of MPL, LC-MPL, MC-MPL, and LMC-MPL. MPL (no TEMPO and inhibition beam off), LC-MPL (no TEMPO and inhibition beam on), MC-MPL (with TEMPO and inhibition beam off), LMC-MPL (with TEMPO and inhibition beam on). *xy*: *xy* cross-section, *yz*: *yz* cross-section. **b** The average (AVG) linewidth fabricated via the four lithography strategies and a profile of the lines in LMC-MPL (FHWM full width at half maximum). Scale bar: 1 µm. The inhibition beam powers were set as the respective optimal values of LC-MPL (2.23 mW) and LMC-MPL

(9.65 mW). $P_{ex}$ = 1.05 mW, $v$ = 50 µm s$^{-1}$. **c** Lateral resolution comparison of the four lithography strategies. The manufacturing parameters are the same as above. Scale bar: 1 µm. **d** Simulation results of the spatial distribution of free radicals in the exposure area of four lithography strategies ($\alpha$ = 0.7 and $\beta$ = 0.92). **e** Simulation results of the LR according to the free radical's distribution for four lithography strategies ($\alpha$ = 0.7 and $\beta$ = 0.92). **f** Statistical results of CD and LR of MPL[18,44–54] compared with electron beam lithography (EBL)[7,55] and extreme ultraviolet lithography (EUV)[8]. All the data listed here are the results of lines written on the substrate, excluding the suspended lines[56].

where $R_0$ is the free radical concentration at the center of the excitation beam, and $R(x)$ represents the concentration of free radicals at the point $x$ (Supplementary Fig. 15b).

For LC-MPL, the inhibition beam also can be seen simply as a Gaussian beam with $0-2\pi$ vortex phase delay. A doughnut-shaped spot is formed at the focus position so that the inhibition process occurs at the tail of the focal beam (Fig. 4a). Considering the inhibition beam, the model describing the $R(x)$ for LC-MPL was built by introducing a correction factor $\kappa(x) = \exp(-\alpha x^2)$ to Eq. (1)[43], and thus leads to Eq. (2).

$$R(x) = R_0 \exp\left(-\frac{x^2}{2}\right) \exp\left(-\alpha x^2\right) \quad (2)$$

$\alpha$ is proportional to the intensity of the inhibition beam without considering the SPA of the inhibition beam. As the value of $\alpha$ increases,

the distribution of $R(x)$ becomes narrower and narrower (Supplementary Fig. 16).

For MC-MPL, what cannot be ignored is the quenching effect and diffusion of the quencher[11]. Assuming the concentration distribution of the quencher is non-equilibrium and quasi-stationary, we introduced another factor $\varphi(x)$ into Eq. (1) to correct the formula[11,29]. So, $R(x)$ of MC-MPL is

$$R(x) = \frac{R_0 \exp\left(-\frac{x^2}{2}\right)}{k_Q Q_0 \left[1 - \beta\sqrt{\frac{\pi 1}{2x}} \operatorname{erf}\left(\frac{x}{\sqrt{2}}\right)\right]} \quad (3)$$

where $k_Q$ is the kinetic constant of termination by the quencher, $Q_0$ is the initial concentration of the quencher, and $\beta \propto 1/Q_0 D_Q$. $D_Q$ is the diffusion coefficient of the quencher. More details are discussed in the supporting information (Supplementary Section A).

Finally, the correction factors $\kappa(x)$ and $\varphi(x)$ were introduced to MPL simultaneously for LMC-MPL. So, the equation for LMC-MPL is found as

$$R(x) = \frac{R_0 \exp\left(-\frac{x^2}{2}\right)\exp(-\alpha x^2)}{k_Q Q_0 \left[1 - \beta\sqrt{\frac{\pi}{2x}}\mathrm{erf}\left(\frac{x}{\sqrt{2}}\right)\right]} \qquad (4)$$

The distribution of free radicals in the exposed area of four lithography strategies are compared intuitively in Fig. 4d, where $\alpha = 0.7$ and $\beta = 0.92$. Setting a dimensionless value of 0.8 as the cured threshold of photoresist, the CD got by LMC-MPL is the thinnest one (Fig. 4d). Similarly, Fig. 4e gives the theoretical simulation of radical distribution between two lines for the different lithography strategies. Considering the proximity effect, it is assumed that free radical concentration that just hits the threshold should be 0.4 at the middle position between two lines. Thus, LMC-MPL will gain a minimum LR as well. Therefore, the mathematical model can qualitatively illustrate LMC-MPL can achieve high-precision result, and in full agreement with experimental ones.

Figure 4f shows the CD and LR of general MPL reported in recent years (Supplementary Table 3)[18,44–54]. Obviously, our work provides the-state-of-the-art lithography performance (CD = 30 nm, LR = 100 nm) in the field of MPL, and the gap with EBL/EUV is getting smaller and smaller[7,8,55]. To verify the possibility of applying LMC-MPL in integrated circuits, we conducted the on-silicon pattern transfer process, including LMC-MPL, etching, and stripping (Supplementary Fig. 19). An optical photograph of one 1-inch silicon-based grating array is given, illustrating that the cured photoresist via LMC-MPL has satisfactory resistance to etching, allowing for pattern transfer on silicon. Further, the CD of the transferred silicon line up to 52 nm, and it keeps good quality at $L = 250$ nm and CD = 75 nm (Supplementary Fig. 19). Currently, higher precision silicon lines cannot be achieved because the cured photoresist of about 30 nm is knocked away by the gas during the etching process due to the absence of a suitable tackifier.

## Discussion

Here, we report a strategy, light, and matter co-confined multiphoton lithography (LMC-MPL) to overcome the inevitable optical diffraction barrier and proximity effect via the combination of photo-inhibition and chemical quenchers. By screening quenchers, TEMPO, with the smallest molecular weight, showed the best quenching effect due to its faster diffusion and higher active site freedom. It was proved that TEMPO plays the role of matter confinement through three quenching paths: static quenching dynamic, quenching, and direct reaction with free radicals. Besides, it was newly discovered that DETC can generate free radicals through not only a low-energy level photoexcitation path $(S_1 \rightarrow T_1)$ but also a high-energy level photoexcitation path $(S_n \rightarrow T_n)$. On this basis, the photoinhibition mechanism is considered as a two-step-STED process $(S_n \rightarrow S_1$ and $S_1 \rightarrow S_0)$. This proposed photo-inhibition mechanism is of great significance for the further understanding of photo-inhibition multi-photon lithography. More importantly, the establishment of the mathematical modeling reveals that the synergy of photoinhibition and quencher can obtain the narrowest distribution of free radicals, thereby gaining the highest precision and lateral resolution. By using LMC-MPL, we improve the critical dimension and lateral resolution to 30 and 100 nm, respectively, which further shortens the gap between EBL and EUV lithography. Besides, LMC-MPL is capable of fabricating excellent 3D structures and realizing high-precision pattern transfer on wafers, allowing it to fabricate nanoscale components for optoelectronics and integrated circuits.

Nevertheless, the improved lithography precision by LMC-MPL is at the expense of the sensitivity of the photoresist to some extent. This will not only increase the energy consumption of the manufacturing but also be harmful to high-speed and large-area lithography. Therefore, developing photoresists with both high sensitivity and precision is one of the future directions in MPL.

## Methods
### Materials
All reagents were purchased from commercial sources without further purification unless otherwise specified. Tricyclodecane dimethanol diacrylate (TCDA, 99%) was purchased from Sartomer Co., Ltd. A resin mixture (EBPFDA–OPPEA, 99%) contained ethoxylated bisphenyl fluorene diacrylate (EBPFDA) and o-phenyl phenoxyethyl acrylate (OPPEA) was supported by KPX Chemical Co., Ltd. 7-diethylamino-3-thenoylcoumarin (DETC, >99.9%), N-tert-Butyl-α-phenylnitrone (PBN, >99.9%) and 5,5-Dimethyl-1-pyrroline N-oxide (DMPO, >99.9%) were purchased from J&K Chemical and the quenchers, Q-1(2,2,6,6-Tetra-methylpiperidinooxy, TEMPO, >99.9%), Q-2 (4-Hydroxy-2,2,6,6-tetra-methyl-piperidinooxy, >99.9%), Q-3 (TEMPO Methacrylate, >99.9%), Q-4 (Bis(2,2,6,6-tetramethyl-4-piperidyl-1-oxyl)Sebacate, >99.9%), Q-5 (Tri-(4-hydroxy-TEMPO) phosphite, >99.9%) were supplied by Tokyo Chemical Industry. 4-Methoxyphenol (MEHQ, >99.9%) and 2-(Dimethylamino)ethyl methacrylate (DMAE-MA, >99.9%) were purchased from Energy Chemistry. All the other solvents were gained from Tansoole.

### Photoresist preparation
A very simple method that directly solves the initiator (DETC) and the quencher (TEMPO) into the cross-linker resin was adopted to prepare the photoresist. The resin is prepared by mixing 87.5 wt% TCDA and 12.5 wt% EBPFDA-OPPEA[19]. After adding 1 wt% (accounting for the total mass of the resin) DETC into the resin, the Pr1 was made successfully. Pr2, Pr3, Pr4, Pr5 and Pr6 was prepared by introducing 1.28 wt% Q-1, 1.42 wt% Q-2, 1.98 wt% Q-3, 2.10 wt% Q-4 and 1.5 wt% Q-5 into Pr1, respectively. All the photoresists contained the same concentration of nitroxide radicals.

### Photo-physical characterization
UV–vis absorption spectra are conducted on an ultraviolet–visible near-infrared spectrometer (UH5700, Hitachi). A transient fluorescence spectrometer (FLS920, Edinburgh Instrument) is utilized to acquire the fluorescence emission spectrum and fluorescence quantum efficiency of the photoresists. The fluorescence lifetime is investigated on another Edinburgh FLS1000 Fluorescence Spectrometer. To eliminate the influence of solvents, all the tested substances were dissolved in the monomer of the photoresist (87.5 wt% TCDA + 12.5 wt% EBPFDA-OPPEA), and all these samples were placed in quartz cuvettes (ex-situ) and tested in air.

### Electron spin resonance (ESR) spectroscopy
Electron spin resonance (ESR) spectroscopy was recorded using a Bruker EMX plus spectrometer operating at a standard cavity with 100 kHz modulation frequency. TEMPO (0.2 mM) and DETC (1 mM) were dissolved in dichloromethane for testing. The ESR experiments are conducted in the air.

### Lithography system
It involves two laser beams, one is an excitation femtosecond laser (525 nm, 80 MHz, 120 fs, TEMA-DUO-100, AVESTA), and another is an inhibition laser (532 nm, 80 MHz, 624 ps, VisUV-532-HP, PicoQuant GmbH). The inhibition beam is modulated by the vortex wave plate (VPP) and then coincides with the center of the excitation beam via the dichromic mirror (DM). Besides, the scan speed can be adjusted by the galvanometer scanner (GS, CTI, 8310K, Lexington, MA), while the switch and power of the excitation and inhibition are all controlled by an acoustic optical modulator (AOM, MT110-A1.5-Vis, AA Opto-Electronic).

## Lithography process

The experimental sample is prepared by dipping a photoresist onto a clean substrate (glass) and attaching it to a piezoelectric stage (PI, P-563.3CD, Karlsruhe, Germany). The oil is dropped on the other side of the glass, and a high numerical aperture objective (Olympus, UPlan-XApo, ×100, NA = 1.45) is then immersed into the oil. The fabrication path of the target structure is controlled by software. After photolithography, the sample was developed in PGMEA for 6 min, then IPA for 2 min. The polymerization threshold ($P_{th}$) is defined as the minimum excitation beam power required for the lines to remain after development. A standard threshold test array where the scan speed and the laser power increase respectively in perpendicular directions. The scan speed varies from 5 to 5000 $\mu m\,s^{-1}$, and the range of laser power is about 0–6 mW. For MPL and MC-MPL tests, the inhibition beam keeps off, while it will switch on as needed in LC-MPL and LMC-MPL. The linewidth of MPL that changes with laser power is acquired by another standard line array, where the laser power gradually increases. To determine the optimal inhibition beam power of the photoresist, a photo-inhibition test was adopted. At a fixed scan speed (50 $\mu m\,s^{-1}$) and an excitation power near the threshold, the inhibition beam is applied with gradually increasing power (0–26.65 mW). All these processes are carried out in a normal air atmosphere in our clean room.

## 3D structure manufacture

Unlike the lithography of lines, the writing of 3D structures uses a Dip-in mode, that is, the microscope is directly immersed in the photoresist.

## Pattern transfer

Reactive ion etching of a fabricated grating array is performed on a wafer-deep etching system (STS, MUC21, UK). The C4F8 (190 SCCM) and SF6 (450 SCCM) were chosen as a reaction gas. Soak in acetone for 12 h and sonicate in isopropanol for 4 h to complete the removal of the cured photoresist.

## Morphological characterization

The structural morphology obtained after development is observed by scanning electron microscopy (SEM, Zeiss, Sigma 300) after coating with 2 nm platinum by a GEVEE-TECH GVC-2000 sputter coater.

## Computational methods

Quantum chemical calculations are carried out in the frame of density functional theory (DFT). The molecule structure of DETC in the ground state and triplet state has been fully optimized. The calculated results are presented by the Becke 3-parameter hybrid functional with Lee–Yang–Parr correlation (B3LYP) and the 6-311G(d) basis set, as implemented in Gaussian 16.

# Data availability

All data are presented in the Article and the Supplementary Information. Source data are provided with this paper. The data are available from the authors on request. Source data are provided with this paper.

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

## Acknowledgements

This work was supported by the National Natural Science Foundation of China (62125504 to C.K., 22105180 to C.C.), National Key R&D Program of China (2021YFF0502700 to C.K.), "Pioneer" and "Leading Goose" R&D Program of Zhejiang Province (2023C01186 to C.C., 2023C01051 to C.K.), Major Program of Natural Science Foundation of Zhejiang Province (LD21F050002 to C.K.), and Zhejiang Provincial Ten Thousand Plan for Young Top Talents (2020R52001 to C.K.).

## Author contributions

L.G.: Investigation, data curation, and analysis, writing—original draft, writing—review & editing. C.C.: Methodology, data analysis, funding acquisition, writing—original draft, writing—review & editing. Xi Liu: Original idea, initial experiments. Y.Q., X.W. and H.L.: Investigation. Q.L., Q.S., D.Z., and C.D.: Optical systems. Z.Y.: Software. C.K. and Xu Liu: Supervision, funding acquisition, and resources.

## Competing interests

The authors declare no competing interests.
