## [Peer Review File · Nature Communications]

REVIEWER COMMENTS

Reviewer #1 (Remarks to the Author):

In their manuscript "Light and Matter Co-confined Multi-photon Lithography", Guan et al. demonstrate the use and benefits of combining chemical quenching and optical depletion techniques in order to enhance the linewidth ("lateral resolution") and resolution ("critical dimension") in 2D and 3D multi-photon lithography.

In the first part of their work, they compare six different photoresin compositions by the achieved minimum linewidth. Five of the six resins contain various quencher molecules, all of which are based on the molecule TEMPO. They show that the overall smallest linewidth is obtained with the photoresin that contains the bare TEMPO molecule. All photoresins make use of DETC as photoinitiator. In contrast to most ordinary photoinitiators, optically excited DETC molecules can be optically depleted by green light. In the second part of their manuscript, the authors investigate the quenching mechanism of TEMPO. The authors propose three quenching pathways: static complex formation (path 1), dynamic quenching via energy transfer (path 2), and radical scavenging (path 3). The authors use a bouquet of techniques to investigate the quenching mechanisms: electron spin resonance, fluorescence lifetime measurement, fluorescence emission spectroscopy, and a Stern-Volmer analysis.

Subsequently, in the third part of the manuscript, the authors show that the minimum linewidth and the resolution of the printing process can be improved further by depleting the peripheral focal region using a donut-shaped picosecond-pulsed focused laser beam. The authors show that one-photon absorption of the depletion beam leads to an increased linewidth for the quencher-free photoresin, but not for the quenched photoresin. In this part, the authors also discuss the optical excitation pathways of DETC.

In the fourth part of the manuscript, the authors show a simple mathematical model, seeking to describe the observed effects of chemical quenching and optical depletion. Furthermore, the authors show an etched silicon grating and compare their work to the results of previous multi-photon lithography, EUV-, and DUV-lithography.

To my knowledge, combining depletion with chemical quenching (apart from oxygen), has not been published so far. Furthermore, investigating the mechanisms by spectroscopic methods is a valuable tool in order to better understand the underlying processes. Furthermore, the community is still discussing the excitation mechanisms in DETC (for instance, consider this recent preprint <https://doi.org/10.21203/rs.3.rs-1797484/v1>). Hence, the contents of the paper could be of interest for the multi-photon 3D printing community.

However, the paper needs extensive rework before publishing. Some critical aspects are raised in the comments below. In the current manuscript, important details for reproduction are missing (printing setup details, see below). Furthermore, some claims are not backed by sufficient evidence (resolution improvement in 3D, see below). The authors completely neglected the absorption of TEMPO and the effect of oxygen in their analysis (see below), which limits or even flaw the conclusions, in my opinion. While the spectroscopic methodologies (ESR and Fluorescence) are well established, they were used ex-situ (in cuvettes with collimated light) and are therefore of limited use to draw conclusions for the 3D printing process (in droplet with tightly focused light).

COMMENTS

#1

Overall, the paper touches many topics instead of focusing on one or two. In my opinion, showing the on-silicon gratings does not add value to the paper. Furthermore, the photochemistry of DETC is touched upon, based on DFT calculations. The photochemistry of DETC is complicated and the results of this paper are not compared and aligned to the results of previous publications. The photochemical experiments (Stern-Volmer, fluorescence lifetime,...) should be expanded and discussed in more detail. For instance, a selection of different photoinitiators could be probed. Moreover, the fluorescence and ESR experiments were all performed ex-situ in glass cuvettes. The results of the fluorescence and ESR experiments cannot be directly carried over to the in-situ case (oxygen, diffusion lengths, timescales).

#2

The used 3D benchmark structure ("Nezha") is very uncommon in the field of multi-photon 3D printing. A much better and quantitative benchmark structure is a woodpile photonics crystal, as shown, e.g., in Fischer, Joachim, and Martin Wegener. "Three-dimensional direct laser writing inspired by stimulated-emission-depletion microscopy." *Optical Materials Express* 1.4 (2011): 614-624.

Focused-ion beam cross sections would be highly desirable in order to quantify and compare the obtained resolution in 3D with previous publications.

#3

The excitation wavelength of 520 nm is mentioned only in the Materials section. In multiphoton lithography, 800 nm is a commonly used wavelength. In order to compare the present results to previous results, this aspect must be disclosed in the main part. Likewise, the numerical aperture of the used objective lens should be mentioned in the main part.

The methods section should be enhanced to reveal more details of the performed experiments. For instance, it is completely unclear if the Stern-Volmer-experiments were performed with the photoresist contained in a glass cuvette.

#4

The mechanistic investigation of TEMPO is (presumably) performed in cuvettes and not in-situ (i.e., under focused-light conditions). This limits the scope of the results heavily. In-situ, quenchers can easily diffuse from a quasi-infinite reservoir outside the laser focus into the laser focus – where quenchers are consumed. In a cuvette that is exposed to a collimated light source, the diffusion length scales and timescales are completely different. This aspect is not addressed in the paper at all.

#5

It is well known that oxygen plays a crucial role in multi-photon 3D printing. See for instance Zandrini, T., et al. "Effect of the resin viscosity on the writing properties of two-photon polymerization." *Optical Materials Express* 9.6 (2019): 2601-2616.

However, oxygen is not mentioned once in the entire paper. It is completely unclear, if the characterization experiments (Stern-Volmer, Fluorescence lifetime,...) were performed in a deoxygenated atmosphere.

#6

The authors write "Atsushi Taguchi et al. used a femtosecond laser with a 400 nm wavelength for MPL". This has been done before by

Wickberg, Andreas, et al. "Second-Harmonic Generation by 3D Laminate Metacrystals." *Advanced Optical Materials* 7.14 (2019): 1801235.

#7

The authors should provide references for the following claims:

“Unfortunately, due to the inevitable power fluctuation and focus drift in the system, the line-edge-roughness (LER) of the CD is not ideal, which is intolerable for further pattern transfer.”

“It is worth noting that the quencher will be consumed rapidly in the exposed area, leading to a continuous diffusion of the quencher, which will further suppress the proximity effect.”

#8

It is unclear what the authors refer to as “decent lateral resolution” in “More importantly, it seems impossible to realize a decent LR since the proximity effect is still there.”

#9

The rationale behind the choice of the used monomers is unclear. Commonly used monomers include pentaerythritol triacrylate or tetraacrylate.

#10

In Figure 1d, the achieved line width for the different photoresist is shown. However, the laser power is kept constant. In my opinion, this is an unfair comparison in the scope of the paper. Preferably, one would compare the laser power for which the smallest linewidth is consistently printable.

#11

In Fig. S1a, it appears as if smoothing splines are used. Splines can be easily confused with actual data and should therefore be avoided.

#12

The paper uses the terms “matter confining capability” and “light confining capability” throughout. However, these terms are never defined and, presumably, these terms cannot be quantified. Therefore, one should refrain from statements like these:

“The matter confining capability of quenchers is inversely proportional to their relative molecular weight...”

#13

The numbers in the micrograph Fig S2 are mirrored and therefore hard to read. Furthermore, it is unclear in what steps the laser power has been increased.

#14

The causality in the following statement is unclear and should be explained clearly:

“but likewise raise the damage power due to its matter confining capability,”

#15

Typo: “multiphoton adsorption” should be “multiphoton absorption”. There are several further typos, please revise carefully.

#16

The abbreviations “PBN” and “DMPO” are nowhere defined.

#17

To really discern between dynamic and static quenching, the Stern-Volmer plots should also be shown for the fluorescence lifetime at different quencher concentrations.

#18

The mechanism of complex formation between ground-state DETC and TEMPO is unclear. Any explanation is welcome.

#19

It is unclear if the fluorescence decay spectra were measured in a diluted solution.

#20

It is shown that the ESR-signal of TEMPO + DETC decreases upon irradiation (Fig. 2g). Does the ESR-signal recover after some time? Does the ESR-signal also decrease for a DETC-free photoresin?

#21

The authors write "A higher optimal Pin of LMC-MPL could be attributed to the quenching path 1 in Pr2. Due to the presence of the TEMPO, almost all the S1-state DETC that excited via SPA of inhibition beam". I am confused: path 1 is depicted as ground-state depletion mechanism and does not interact with the S1-state molecule. This error seems to occur throughout the text, making some parts of the text hard to understand.

#22

The laser foci are nowhere characterized. Intensity cross-sections are highly desirable. For instance, see Fischer, Joachim, and Martin Wegener. "Three-dimensional direct laser writing inspired by stimulated-emission-depletion microscopy." *Optical Materials Express* 1.4 (2011): 614-624.

#23

Fig. 3e: the blue shading obscures important parts of the micrograph. Furthermore, the ordering of the panels in the figure is confusing (neither vertically nor horizontally ordered).

#24

TEMPO also absorbs light at 520 nm wavelength. No absorption spectrum of TEMPO is shown. Is the photochemistry of TEMPO in any way affected by the absorption of light? TEMPO also absorbs the used depletion beam. How does this influence the shown results?

#25

In Fig. 4, the results of the manuscript are compared to previous results. It is unclear where the datapoints of previous results originate from and how the datapoints for EUV and EBL were calculated. References are required.

#26

In the methods, critical details of the used lasers are missing. For instance, the repetition rate and pulse length of the picosecond laser is unclear. Furthermore, the model and vendor of the laser ins not provided. Likewise for the AOM, galvanometer scanner, piezo stage, and rotating scanner.

#27

In the methods section, it is written that the 3D structures were writing in dip-in mode. The paper does not explain how the authors took care of refractive index matching.

Reviewer #2 (Remarks to the Author):

The manuscript is of great interest to the community and is a timely topic worthy of further research. In this manuscript, the authors presented a new strategy of light and matter co-confined multi-photon lithography (LMC-MPL) to improve the resolution of MPL. The authors realized better critical dimension (CD) and lateral dimension (LR) performance of 30 nm CD and 100 nm LR. The authors discussed the mechanisms of matter confined-MPL (MC-MPL) and light confined-MPL (LC-MPL). They further demonstrated the model to explain the process in LC-MPL. Mathematical model is illustrated to describe the combined influence of quencher and photo-inhibiter. The influence of different quenchers were explored. In addition, the authors conducted experiments to demonstrate the high-resolution and 3D

printing capability of LMC-MPL. While others have shown demonstrations of STED lithography and the effects of quenchers separately, the effects of both have not been explored. Hence, the manuscript exhibits a sufficient level of novelty with new insights. One of the main concerns I have for this manuscript is the definition and demonstration of high resolution. In lithography, resolution is defined as the minimum pitch that structures can be patterned. Minimum feature size or CD is not a good measure of resolution as one can simply reduce the exposure dose and space structures really far apart. Thus, it would be important for the authors to show free-standing structures with minimum possible pitch, e.g. woodpile structures. Gratings printed on substrates might not show the actual resolution as the voxel can be partially embedded in the substrate, which could give the impression of a better resolution than what the system can actually achieve. Below are some detailed suggestions to improve the manuscript:

1. The authors should state clearly the definition of resolution, i.e. differentiate critical dimension and lateral resolution.
2. What is the size of the hair lines in the Fig.1e models? And can the hair lines in 3D keep the same CD and LR as the same in 2D? What is the purpose of showing them printed in smaller sizes? The minimum feature sizes are still very large.
3. The authors tested the different dose and writing speed. The critical dimension is easy to achieve by lowering the dose. But it is not a good measure of resolution. To show high resolution, the authors should look at the minimum pitch.
4. The claimed minimum resolution is 100 nm, but there is no figure in the manuscript to support this. The result should be placed in the manuscript. In Fig. S7, what are the fabrication parameters (dose and writing speed) to realize the 100 nm resolution. From the SEM, the line is close to be discontinuous, is it caused by the low dose?
5. In addition to MC-MPL, LC-MPL and LMC-MPL, a post processing shrinking method is also effective in reducing pitch and CD, e.g. Liu Y, Nature Commun., 2019, 10(1): 4340. As this method does not require modification of the resin composition nor multi-wavelength laser alignment, it is a compelling approach to achieve higher resolution structures and should be discussed.
6. In Fig.2a, the manuscript mentions that the Quenching path 1 is the energy transfer from s1-DETC to TEMPO. This description is confusing and does not match the illustration in Fig.2a.
7. There has been some other works on MC-MPL in which they utilized different quenchers. It is better to cite some of these papers to compare the performances of the quenchers in this paper with others. Is the quencher in this work better?
8. The explanation of quenching path 2 mechanism is not clear. Besides, which path among the 3 quenching paths takes the most significant and efficient influence on the quenching process by using this DETC-TEMPO material?
9. Can LMC-MPL keep the same fine fabrication in 2.5D or 3D? If can, it will be better to show a 2.5D model in SEM to demonstrate the ability.
10. Is it possible to fabricate free standing structures and what is the smallest pitch it can achieve?
11. What is the influence on the quenching process once the inhibition beam is turned on?
12. Is the lateral resolution (LR) mainly influenced and improved by the light-confinement process?
13. In Fig.4a LC-MPL, the second scale bar is not correct. Typo in line 250.
14. In terms of highest resolution patterns achieved using EBL, it is worth taking a look at Yang J K W, J. Vac. Sci. Technol., 2009, 27(6): 2622-2627. in which they realize 9 nm pitch in HSQ. Again, it is the pitch that matters, and not CD. It is important to clarify that in the references 47, Saifullah et al. shows 4-nm

feature size, but the minimum pitch is large. There is a difference between resolution and minimum feature size. A minimum feature size is not difficult to realize. However, packing them closely is what matters.

15. Fig 4e would be clearer shown in log coordinates instead of the linear coordinates. And this figure is not complete, e.g. please see Wang S, Nano Futures, 2018, 2(2): 025006. In this work, they realized 7 nm width nano-webs by two-photon lithography.

16. There are some repeated figures, and it will be better to reorganize the figures in the manuscript for clarity.

Reviewer #3 (Remarks to the Author):

The paper describes the benefit of combining light-confined or matter-confined MPL to achieve higher resolution in resists. The synergistic effects of appropriate quenchers is described which gives some indication in which direction future research could go. The results were mainly achieved by screening quenchers, and TEMPO showed the best quenching effects. Furthermore a two-step-STED process was identified, that enhances resolution further. It is good that in the conclusion the shortcomings of the new method are presented, that indicate that the method requires more sensitive resists that would allow higher throughput.

Overall, the paper is a good overview about the combination of excitation and inhibition by the MPL method, which derives from STED, and therefore helps to understand the underlying effects. However, although the language is mostly very scientific, there is a certain sloppiness that would need to be improved. First, some terms are not well defined, or at least it is assumed that everybody knowing the state of the art understands them, e.g., critical dimension (CD) and line resolution (LR). Second, there are sometimes assumptions that are simply not true, or not precise: The resolutions presented are certainly achievements that go beyond the state of the art of MPL. However, when comparing with conventional lithographies, the authors compare pears and apples, i.e., their results with published results of DUV and EUV resolutions achieved in production. For me this is sloppiness, but it could also be interpreted as overselling results.

Abstract: What means cheaper and readily available in this sense? With current resist systems (and pulsed lasers), can we aim for high throughput comparable or even better than conventional lithography? Instead of cheap, use "low cost" and refer to what kind of cost you mean (production with 200 wafers/h or simply a research tool that can be used instead of EBL or another MPL? Since it is in the abstract, it seems an assumption that does not need proof here, but if you claim this in the introduction, a reference or at least the basis for a comparison is needed - and I think that this has to be specified in the introduction. And since the sensitivity will be lower, will there ever be a chance to be high throughput? Higher resolution also means - in a serial process - writing of more lines and thus longer writing time.

Why only referencing LC-MPL 12, and not STED by Stefan Hell?

Describe difference between critical dimension and line resolution, is the latter the half pitch? Or is it the distance between resist lines? The terms are not generally used and should be defined (best would be in a figure).

MPL and LC-MPL performance of the two photoresists were performed, respectively." Is there something "better" than a synergistic effect of subsequent LC-MPL and MC-MPL processes?

40: Reducing the wavelength of excitation laser is an effective method. Yes, this is a general approach, why is it not used here? And right, we probably do not have commercial resists for 0.5x157nm, 0.5x193nm resists?

49: "inevitable power fluctuation and focus drift in the system". This means the system used? Well, this could be improved and is therefore not a fundamental problem?

51: what is a decent LR? 140 nm is great, depending on the point of view. Better: A LR below 100 nm or comparable to conventional DUV lithography.

55: Definition of the diffusion-based proximity effect!

Typical resist manufacturers count on the effect of diffusion, otherwise crosslinking and chemically amplified resists would not be possible!

64: a continuous diffusion of quencher molecules from other, non-depleted areas.

67: I guess that LMC-MPL also has limitations in terms of CD or LR, simply different ones?

68: "Achieving stable sub-50 nm linewidth and 100 nm resolution simultaneously." What does this mean? Sub-50 nm CD as isolated lines and 100 nm resolution (half pitch) of dense lines? And why do you mention at the same time, does this mean in the same resist, with the same process parameters?

74: What does "To demonstrate LMC-MPL has a better synergistic effect than using light-confined or matter-confined alone, MPL and LC-MPL performance of the two photoresists were performed, respectively." Is there something "better" than a synergistic effect of subsequent LC-MPL and MC-MPL processes?

284: The comparison with EUV lithography seems somehow odd to me. Is this a comparison with current production CD and linewidths or such obtained in laboratory? then rather write that this is comparable with CDs and linewidths currently employed in production using DUV and EUV litho. Are there references for DUV and EUV proving the claim? Also for minimum resolution.

317: Generally use μm , if possible, not um .

390: "is not conductive"? What does this mean? And can you present a factor of sensitivity decrease?

439 What means (0-0-26.65 mW)?

466: CRediT?

Do the quenchers interact chemically with the initiators? What would be the case if initiators with higher sensitivity would be used?

What if you write two times over the same resist? Larger areas composed of adjacent narrow lines. Would you then switch off the inhibition beam to merge lines?

Fig. 2: b and g Guass? -> Gauss? Better say "Magnetic field strength"

Figure 3:

a) barely to see details in this image resolution

e) Piont C, Does it mean Point C?

The overall slide seems to busy to me, it contains SEM micrographs, evaluations, and schematics. Why not taking a and b together, then c, d and then e to g and then h and i?

Showing the 3D Nezza figures is nice (the resolution in the presented version for review is too low to see major difference), but there is one shortcoming: The "writing element" is a voxel, that typically is an extended ellipsoid that is longer in z than in xy. Can the authors provide the real size of the voxel?

Fig. 4e: Is LMC-MPL then better than or comparable to EUV, as mentioned in the summary/conclusion 383 "By using LMC-MPL, we improve the critical dimension and lateral resolution to 30 nm and 100 nm, respectively, which are comparable to DUV and EUV lithography."? OK, with some tolerance, but the actual resolution of EUV achieved in labs is higher (7 nm). I do not see this. Where is DUV?

References: Are the first relevant publications [1,2] really from 2018 and 2021? There must be earlier references on MPL.

Point-to-point Response to Reviewers

Reviewer #1 (Remarks to the Author):

In their manuscript “Light and Matter Co-confined Multi-photon Lithography”, Guan et al. demonstrate the use and benefits of combining chemical quenching and optical depletion techniques in order to enhance the linewidth (“lateral resolution”) and resolution (“critical dimension”) in 2D and 3D multi-photon lithography. In the first part of their work, they compare six different photoresin compositions by the achieved minimum linewidth. Five of the six resins contain various quencher molecules, all of which are based on the molecule TEMPO. They show that the overall smallest linewidth is obtained with the photoresin that contains the bare TEMPO molecule. All photoresins make use of DETC as photoinitiator. In contrast to most ordinary photoinitiators, optically excited DETC molecules can be optically depleted by green light. In the second part of their manuscript, the authors investigate the quenching mechanism of TEMPO. The authors propose three quenching pathways: static complex formation (path 1), dynamic quenching via energy transfer (path 2), and radical scavenging (path 3). The authors use a bouquet of techniques to investigate the quenching mechanisms: electron spin resonance, fluorescence lifetime measurement, fluorescence emission spectroscopy, and a Stern-Volmer analysis. Subsequently, in the third part of the manuscript, the authors show that the minimum linewidth and the resolution of the printing process can be improved further by depleting the peripheral focal region using a donut-shaped picosecond-pulsed focused laser beam. The authors show that one-photon absorption of the depletion beam leads to an increased linewidth for the quencher-free photoresin, but not for the quenched photoresin. In this part, the authors also discuss the optical excitation pathways of DETC. In the fourth part of the manuscript, the authors show a simple mathematical model, seeking to describe the observed effects of chemical quenching and optical depletion. Furthermore, the authors show an etched silicon grating and compare their work to the results of previous multi-photon lithography, EUV-, and DUV-lithography. To my knowledge, combining depletion with chemical quenching (apart from oxygen), has not been published so far. Furthermore, investigating the mechanisms by spectroscopic methods is a valuable tool in order to better

understand the underlying processes. Furthermore, the community is still discussing the excitation mechanisms in DETC (for instance, consider this recent preprint <https://doi.org/10.21203/rs.3.rs-1797484/v1>). Hence, the contents of the paper could be of interest for the multi-photon 3D printing community. However, the paper needs extensive rework before publishing. Some critical aspects are raised in the comments below. In the current manuscript, important details for reproduction are missing (printing setup details, see below). Furthermore, some claims are not backed by sufficient evidence (resolution improvement in 3D, see below). The authors completely neglected the absorption of TEMPO and the effect of oxygen in their analysis (see below), which limits or even flaw the conclusions, in my opinion. While the spectroscopic methodologies (ESR and Fluorescence) are well established, they were used ex-situ (in cuvettes with collimated light) and are therefore of limited use to draw conclusions for the 3D printing process (in droplet with tightly focused light).

Response:

We appreciate the reviewer's positive comments and constructive suggestions. We have supplemented the experiments as requested by the reviewers and discussed them accordingly in the revised manuscript. First, we have added detailed parameters about our setup in the revised manuscript for reproduction, including excitation laser, inhibition laser, objective lens and some other devices (acoustic optical modulator, galvanometer scanner, piezoelectric stage). Meanwhile, more detailed experimental procedures are disclosed in the section of **Methods**. Second, the testing of woodpile photonic crystal has been added in the revised manuscript (Supplementary **Fig. 7** and Supplementary **Fig. 8**) to verify the resolution improvement in 3D by LMC-MPL (details see below). Third, the UV-vis absorption spectra of TEMPO dissolved in the monomer has been given in Supplementary **Fig. 5b**, demonstrating the absorption of TEMPO does not affect the multiphoton absorption and photoinhibition processes. Fourth, we gave the explanation of the effects of oxygen and ex-situ testing below as well. Details see below: **Response #1-#27..**

1. Overall, the paper touches many topics instead of focusing on one or two. In my opinion, showing the on-silicon gratings does not any add value to the paper. Furthermore, the photochemistry of DETC is touched upon, based on DFT calculations. The photochemistry of DETC is complicated and the results of this paper are not compared and aligned to the results of previous publications. The photochemical experiments (Stern-Volmer, fluorescence lifetime,...) should be expanded and discussed in more detail. For instance, a selection of different photoinitiators could be probed. Moreover, the fluorescence and ESR experiments were all performed ex-situ in glass cuvettes. The results of the fluorescence and ESR experiments cannot be directly carried over to the in-situ case (oxygen, diffusion lengths, timescales).

Response:

We appreciate the comments. The focus of our work is mainly on the proposal of a new strategy (LMC-MPL) to realize a high-precision and high-resolution lithography. And it is worth emphasizing that the acquisition and transfer of high precision patterns on silicon substrates is the first step for producing diverse micro devices, including semiconductor manufacturing. The experiment of on-silicon gratings certified the possibility of applying our strategy in above fields. That's why we put this part into the previous manuscript. Of course, that's not the point of this work. Therefore, as request by the reviewer, we have put this part in the Supplementary Information (**Supplementary Fig. 19**).

In the manuscript, we proposed that the photochemistry of DETC may generate radicals through two paths ($S_1 \rightarrow T_1$ and $S_n \rightarrow T_n$) according to the experiments, and the DFT calculations certified the probability of this energy level transition. In fact, we are not the first to make this point. Martin Wegener et al once proposed that "when using DETC without a co-initiator, polymerization is initiated via two competing pathways, one of which obeys an $N = 3$ scaling and the other follows an $N = 4$ scaling. While the first one can be depleted (as it passes through S_1 and T_1), the other one cannot be interrupted, as it passes neither the relaxed S_1 state nor the T_1 state (Fig. R1)." [Advanced Optical Materials. 3, 221-232 (2015)]. This is exactly consistent with our

experiments and mechanistic explanation. But, Martin Wegener et al believe that the two polymerization paths of DETC are achieved by three photon absorption and four photon absorption because they observed that a different nonlinearity absorption exponent ($N=3$ and $N=4$) versus repetition rate. Martin Wegener et al also got a result of $N>3$ for PETA+DETC in a more recent paper (**Fig. R2**) [*Advanced Optical Materials. 7, 1901040 (2019)*], but this time they attributed $N=4$ to the absorption of PETA monomer. Therefore, based on the above analysis and our own results (**Fig. 1g**), we are more inclined to think that all the absorption of photon for DETC are achieved by a three-photon absorption process. And subsequently it has two paths to go. One is it may transition from S_n to T_n state (ISC, $S_n \rightarrow T_n$) and generate free radicals at T_n state, and the other is it may return to S_1 state through vibrational relaxation (VR), then transition to T_1 state (ISC, $S_1 \rightarrow T_1$), and finally generate free radicals at T_1 state. We also cited these two references in the revised manuscript and added a brief discussion (**highlight in blue, Page10**).

[REDACTED]

Fig. R1 Adapted molecular model explaining the new observations by Martin Wegener et al. [*Advanced Optical Materials. 3, 221-232 (2015)*]

[REDACTED]

Fig. R2 The testing of multiphoton absorption coefficient (threshold power versus exposure time) for PETA with different concentrations of DETC. [*Advanced Optical Materials*. 7, 1901040 (2019)]

As for the photochemical experiments of DETC, we have further tested ESR spectra of TEMPO and TEMPO+DETC before and after UV irradiation (placed in the dark for 0 min, 20 min or 60 min) in **Fig. 2c**. Additionally, we also supplemented the fluorescence decays spectra of the photoresists (DETC) with different concentration (0-40 mM) of TEMPO in **Fig. 2f** and the Stern-Volmer plots of fluorescence lifetime of DETC in **Fig. 2g**. The linear change of the lifetime indicates that dynamic quenching dominates at lower concentration of TEMPO (**Fig. 2g**). In the case of higher concentrations of TEMPO, the two processes (static and dynamic quenching) may be competitive, which results in a nonlinear relationship of lifetime and fluorescence. The combination of static and dynamic quenching almost completely quenched the fluorescence of Pr2 (**Supplementary Fig. 5a** and **Fig. 2d**). These further discussions about the photochemical process have been discussed in the revised manuscript (**highlight in blue, Page7**).

DETC (7-diethylamino-3-thenoylcoumarin) was chosen as the photoinitiator in our photoresist because the initiator must not only have photoinhibition properties but also be wavelength-matched to our lasers. Up to now, there are only few relatively mature initiators with photoinhibition properties in MPL, such as DETC and ITX (Isopropyl thioxanthone) [*Laser & Photonics Reviews*. 16, 2100229 (2022)]. So, we don't have much choice for initiators with photoinhibition property. As for why we didn't choose ITX as another research object, because the photophysical and photochemical mechanism of ITX and DETC are not exactly same, such as their different performance in the pump-probe experiments (**Fig. R3**) [*Opt. Lett.* 36, 3188-3190 (2011)]. Therefore, the study on the interaction between DETC and TEMPO in this paper has its own uniqueness and the research on other photoinitiators is of little significance to the discussion and research based on DETC in this paper. That is why our entire work focuses on DETC and does not discuss other photoinitiators. Of course, we believe that using quenchers as a

tool to study the photophysical and photochemical mechanisms of other photoinitiators, such as ITX, would be another feasible and interesting work.

[REDACTED]

Fig. R3 The different performance of ITX (left) and DETC (right) in the pump-probe experiments. (Time-dependent change of the optical density (OD) after 387.5 nm excitation for different probe wavelengths as indicated). [*Opt. Lett.* **36**, 3188-3190 (2011)]

The fluorescence and ESR experiments were indeed performed ex-situ which has been emphasized in the revised manuscript (**highlight in blue, Page7**) and **Methods (Photo-Physical characterization)** section. In fact, we had initially hoped to conduct an in situ test as well, but were not successful. We have not found commercial instruments for us to perform the in-situ fluorescence and ESR experiments, since such experiments are extremely rare and demanding. We have also tried our best to build our own instruments for in-situ fluorescence testing within the recent three months, but we failed. We found that not only did it take a long time, but it was so expensive that we couldn't afford it. As for in-situ ESR test, it is theoretically not feasible because

the laser beam will be focused into a point inside the photoresist and the radical distribution generated at this point cannot be tested by the current ESR instrument. In addition, although the ex-situ fluorescence and ESR tests cannot take the influence of oxygen (diffusion lengths and timescales) into account, this does not negate our use of ex-situ tests to explore the quenching mechanism of TEMPO. It can only be said that our ex-situ test cannot take into all the factors during the in-situ lithography, and it can indeed partially reflect the influence of TEMPO on the fluorescence and ESR signal of DETC. And these effects must be taken into account in the analysis of the in-situ MPL.

2. The used 3D benchmark structure ("Nezha") is very uncommon in the field of multi-photon 3D printing. A much better and quantitative benchmark structure is a woodpile photonics crystal, as shown, e.g., in Fischer, Joachim, and Martin Wegener. "Three-dimensional direct laser writing inspired by stimulated-emission-depletion microscopy." *Optical Materials Express* 1.4 (2011): 614-624. Focused-ion beam cross sections would be highly desirable in order to quantify and compare the obtained resolution in 3D with previous publications.

Response:

We appreciate the comments. First of all, we would like to explain the reasons for choosing "Nezha" for testing. On the one hand, we want to demonstrate the 3D fabrication capability of LMC-MPL. On the other hand, we want to qualitatively compare the lithography results of the three strategies (MPL, MC-MPL and LMC-MPL). As requested by reviewer, the woodpile photonic crystal has been added in the revised manuscript. We conducted two experiments. The first one is to compare the woodpile structures (the line period is fixed at 300 nm) of MPL, MC-MPL and LMC-MPL exactly under the same test conditions (fixed scan speed and laser power) in **Supplementary Fig. 7**. It shows LMC-MPL can achieve a finest photonic crystal (41 nm) compared with MC-MPL (78 nm) and MPL (260 nm). The second experiment is to continuously reduce the line period (d) between two lines for obtaining the minimum resolution of MPL, MC-MPL and LMC-MPL (**Supplementary Fig. 8**). It can be seen LMC-MPL can obtain the highest resolution of (LR=200 nm), which cannot be reached by MC-MPL (LR=400 nm) and MPL (LR=250 nm). As an alternative

method of focused-ion beam to investigate the cross sections of photonic crystal, we used feasible SEM to observe the side of the photonic crystal (with the x-y plane tilted 45°, **Supplementary Fig. 7**), by which we can also quantify and judge the resolution.

Supplementary Fig. 7 (a) Schematic of a 3D woodpile structure with lateral period d and axial period $\sqrt{2}d$. (b) Schematic diagram of voxels, where a represents the size of the voxel along the y direction, b represents the size of the voxel along the z direction). (c) SEM morphology of a woodpile structure ($d=300$ nm) made by MPL, MC-MPL and LMC-MPL with 1.12 mW excitation beam power, 9.65 mW inhibition beam power and $50 \mu\text{m s}^{-1}$ writing speed. The images above (A, B and C) are taken perpendicular to the xy plane (scale bar: 300 nm), and images below (D, E and F) are taken with the xy plane tilted 45° (scale bar: 1 μm). At a period of $a=300$ nm, the lines of woodpile structure obtained by MPL cannot be separated, so the size of the voxel cannot be measured. Therefore, while maintaining the same exposure dose (P_{ex} and scan speed), the period of the MPL is expanded to $a=700$ nm (inset of D image), thereby obtaining the voxel size in the MPL mode.

Supplementary Fig. 8 SEM morphology of woodpile structures with different period (d) made by MPL, MC-MPL and LMC-MPL (1.10 mW excitation beam power, 9.65 mW inhibition beam power and $50 \mu\text{m s}^{-1}$ writing speed). Scale bar: 300 nm.

3. The excitation wavelength of 520 nm is mentioned only in the Materials section. In multiphoton lithography, 800 nm is a commonly used wavelength. In order to compare the present results to previous results, this aspect must be disclosed in the main part. Likewise, the numerical aperture of the used objective lens should be mentioned in the main part. The methods section should be enhanced to reveal more details of the performed experiments. For instance, it is completely unclear if the Stern-Volmer-experiments were performed with the photoresist contained in a glass cuvette.

Response:

We appreciate the comments. We have added these parameters of excitation beam (525 nm, 80 MHz, 120 fs, TEMA-DUO-100, AVESTA), inhibition beam (525 nm, 80 MHz, 120 fs, TEMA-DUO-100, AVESTA) and objective lens (100x, 1.4 NA), galvanometer scanner (GS, CTI, 8310K,

Lexington, Massachusetts), acoustic optical modulator (AOM, MT110-A1.5-Vis, AA OPTO-ELECTRONIC), piezoelectric stage (PI, P-563.3CD, Karlsruhe, Germany) in the revised manuscript (**highlight in blue below Fig. 1 and Fig. 3**) and **Methods (Lithography System)**. Meanwhile, the Methods section have been enhanced with more details of the performed experiments (**highlight in blue, page 15-16**). For Stern-Volmer-experiments, different amounts of DETC were dissolved in the photoresist, and further placed in quartz cuvettes for the tests of fluorescence emission, fluorescence lifetime, fluorescence quantum efficiency and UV-vis absorption.

4. The mechanistic investigation of TEMPO is (presumably) performed in cuvettes and not in-situ (i.e., under focused-light conditions). This limits the scope of the results heavily. In-situ, quenchers can easily diffuse from a quasi-infinite reservoir outside the laser focus into the laser focus – where quenchers are consumed. In a cuvette that is exposed to a collimated light source, the diffusion length scales and timescales are completely different. This aspect is not addressed in the paper at all.

Response:

We appreciate the comments, and this is a very good question. Mechanistic investigation of TEMPO is mainly based on fluorescence (intensity, lifetime), ESR and UV-vis absorption tests. These tests are indeed performed ex-situ, which has been emphasized in the revised manuscript (**highlight in blue, Page7**) and **Methods (Photo-Physical characterization)**.

As we mentioned above, we had initially hoped to conduct an in situ test as well, but were not successful. In-situ fluorescence or UV-vis absorption tests need to integrate lithography equipment and spectroscopy equipment. As far as we know, there are no mature commercial instruments for us to perform the in-situ fluorescence and ESR experiments, since such experiments are extremely rare and demanding. We have also tried our best to build our own instruments for in-situ fluorescence testing within the recent three months, but we failed. We found that not only did it take a long time, but it was so expensive that we couldn't afford it. As for in-situ ESR test, it is theoretically not feasible because the laser beam will be focused into a point inside the photoresist and

the radical distribution generated at this point cannot be tested by the current ESR instrument.

Even so, although the ex-situ fluorescence and ESR tests cannot take the influence of oxygen (diffusion lengths and timescales) into account, this does not negate our use of ex-situ tests to explore the quenching mechanism of TEMPO. It can only be said that our ex-situ test cannot take into all the factors during the in-situ lithography, but it can indeed partially reflect the influence of TEMPO on the fluorescence and ESR signal of DETC. And these effects must be taken into account in the analysis of the in-situ MPL.

5. It is well known that oxygen plays a crucial role in multi-photon 3D printing. See for instance: Zandrini, T., et al. "Effect of the resin viscosity on the writing properties of two-photon polymerization." *Optical Materials Express* 9.6 (2019): 2601-2616. However, oxygen is not mentioned once in the entire paper. It is completely unclear, if the characterization experiments (Stern-Volmer, Fluorescence lifetime,...) were performed in a deoxygenated atmosphere.

Response:

We appreciate the comments. All the lithography process and the characterization experiments (Stern-Volmer, Fluorescence lifetime, UV-vis absorption and ESR) were performed in air. We have added this point to **Methods (Lithography Process)** section. As a free radical quencher, oxygen plays a crucial role in free radical photopolymerization, which especially cannot be ignored in the study of polymerization kinetics. These has been discussed in many references, such as [Advanced Materials. 26, 6566-6571 (2014)], [Macromol. Theory Simul. 2006, 15, 176–182] and [Optical Materials Express 9.6 (2019): 2601-2616] (as the reviewer mentioned). It is believed that oxygen only interact with free radicals that have already been generated, but is not involved in the generation process of free radical from DETC. While oxygen and TEOMPO are competitive for free radical quenching, the amount of oxygen within the laser focus is extremely limited (even when diffusion is taken into account), and is not an order of magnitude compared to TEMPO.

In addition, the experiment we did were a series of controlled tests (four lithography strategies: MPL, LC-MPL, MC-MPL and LMC-MPL), and all the

lithography tests were performed in air atmosphere (**Fig. 4**). Therefore, from the perspective of control variabilities, the influence of oxygen can be excluded from the comparison results. So, we consider our results and discussion to be credible. We have also added a description about oxygen in the revised manuscript (**highlight in blue, Page 13**).

6.The authors write “Atsushi Taguchi et al. used a femtosecond laser with a 400 nm wavelength for MPL”. This has been done before by Wickberg, Andreas, et al. "Second-Harmonic Generation by 3D Laminate Metacrystals." *Advanced Optical Materials* 7.14 (2019): 1801235.

Response:

We appreciate the comments. In the previous manuscript, we quote the reference published by Atsushi Taguchi to illustrate that reducing the wavelength of excitation laser is an effective method to overcome the optical diffraction barrier. Atsushi Taguchi et al emphasized the “Deep-Ultraviolet Photolithography” and explored the accuracy (80 nm) and resolution (160 nm). Although Wickberg Andreas et al used a laser with 405 nm wavelength earlier than Atsushi Taguchi, they did not discuss the influence of wavelength on the writing accuracy in detail. Their focus is more on the second-harmonic generation of a photonic crystal. Therefore, we did not modify the text “Atsushi Taguchi et al. used a femtosecond laser with a 400 nm wavelength for MPL” in the part of Introduction, but we cited the reference [*Advanced Optical Materials* 7.14 (2019): 1801235] in the revised manuscript.

7.The authors should provide references for the following claims: “Unfortunately, due to the inevitable power fluctuation and focus drift in the system, the line-edge-roughness (LER) of the CD is not ideal, which is intolerable for further pattern transfer.” “It is worth noting that the quencher will be consumed rapidly in the exposed area, leading to a continuous diffusion of the quencher, which will further suppress the proximity effect.”

Response:

We appreciate the comments. In the original manuscript, the complete expression is “It is possible to create remarkable 36 nm CD ($\lambda/14.8$) on the sub-diffraction scale with a 140 nm LR by LC-MPL. Unfortunately, due to the inevitable power fluctuation and focus drift in the system, the line-edge-roughness (LER) of the CD is not ideal, which is intolerable for further pattern transfer.” Here, we cited the work reported by Minfei He et al. [*Photonix*. 3, 25 (2022)]. In their work, although 140 nm resolution had been achieved, the line-edge-roughness were very poor (**Fig. R4**). And, they proposed that “we attributed linewidth instability to insufficient mechanical strength, laser power fluctuations, and optical system drift as reported in many previous studies”. So, this claim just quotes the views in the original paper. Maybe our expression here is not clear enough, so we have modified the expression here.

As for “It is worth noting that the quencher will be consumed rapidly in the exposed area, leading to a continuous diffusion of the quencher, which will further suppress the proximity effect.”, this claim can be supported by the two references: [*ACS Nano*. 6, 2302-2311 (2012)] and [*AIP Advances*. 5, 127215 (2015)]. In the reference of [*ACS Nano*. 6, 2302-2311 (2012)], the author proposed that “Due to diffusion, the consumption of the quencher in the irradiated volume can be compensated by its transfer from nonirradiated domains.” And they gave a model to illustrate how the quencher influence the threshold (**Fig. R5**). We have added the references into the revised manuscript.

[REDACTED]

Fig. R4 The 140 nm resolution results obtained by Minfei He et al. using LC-MPL. [*Photonix*. 3, 25 (2022)]

[REDACTED]

Fig. R5 (a) Schematic representation of conversion profiles in threshold polymerization regime for different irradiation doses. (b) Schematic representation of conversion profiles in a model of stationary quencher diffusion for different irradiation times and fixed irradiation intensity. [ACS Nano. 6, 2302-2311 (2012)]

8. It is unclear what the authors refer to as “decent lateral resolution” in “More importantly, it seems impossible to realize a decent LR since the proximity effect is still there.”

Response:

We appreciate the comments. For “decent lateral resolution”, what we want to say is “excellent lateral resolution” or “better lateral resolution”. Since this can be confusing, so we changed “decent lateral resolution” to “better lateral resolution” in the revised manuscript. Besides, we have also added the definition of CD and LR in the revised manuscript (**highlight in blue, page1**). Among them, CD represents the minimum achievable linewidth of an isolated line and LR means the minimum grating period between two separated lines [Nature Photonics. 15, 932-938 (2021)]. In order to allow readers to understand more intuitively, we have given a schematic diagram in the Supplementary Information (**Supplementary Fig. 12**).

Supplementary Fig. 12 The schematic diagram of CD and LR.

9. The rationale behind the choice of the used monomers is unclear. Commonly used monomers include pentaerythritol triacrylate or tetraacrylate.

Response:

We appreciate the comments, this is a very good question. The reason for monomer selection is to obtain a photoresist with suitable refractive. The refractive index of TCDA (that we used) and EBPFDA-OPPEA (that we used) are 1.503 and 1.608 respectively. Through adjusting the ratio of the two monomers, we can achieve a photoresist with a refractive index of 1.518, which exactly matches the refractive index of our objective lens. This will help improve the resolution of MPL. Once the laser beams are focused into a photoresist with mismatching refractive index, the focus will still be distorted due to the presence of optical aberration, which could give rise to a decrease in the resolution. We had explored the issue in detail in our previous work [*ACS Applied Materials & Interfaces*. 14, 31332-31342 (2022)]. In addition, we have explained the reason in the revised manuscript (**highlight in blue, below Fig. 1**) and cited the reference so that interested readers can learn more about it.

10. In Figure 1d, the achieved line width for the different photoresist is shown. However, the laser power is kept constant. In my opinion, this is an unfair comparison in the scope of the paper. Preferably, one would compare the laser power for which the smallest linewidth is consistently printable.

Response:

We appreciate the comments. The purpose of this experiment in **Fig. 1d** is to compare the inhibition capability (matter confining) of different quenchers, so

that we can choose the best one. In fact, under the same test conditions (laser power and scan speed), allows for a very intuitive comparison of the matter confining capacity of different quenchers. Therefore, it is a controlled experiment instead of an unfair comparison.

In addition, we have also investigated and compared the smallest linewidth of the photoresists with different quenchers. As shown in **Supplementary Fig. 2**, Pr2 also exhibit the smallest linewidth, demonstrating that TEMPO has the best matter confining capability among these quenchers. But, it is worth emphasizing that the difference between the smallest linewidth of the photoresists, is just on the scale of several nanometers. The extreme difference of this set of data in **Supplementary Fig. 2** is only 11 nm. Further considering the systematic errors (laser power fluctuations and optical system drift), the method of comparing the smallest linewidth of the photoresists is unreliable. Therefore, we still retain the line width comparison data under the same power in the revised manuscript, rather than using the limit line width for comparison.

Supplementary Fig. 2 Smallest linewidth of Pr1, Pr2, Pr3, Pr4, Pr5, and Pr6 at their threshold powers respectively (writing speed: $50 \mu\text{m s}^{-1}$). (b) And the corresponding SEM images. Scale bar: 500 nm.

11. In Fig. S1a, it appears as if smoothing splines are used. Splines can be easily confused with actual data and should therefore be avoided.

Response:

We appreciate the comments. We have changed the splines (**Supplementary Fig. 1a** in the revised manuscript) to straight line.

Supplementary Fig. 1 (a) The threshold power (P_{th}) curves of Pr1, Pr2, Pr2, Pr4, Pr5 and Pr6 at different writing speed. P_{th} is defined as the minimum laser power that will allow the photoresist to be retained after development (b) SEM images of the linewidth for Pr3, Pr4, Pr5, and Pr6. Scale bar: 1 μm .

12. The paper uses the terms “matter confining capability” and “light confining capability” throughout. However, these terms are never defined and, presumably, these terms cannot be quantified. Therefore, one should refrain from statements like these: “The matter confining capability of quenchers is inversely proportional to their relative molecular weight...”

Response:

We appreciate the comments. We have given the definition of the terms “matter confining capability” and “light confining capability” in the revised manuscript. Here, matter confining capability refers to the ability of quenchers to inhibition polymerization at the edge of the focal spot by chemical and physical quenching process (**highlight in blue, Page3**) in MPL. Light confining capability refers to the ability of inhibition beam to hinder polymerization at the edge of the focal spot via STED process (**highlight in blue, Page8**). While this definition is not a proprietary term, it is sufficient for the reader to clearly understand its meaning in our work

In addition, as requested by reviewer, we have modified the above statements in the revised manuscript to “*The inhibition capability of quenchers is inversely proportional to the relative molecular weight of the quenchers.*”

(highlight in blue, Page4), to qualitatively describe the relationship between the matter confining capability of the quenchers and their molecular weight.

13.The numbers in the micrograph Fig S2 are mirrored and therefore hard to read. Furthermore, it is unclear in what steps the laser power has been increased.

Response:

We appreciate the comments. The mirrored tag is due to a small bug in our writing program. We have manually marked the numbers (scan speed) in the revised manuscript (**Supplementary Fig. 4**), so that the readers can obtain the information more intuitively. However, we cannot give the step value of the laser power (mW) in **Supplementary Fig. 4**, because we increase the laser power by controlling the voltage (Voltage step: 0.01 V, voltage start: 0 V, the number of steps: 100 times). And it is not a linear relationship between the voltage and laser power, So, the step of laser power is not a fixed value. Therefore, we gave a corresponding plot between laser power and voltage in **Supplementary Fig. 4c**. Here the laser power in our system is adjustable, so the values of the voltage and power will not always maintain the corresponding relationship above in **Supplementary Fig. 4c**. But the nonlinear trend of this line remains unchanged.

Supplementary Fig. 4 Threshold test of (a) MPL and (b) MC-MPL at different writing speed (5-5000 $\mu\text{m s}^{-1}$) and laser power. The laser power is controlled by the voltage (from right to left: 0-1 V, Step: 0.01 V). In the damaged area, the photoresist cannot undergo controlled polymerization, but will be directly destroyed by the laser. The adjacent boundary power between the damage

area and the processing window is defined as the damage power. The processing window is defined as the range of excitation laser power between the damage power and the threshold power. Scale bar: 10 μm . (c) A corresponding plot between laser power and voltage. Here the laser power in our system is adjustable, so the values of the voltage and power will not always maintain the corresponding relationship above.

14. The causality in the following statement is unclear and should be explained clearly: “but likewise raise the damage power due to its matter confining capability,”

Response:

We appreciate the comments. This statement is indeed not stated clearly enough. Before understanding the statement, some definitions must first be clarified: the threshold power (P_{th}) is defined as the minimum excitation beam power required for the lines to remain after development (**highlight in blue, below Fig. 1**). In the damaged area, the photoresist cannot undergo controlled polymerization, but will be directly destroyed by the laser (**below Supplementary Fig. 4**). The adjacent boundary power between the damage area and the processing window is defined as the damage power (**below Supplementary Fig. 4**). The processing window is defined as the range of excitation laser power between the damage power and the threshold power (**below Supplementary Fig. 4**). Based on these, what we want to express here is the quencher not only make the P_{th} of MC-MPL higher than MPL, but also raise the damage power of MC-MPL. The simultaneous increase in P_{th} and damage power of MC-MPL, resulted in little change in the width of the entire processing window. We have modified the statement to “... *but also raise the damage power of MC-MPL. The simultaneous increase in P_{th} and damage power result in little impact on the width of processing window.....*” in the revised manuscript (**highlight in blue, Page5**).

15. Typo: “multiphoton adsorption” should be “multiphoton absorption”. There are several further typos, please revise carefully.

Response:

We appreciate the comments, we've corrected the mistakes in the revised manuscript.

16.The abbreviations “PBN” and “DMPO” are nowhere defined.

Response:

We appreciate the comments. We are sorry for missing the definition of them. The full names of PBN (N-tert-Butyl- α -phenylnitron) and DMPO (5,5-Dimethyl-1-pyrroline N-oxide) have been added in the revised manuscript (**Methods section** and below **Fig. 2** as well).

17.To really discern between dynamic and static quenching, the Stern-Volmer plots should also be shown for the fluorescence lifetime at different quencher concentrations.

Response:

We appreciate the comments. The fluorescence lifetime (τ) of DETC at different quencher concentrations have been given (**Fig. 2f-g** and **Fig. R6**) in the revised manuscript. With the increasing of concentration of TEMPO, the lifetime of DETC decreased obviously. The lifetime is invariant with the concentration of analyte if only static quenching exists. On the contrary, the fluorescence lifetime should diminish as quencher is added with dynamic quenching. Therefore, the changes in fluorescence lifetime further confirmed the existence of dynamic quenching. In addition, the linear change of the lifetime indicates that dynamic quenching exists dominates at lower concentration of TEMPO (**Fig. 2g** and **Fig. R6**). In the case of higher concentrations of TEMPO, the two processes (static and dynamic quenching) may be competitive, which results in a nonlinear relationship of lifetime and fluorescence [*Journal of the American Chemical Society. 125, 3821-3830 (2003)*]. We have also added these discussions into the revised manuscript (**highlight in blue, Page7**)

Fig. R6 Fluorescence decays spectra of the photoresists (DETC) with different concentration (0-40 mM) of TEMPO (left). The Stern-Volmer plots (right) of fluorescence lifetime for quenching process of DETC by TEMPO in photoresist, where τ_0 and τ is the fluorescence lifetime in the presence and absence of quenchers.

18. The mechanism of complex formation between ground-state DETC and TEMPO is unclear. Any explanation is welcome.

Response:

We appreciate the comments. First, whether the nonlinearity of the Stern-Volmer plots of fluorescence intensity and fluorescence lifetime, or the reduction of the ESR peak of TEMPO due to the presence of ground state DETC, all prove the existence of static quenching. And it had been reported that “In static quenching a complex is formed between the fluorophore and the quencher, and this complex is nonfluorescent.” [Lakowicz, J. R. Principles of Fluorescence Spectroscopy; Plenum Press: New York, 1986]. Unfortunately, how the complexes are formed and the specific molecular structures of complex are not reported in this reference, since the complex is difficult to detect by existing means. In our work, the fluorescence quenching of DETC by TEMPO has been consistent with the above literature, but the formation mechanism of the complex between ground DETC and TEMPO is really difficult for us to investigate at present. As shown in **Supplementary Fig. 5b**, the introduction of TEMPO has almost no effect on the UV-vis absorption spectrum of DETC, indicating that TEMPO does not change the conjugation system of DETC. Therefore, this static quenching is likely to be from some weak interaction. We

surmise that there two kinds of weak interaction between them: 1. van der Waals force, which is an ordinary force. For example, the motion of electrons in a TEMPO molecule generates a transient dipole moment, which transiently polarizes neighboring DETC molecules, and the polarized DETC in turn enhances the transient dipole moment of the TEMPO molecule. This force often needs to be manifested when two molecules are in very close proximity, which is consistent with the concentration dependence of TEMPO. 2. shielding effect of TEMPO, TEMPO may act as a polarizing molecule, high concentration TEMPO may shield the DETC molecule, making the two DETC molecules inaccessible and inhibiting the aryl-ring stacking effect, which can accelerate the rapid rotation and vibration of a single DETC molecule, resulting in a high non-radiative decay rate and hence fluorescence burst.

19. It is unclear if the fluorescence decay spectra were measured in a diluted solution.

Response:

We appreciate the comments. To eliminate interference factors as much as possible, all the tested substances for fluorescence were dissolved in the photoresist and all these samples were placed in quartz cuvettes for testing. We have also emphasized this point in the revised manuscript. (**Methods, Photo-Physical characterization**)

20. It is shown that the ESR-signal of TEMPO + DETC decreases upon irradiation (Fig. 2g). Does the ESR-signal recover after some time? Does the ESR-signal also decrease for a DETC-free photoresin?

Response:

We appreciate the comments. As requested by reviewer, we have performed more ESR experiments, as shown in **Fig. 2c**. It was found the peak intensity of TEMPO is significantly reduced when DETC is introduced (before UV), which proves that there is indeed interaction between ground state DETC and TEMPO. Interestingly, the peak intensity of TEMPO+DETC increases after UV irradiation, and it will gradually recover as the UV lamp is turned off. Maybe, the excitation of DETC by the UV light weakens the static interaction between

DETC and TEMPO. Once the UV lamp is turned off, the excited DETC returns to the ground state, so the intensity of the ESR peak will also recover.

Theoretically, when the UV light is turned on, the free radicals produced by the excited state DETC also deplete the TEMPO via quenching path 3 to reduce the ESR signal. Therefore, quenching path 1 and quenching path 3 have opposite effects on the ESR signal of the TEMPO, i.e., they are a competitive pair. Since the quantum efficiency of radical generation by excited DETC is less than 1, the effect of static quenching on the ESR signal of TEMPO should dominate, resulting in the experimental phenomenon in **Fig. 2c**. In addition, the ESR signal of TEMPO (DETC-free photoresin) keeps constant before and after UV irradiation, which excludes the impact of UV irradiation on TEMPO.

The relevant discussion also has been added in the revised manuscript. **(Highlight in blue, Page 6-7)**

21. The authors write "A higher optimal P_{in} of LMC-MPL could be attributed to the quenching path 1 in Pr2. Due to the presence of the TEMPO, almost all the S1-state DETC that excited via SPA of inhibition beam". I am confused: path 1 is depicted as ground-state depletion mechanism and does not interact with the S1-state molecule. This error seems to occur throughout the text, making some parts of the text hard to understand.

Response:

We appreciate the comments. We are sorry for making this mistake. Indeed, the quenching path 1 and 2 in previous manuscript don't correspond to that in **Fig. 2a**. We have corrected the error in the revised manuscript.

22. The laser foci are nowhere characterized. Intensity cross-sections are highly desirable. For instance, see Fischer, Joachim, and Martin Wegener. "Three-dimensional direct laser writing inspired by stimulated-emission-depletion microscopy." *Optical Materials Express* 1.4 (2011): 614-624.

Response:

We appreciate the comments. We have provided the real laser spots located near the x-y and x-z planes of excitation beam and inhibition beam in **Supplementary Fig. 6a and Fig. R7**.

Fig. R7. Real picture of laser spots located near the x-y and x-z planes of excitation beam and inhibition beam. Scale bar: 500 nm.

23. Fig 3e: the blue shading obscures important parts of the micrograph. Furthermore, the ordering of the panels in the figure is confusing (neither vertically nor horizontally ordered).

Response:

We appreciate the comments. We have removed the blue shading obscures in **Fig. 3e** so that readers can see the results clearly. In addition, we have rearranged the ordering of the images in **Fig. 3**, making them horizontally ordered.

24. TEMPO also absorbs light at 520 nm wavelength. No absorption spectrum of TEMPO is shown. Is the photochemistry of TEMPO in any way affected by the absorption of light? TEMPO also absorbs the used depletion beam. How does this influence the shown results?

Response:

We appreciate the comments. The UV-vis absorption spectra of TEMPO that dissolved in the photoresist has been given in **Supplementary Fig. 5b**. It can be seen clearly that TEMPO has almost no absorption at the wavelength of 350-550 nm. Additionally, the ESR test of TEMPO before and after UV irradiation shows no difference (**Fig. 2c**). So, we think the photochemistry of TEMPO is hardly affected by the absorption of light (including excitation beam and inhibition beam). Based on the analysis above, TEMPO does not absorb the depletion beam, so it does not affect the writing results from this aspect. The specific interaction process between TEMPO and DETC has been further discussed in the above response (**Response #18 and #20**) and the revised manuscript (**highlight in blue, Page 6-7**).

25. In Fig. 4, the results of the manuscript are compared to previous results. It is unclear where the data points of previous results originate from and how the datapoints for EUV and EBL were calculated. References are required.

Response:

We appreciate the comments. In fact, we had cited these references of the data points in **Fig. 4** in the original manuscript, including the references about EUV and EBL (**highlight in blue, Page 14**). In the body of the text, it may not be easy for the reader to notice. Therefore, we have also cited these references just below **Fig. 4f**. Apart from this, we have also listed these data and the references in the Supplementary Information (**Supplementary Table. 3**).

26. In the methods, critical details of the used lasers are missing. For instance, the repetition rate and pulse length of the picosecond laser is unclear. Furthermore, the model and vendor of the laser are not provided. Likewise, for the AOM, galvanometer scanner, piezo stage, and rotating scanner.

Response:

We appreciate the comments. As requested by reviewer, all the information about our system have been added in **Methods section (Lithography System)**: Femtosecond laser (525 nm, 80 MHz, 120 fs, TEMA-DUO-100, AVESTA), inhibition laser (532 nm, 80 MHz, 624 ps, VisUV-532-HP, PicoQuant GmbH), galvanometer scanner (GS, CTI, 8310K, Lexington, Massachusetts), acoustic optical modulator (AOM, MT110-A1.5-Vis, AA OPTO-ELECTRONIC), piezoelectric stage (PI, P-563.3CD, Karlsruhe, Germany). Our system uses galvanometer scanner (GS) instead of rotating scanner (RS). So, we have made changes, removing RS, in the revised manuscript and **Supplementary Fig. 6a**.

27. In the methods section, it is written that the 3D structures were written in dip-in mode. The paper does not explain how the authors took care of refractive index matching.

Response:

We appreciate the comments. As we mentioned in **Response #9**. The reason for monomer selection is to obtain a photoresist with suitable refractive. The refractive index of TCDA (that we used) and EBPFDA-OPPEA (that we used) are 1.503 and 1.608 respectively. Through adjusting the ratio of the two monomers (87.5 wt% TCDA + 12.5 wt% EBPFDA-OPPEA), we can achieve a photoresist with a refractive index of 1.518, which exactly matches the refractive index of our objective lens. This will help improve the resolution of MPL. Once the laser beams are focused into a photoresist with mismatching refractive index, the focus will still be distorted due to the presence of optical aberration, which could give rise to a decrease in the resolution. We had explored the issue in detail in our previous work [*ACS Applied Materials & Interfaces*. 14, 31332-31342 (2022)]. In addition, we have explained the reason in the revised manuscript (**highlight in blue, below Fig. 1**) and cited the reference so that interested readers can learn more about it.

We thank the reviewer again for the chance of revision/resubmission and also for the time and attention given to this manuscript.

Reviewer #2 (Remarks to the Author):

The manuscript is of great interest to the community and is a timely topic worthy of further research. In this manuscript, the authors presented a new strategy of light and matter co-confined multi-photon lithography (LMC-MPL) to improve the resolution of MPL. The authors realized better critical dimension (CD) and lateral dimension (LR) performance of 30 nm CD and 100 nm LR. The authors discussed the mechanisms of matter confined-MPL (MC-MPL) and light confined-MPL (LC-MPL). They further demonstrated the model to explain the process in LC-MPL. Mathematical model is illustrated to describe the combined influence of quencher and photo-inhibitor. The influence of different quenchers was explored. In addition, the authors conducted experiments to demonstrate the high-resolution and 3D printing capability of LMC-MPL. While others have shown demonstrations of STED lithography and the effects of quenchers separately, the effects of both have not been explored. Hence, the manuscript exhibits sufficient level of novelty with new insights. One of the main concerns I have for this manuscript is the definition and demonstration of high resolution. In lithography, resolution is defined as the minimum pitch that structures can be patterned. Minimum feature size or CD is not a good measure of resolution as one can simply reduce the exposure dose and space structures really far apart. Thus, it would be important for the authors to show free-standing structures with minimum possible pitch, e.g. woodpile structures. Gratings printed on substrates might not show the actual resolution as the voxel can be partially embedded in the substrate, which could give the impression of a better resolution than what the system can achieve actually. Below are some detailed suggestions to improve the manuscript:

Response:

We appreciate the reviewer's positive comments and constructive suggestions. First, we have added the definition of CD and LR in the revised manuscript (**highlight in blue, Page1**) and given a schematic diagram in the Supplementary Information (**Supplementary Fig. 12**). Among them, critical dimension (CD) represents the minimum achievable linewidth of an isolated line and lateral resolution (LR) means the minimum grating period between two separated lines [*Nature Photonics*. 15, 932-938 (2021)]. From this point of view,

the LR in our paper can corresponds to the pitch mentioned by the reviewer. We indeed compare the LR made by MPL, MC-MPL, LC-MPL and LMC-MPL (**Fig. 4c** and **Supplementary Fig. 13**) instead of just considering the CD. In addition, the feature size is indeed can be reduced by reducing the exposure dose just like **Fig. 1i** showing. But, the feature size will not decrease infinitely as the exposure dose decreases. There will be a threshold at a fixed scan speed, corresponding to generate a minimum linewidth (CD). In addition, the premise of obtaining higher resolution (smaller pitch) is to gain a sufficiently small CD. Therefore, the CD is an important parameter for MPL.

Supplementary Fig. 12 The schematic diagram of CD and LR.

Supplementary Fig. 13 Lateral resolution test of (a) MPL, LC-MPL, (b) (c)MC-MPL, and LMC-MPL with 1.05 mW excitation beam power, 9.65 mW inhibition beam power and 50 $\mu\text{m s}^{-1}$ writing speed. Scale bar: 1 μm .

At the reviewer's request, we have added the experimental results of woodpile structures in **Supplementary Fig. 7** and **Supplementary Fig. 8**. We conducted two experiments. The first one is to compare the woodpile structures (the line period is fixed at 300 nm) of MPL, MC-MPL and LMC-MPL exactly under the same test conditions (fixed scan speed and laser power) in **Supplementary Fig. 7**. it shows LMC-MPL can get a finest photonic crystal (41 nm) compared with MC-MPL (78 nm) and MPL (260 nm). The second experiment is to continuously reduce the line period (d) of the woodpile structure for obtaining the minimum resolution of MPL, MC-MPL and LMC-MPL (**Supplementary Fig. 8**). It can be seen LMC-MPL can obtain the highest resolution of (LR=200 nm), which cannot be reached by MC-MPL (LR=400 nm) and MPL (LR=250 nm). Apart from the woodpile structure, we also supplemented the CD (20 nm) and LR (80 nm) of suspended line for LMC-MPL (**Supplementary Fig. 14**).

Supplementary Fig. 7 (a) Schematic of a 3D woodpile structure with lateral period d and axial period $\sqrt{2}d$. (b) Schematic diagram of voxels, where a represents the size of the voxel along the y direction, b represents the size of the voxel along the z direction). (c) SEM morphology of a woodpile structure ($d=300$ nm) made by MPL, MC-MPL and LMC-MPL with 1.12 mW excitation beam power, 9.65 mW inhibition beam power and $50 \mu\text{m s}^{-1}$ writing speed. The images above (A, B and C) are taken perpendicular to the xy plane (scale bar: 300 nm), and images below (D, E and F) are taken with the xy plane tilted 45°

(scale bar: 1 μm). At a period of $a=300$ nm, the lines of woodpile structure obtained by MPL cannot be separated, so the size of the voxel cannot be measured. Therefore, while maintaining the same exposure dose (P_{ex} and scan speed), the period of the MPL is expanded to $a=700$ nm (inset of D image), thereby obtaining the voxel size in the MPL mode.

Supplementary Fig. 8 SEM morphology of woodpile structures with different period (d) made by MPL, MC-MPL and LMC-MPL (1.10 mW excitation beam power, 9.65 mW inhibition beam power and $50 \mu\text{m s}^{-1}$ writing speed). Scale bar: 300 nm.

Supplementary Fig. 14 The (a) CD (20 nm) and (b) LR (80 nm) of suspended line for LMC-MPL with 1.18 mW excitation beam power, 9.65 mW inhibition beam power and $50 \mu\text{m s}^{-1}$ writing speed. Scale bar: 200 nm.

In conclusion, the experimental results of both substrate lines and suspended lines can prove the superiority of LMC-MPL. Although the linewidth on the substrate is indeed affected by the position of focal plane, it is a controllable factor and the experimental results is repeatable. However, since the suspended line will inevitably shrink during the development process, it also has defects in reflecting the actual resolution while the lines on the substrate are not affected by such shrinkage.

28. The authors should state clearly the definition of resolution, i.e. differentiate critical dimension and lateral resolution.

Response:

We appreciate the comments. We have added the definition of CD and LR in the revised manuscript (**highlight in blue, Page 1**) and given a schematic diagram in the Supplementary Information (**Supplementary Fig. 12**). Among them, critical dimension (CD) represents the minimum achievable linewidth of an isolated line and lateral resolution (LR) means the minimum grating period between two separated lines [*Nature Photonics. 15, 932-938 (2021)*].

29. What is the size of the hair lines in the Fig.1e models? And can the hair lines in 3D keep the same CD and LR as the same in 2D? What is the purpose of showing them printed in smaller sizes? The minimum feature sizes are still very large.

Response:

We appreciate the comments. As shown in **Fig. R8**, the resolution of the hair lines of Nezha (40 μm height) is approximately between 270-450 nm and the resolution of the hair lines of Nezha with other sizes is proportionally reduced. The resolution of a 3D structure will reflect the actual voxel size of MPL, while the resolution of a 2D structure on a substrate will be affected by the position of the focal plane. Therefore, theoretically, 2D and 3D structures will have different CD and LR.

Fig. R8 Top view of the 40 μm tall model of Nezha. Scale bar: 2 μm .

The smaller size of the 3D structure, the higher the requirements for lithography resolution. Therefore, by comparing the small-size 3D writing results of MPL, MC-MPL and LMC-MPL, we can intuitively and qualitatively compare the writing resolution of these three methods in 3D structure. As for “the minimum feature sizes are still very large”, the 3D results here is, on the one hand, to show our strategy can be used for 3D structures, and on the other hand, it is just to qualitatively and intuitively compare the resolution of MPL, MC-MPL and LMC-MPL. So, the minimum feature size is not the pursued here. We used woodpile structure (**Supplementary Fig. 7** and **Supplementary Fig. 8**) and grating lines (**Fig.4b** and **Fig. 4c**) to further explore the minimum feature size and resolution of 3D and 2D structures in the revised manuscript.

30. The authors tested the different dose and writing speed. The critical dimension is easy to achieve by lowering the dose. But it is not a good measure of resolution. To show high resolution, the authors should look at the minimum pitch.

Response:

We appreciate the comments. According to our definition of critical dimension (CD) and lateral resolution (LR), the LR in our paper can corresponds to the pitch mentioned by the reviewer (**Supplementary Fig. 12**). Firstly, the feature size is indeed can be reduced by reducing the exposure dose, as showin in **Fig. 1i**. But, the feature size will not decrease infinitely as the exposure dose decreases. There will be a threshold (the minimum excitation laser power required for the lines to remain after development) at a fixed scan speed, corresponding to the critical dimension (CD). In addition, the

premise of obtaining higher resolution (smaller pitch) is to gain a sufficiently small CD. Therefore, the CD is indeed an important parameter for MPL.

In fact, we have compared the minimum resolution made by MPL, MC-MPL, LC-MPL and LMC-MPL (**Fig. 4c** and **Supplementary Fig. 13**) instead of just considering the CD in the manuscript. Among them, LMC-MPL can reach a 100 nm resolution (minimum pitch), which cannot be achieved by MPL, LC-MPL and MC-MPL. In addition, we have supplemented the resolution experiments with woodpile structure for MPL, MC-MPL and LMC-MPL (**Supplementary Fig. 8**). It can be seen LMC-MPL can obtain the highest resolution of (LR=200 nm), which cannot be reached by MC-MPL (LR=400 nm) and MPL (LR=250 nm).

31.The claimed minimum resolution is 100 nm, but there is no figure in the manuscript to support this. The result should be placed in the manuscript. In Fig. S7, what are the fabrication parameters (dose and writing speed) to realize the 100 nm resolution. From the SEM, the line is close to be discontinuous, is it caused by the low dose?

Response:

We appreciate the comments. We have placed the result of resolution into the revised manuscript (as shown in **Fig. 4c**), and added the fabrication parameters below **Fig. 4c** and **Supplementary Fig. 13**. We believe that the discontinuities in the lines are mainly due to the instability of lithography system. The voxels of LMC-MPL are very small ($a = 41$ nm and $b = 148$ nm, **Supplementary Fig. 7c**). The size of voxels changes with fluctuations in laser energy, especially near the threshold laser power. In addition, any system jitter can cause the relative position change between the voxel and the substrate, especially in the z-direction, which tends to underexpose the voxel, resulting in discontinuous lines from time to time. Thus, it has extremely high requirements for the stability of the lithography system, such as constant temperature and humidity, low noise, low electromagnetic interference, high vibration-resistant ground, high stability power supply system, and high stability system platform. Unfortunately, our self-constructed lithography system is currently in the early stages of scientific research, and it is difficult to fully meet the above

requirements. For instance, we have provided some jitter lines (**Fig. R9**) measured during a period when our system was very unstable that might be caused by electromagnetic interference, which is sufficient to prove the existence of system instability. Even so, this doesn't detract from the advancement of the stated LMC-MPL strategy.

Fig. R9 Jitter lines written by our system under unstable conditions.

32. In addition to MC-MPL, LC-MPL and LMC-MPL, a post processing shrinking method is also effective in reducing pitch and CD, e.g. Liu Y, Nature Commun., 2019, 10(1): 4340. As this method does not require modification of the resin composition nor multi-wavelength laser alignment, it is a compelling approach to achieve higher resolution structures and should be discussed.

Response:

We appreciate the comments. We were concerned about this strategy earlier. Post processing shrinking method is indeed an effective method in reducing the resolution, which does not require modification of the resin composition nor multi-wavelength laser alignment. However, this method is usually only applicable to 3D structures and unsupported structures, and it is difficult to improve the resolution of the pattern on the substrate because the photoresist on the substrate will be firmly anchored by the substrate and effective shrinkage in the XY direction cannot be realized (**Fig. R10**). Meanwhile, the shrinkage anisotropy is also difficult to overcome, give rise to distortion of the target structure.

We have added the discussion about the heat-shrinking method in the introduction of the revised manuscript (**highlight in blue, Page 2**). Moreover, we can achieve a higher-precision woodpile structure with CD= 41 nm and LR=200 nm (**Supplementary Fig. 7** and **Supplementary Fig. 8**), which is

better than that (CD= 100 nm and LR=350 nm, **Fig. R10**) obtained by Y Liu et al. using the heat-shrinking method.

Fig. R10 The ultimate resolution of woodpile structure obtained by Y Liu et al. using the heat-shrinking method. [Nature Commun, 2019, 10(1): 4340.]

33. In Fig. 2a, the manuscript mentions that the Quenching path 1 is the energy transfer from s1-DETC to TEMPO. This description is confusing and does not match the illustration in Fig. 2a.

Response:

We appreciate the comments. We are sorry for making this mistake. Indeed, the quenching path 1 and 2 in previous manuscript don't correspond to that in **Fig. 2a**. We have corrected the error in the revised manuscript.

34. There has been some other works on MC-MPL in which they utilized different quenchers. It is better to cite some of these papers to compare the performances of the quenchers in this paper with others. Is the quencher in this work better?

Response:

We appreciate the comments. As mentioned by reviewer, we have found several quenchers that had been reported for MC-MPL, which are collected in the following **Table. R1**.

Table. R1 Quenchers that have been reported for MC-MPL.

Abbreviation	Full name	CD/ nm	Reference
MEHQ	4-Methoxyphenol	118	Journal of Materials Chemistry. 21, 5650-

			5659 (2011)
DMAE-MA	2-(Dimethylamino)ethyl methacrylate	60	ACS Nano. 6, 2302-2311 (2012)
BHT	Butylated hydroxytoluene	-	International Journal of Molecular Sciences. 24, 1370 (2023)
TEMPO	2,2,6,6-tetramethyl-4-piperidyl-1-oxyl	122	Advanced Engineering Materials. 20, 1800320 (2018)
BTPOS	bis(2,2,6,6-tetramethyl-4-piperidyl-1-oxyl)sebacate	<100	Nature Photonics. 15, 932-938 (2021)

In contrast, by using nitroxide radicals (BTPOS) as radical quenchers, CDs below 100 nm also can be achieved (Nature Photonics, 15, 932-938, 2021), and the smallest CD up to 60 nm had been realized by using DMAE-MA (ACS Nano, 6, 2302-2311, 2012).

Apart from this, we have also selected several quenchers and added them to Pr1 in equimolar amounts to test their threshold arrays and the linewidth under the same test condition. As shown in **Fig. R11**, nitroxide radicals (BTPOS or TEMPO) have the highest threshold and the smallest linewidth among them. Therefore, TEMPO should be an excellent quencher for MC-MPL. We have cited these references into the introduction of (**highlight in blue, Page 2**) and added the data (**Supplementary Fig. 3** and **highlight in blue in Page 4**) into the revised manuscript

Fig. R11 (a) The threshold power (P_{th}) curves of photoresists with different quenchers (TEMPO, BTPOS, DMAE-MA, MEHQ) at different writing speed. (b) SEM images of the linewidth for these photoresists at the same processing

parameters (excitation beam power: 2.7 mW, writing speed: 500 $\mu\text{m s}^{-1}$). Scale bar: 1 μm .

35 The explanation of quenching path 2 mechanism is not clear. Besides, which path among the 3 quenching paths takes the most significant and efficient influence on the quenching process by using this DETC-TEMPO material?

Response:

We appreciate the comments. There is an annotation error about Quenching path 1 and Quenching path 2 in the previous manuscript, we guess the reviewer refers to static quenching here. First, whether the nonlinearity of the Stern-Volmer plots of fluorescence intensity and fluorescence lifetime, or the reduction of the ESR peak of TEMPO due to the presence of ground state DETC, all prove the existence of static quenching. And it had been reported that *"In static quenching a complex is formed between the fluorophore and the quencher, and this complex is nonfluorescent."* [Lakowicz, J. R. Principles of Fluorescence Spectroscopy; Plenum Press: New York, 1986]. Unfortunately, how the complexes are formed and the specific molecular structures of complex are not reported in this reference, since the complex is difficult to detect by existing means. In our work, the fluorescence quenching of DETC by TEMPO has been consistent with the above literature, but the formation mechanism of the complex between ground DETC and TEMPO is really difficult for us to investigate at present. As shown in **Supplementary Fig. 5b**, the introduction of TEMPO has almost no effect on the UV-vis absorption spectrum of DETC, indicating that TEMPO does not change the conjugation system of DETC. Therefore, this static quenching is likely to be from some weak interaction. We surmise that there two kinds of weak interaction between them: 1. van der Waals force, which is an ordinary force. For example, the motion of electrons in a TEMPO molecule generates a transient dipole moment, which transiently polarizes neighboring DETC molecules, and the polarized DETC in turn enhances the transient dipole moment of the TEMPO molecule. This force often needs to be manifested when two molecules are in very close proximity, which is consistent with the concentration dependence of TEMPO. 2. shielding effect

of TEMPO, TEMPO may act as a polarizing molecule, high concentration TEMPO may shield the DETC molecule, making the two DETC molecules inaccessible and inhibiting the aryl-ring stacking effect, which can accelerate the rapid rotation and vibration of a single DETC molecule, resulting in a high non-radiative decay rate and hence fluorescence burst.

Static quenching only works at the high concentration of TEMPO, so clearly it is not the most effective one. For quenching path 3, we have mentioned in the manuscript that even if the concentration of TEMPO increases to almost completely quench the fluorescence of DETC (S_1 -state DETC), initiation and polymerization in Pr2 can still occur through the second photo-excitation path ($S_n \rightarrow T_n$). This shows that the free radicals generated at T_n state cannot be quenched by quenching path 3 completely, and proves the quenching efficiency of quenching path 3 is not high. Additionally, dynamic quenching also affects the photo-inhibition process that interacts with the S_1 -state DETC (**Fig. 3f**). Therefore, we believe that dynamic quenching takes the most significant and efficient influence on the quenching process by using this DETC-TEMPO material.

36.Can LMC-MPL keep the same fine fabrication in 2.5D or 3D? If can, it will be better to show a 2.5D model in SEM to demonstrate the ability.

Response:

We appreciate the comments. LMC-MPL can also keep higher fine fabrication in 2.5D or 3D structure. For 3D fabrication, we have added the experimental results of a woodpile structures. We conducted two experiments. The first one is to compare the woodpile structures (the line period is fixed at 300 nm) of MPL, MC-MPL and LMC-MPL exactly under the same test conditions (fixed scan speed and laser power) in **Supplementary Fig. 7**. It shows LMC-MPL can get a finest photonic crystal (41 nm) compared with MC-MPL (78 nm) and MPL (260 nm). The second experiment is to continuously reduce the line period (d) between two lines for obtaining the minimum resolution of MPL, MC-MPL and LMC-MPL (**Supplementary Fig. 8**). It can be seen LMC-MPL can obtain the highest resolution of (LR=200 nm), which cannot be reached by MC-MPL (LR=400 nm) and MPL (LR=250 nm).

For 2.5D fabrication, we fabricated a 2.5D quadrangular pyramid structure with widths of 6.5 μm and 13 μm for MPL, MC-MPL and LMC-MPL respectively (**Supplementary Fig. 9**). The quadrangular pyramid, especially the smaller one, made by LMC-MPL has sharper edges and vertices compared with those made by MPL or MC-MPL, which can prove the excellent manufacturing capabilities of LMC-MPL. The relevant discussion also had added in the revised manuscript (**highlight in blue, Page8**).

Supplementary Fig. 9 Square pyramid structures of different sizes (6.5 μm wide and 13 μm wide) with excitation beam power and writing speed fixed as 5.35 mW and 5 mm s⁻¹. Scale bar: 1 μm .

37. Is it possible to fabricate free standing structures and what is the smallest pitch it can achieve?

Response:

We appreciate the comments. LMC-MPL is fully capable of manufacturing free standing structures. The CD (20 nm) and LR (smallest pitch, 80 nm) of the suspended lines for LMC-MPL were given in **Supplementary Fig. 14**. In addition to suspended lines, a woodpile structure can also be made by LMC-MPL, and the smallest pitch of the woodpile photonic crystal achieves 200 nm (**Supplementary Fig. 8**).

Supplementary Fig. 14 The (a) CD (20 nm) and (b) LR (80 nm) of suspended line for LMC-MPL with 1.18 mW excitation beam power, 9.65 mW inhibition beam power and 50 $\mu\text{m s}^{-1}$ writing speed. Scale bar: 200 nm.

38. What is the influence on the quenching process once the inhibition beam is turned on?

Response:

We appreciate the comments, this is an interesting question. We have analyzed the influence of the inhibition beam on the three quenching paths respectively.

Quenching path 1: Static quenching only occurs between the ground state TEMPO and the ground state DETC. The UV-vis spectra of TEMPO (**Supplementary Fig. 5b**) shows that the TEMPO has no absorption of the inhibition light (532 nm). And when the inhibition laser power is turned on (**Fig. 3f**), the inhibition light will mainly interact with the excited-state DETC, making it back to ground state. Therefore, if the amount of DETC in the ground state increases by turning on the inhibition beam, the static interaction may be strengthened, compared to only the excitation beam on.

Quenching path 2: dynamic quenching and STED- K_{S1} by the inhibition light are both the depletion process of S_1 -state DETC. Therefore, there is a synergistic and competitive relationship between dynamic quenching and inhibition beam. The synergy is reflected in the fact that LMC-MPL can produce higher-precision results than using light-confined or matter-confined alone (**Fig. 4b**). The competitive relationship can be reflected from the decrease of the slope of Pr2 in the decreasing part than that of Pr1 in **Fig. 4f**. It shows the presence of TEMPO reduces the efficiency of the photoinhibition.

Quenching path 3: active free radical scavenging by nitroxide radicals within TEMPO. The inhibition light will further reduce the amounts of active free radicals, so it may reduce the quenching path 3 process.

Overall, there is a synergistic and competitive relationship between inhibition light and the quenching process, but we think the quenching process dominates. As shown in **Fig. 4b**, from 139 nm (MPL) to 55 nm (MC-MPL), the average linewidth is narrowed by 60.4% via quenching process, while that is only 13.7% for LC-MPL (120 nm), indicating that quenching process seems to be more efficient than the inhibition beam. The same goes for the comparison of resolution (**Fig. 4c** and **Supplementary Fig. 13**).

Since it is difficult to provide definitive evidence for the above speculations, they are discussed here only and have not been added to the revised manuscript.

39. Is the lateral resolution (LR) mainly influenced and improved by the light-confinement process?

Response:

We appreciate the comments. As we mentioned above (Response #38), we believe that the matter-confining process plays a more important role than the light-confining process. In the resolution comparison experiment (**Fig. 4c** and **Supplementary Fig. 13**), 300 nm is almost the ultimate resolution of MPL. While only with the help of the inhibition beam, the resolution of LC-MPL has not been improved much (300 nm). But, the MC-MPL has a much better LR of 175 nm using matter-confined alone. The controlled experiment indicates that the matter-confining process performs more efficient than the light-confining process in improving the resolution.

40. In Fig. 4a LC-MPL, the second scale bar is not correct. Typo in line 250.

Response:

We appreciate the comments. We have corrected the error in the revised manuscript.

41. In terms of highest resolution patterns achieved using EBL, it is worth taking

a look at Yang J K W, J. Vac. Sci. Technol., 2009, 27(6): 2622-2627. in which they realize 9 nm pitch in HSQ. Again, it is the pitch that matters, and not CD. It is important to clarify that in the references 47, Saifullah et al. shows 4-nm feature size, but the minimum pitch is large. There is a difference between resolution and minimum feature size. A minimum feature size is not difficult to realize. However, packing them closely is what matters.

Response:

We appreciate the comments. We thank the reviewers for sharing the literature, which we read carefully. In the reference 47 [Nano Letters. 22, 7432-7440 (2022)], Saifullah et al. achieved 4 nm feature size and 25 nm pitch base on EBL. While in the reference provided by the reviewer [Sci. Technol., 2009, 27(6): 2622-2627], Joel K. Yang et al. reported 9 nm pitch, which indeed has a higher resolution. So, we also cited this reference as well in **Fig. 4f**.

We clearly understand the difference between feature size (or CD called resolution in our manuscript) and minimum pitch (or called resolution in our manuscript). In our manuscript, we compared the minimum resolution (minimum pitch) made by MPL, MC-MPL, LC-MPL and LMC-MPL (**Fig. 4c** and **Supplementary Fig. 13**) as well instead of just considering the CD. Among them, LMC-MPL can reach a 100 nm resolution (minimum pitch). Apart from the lines on the substrate, we also tested and compared the resolution (minimum pitch) of woodpile photonic crystal (**Supplementary Fig. 7** and **Supplementary Fig. 8**). All these results confirmed the superiority of LMC-MPL in improving resolution (minimum pitch) compared with the MPL, MC-MPL and LC-MPL. Although 100 nm resolution (minimum pitch) is good enough in MPL [Laser & Photonics Reviews. 16, 2100229 (2022)], it must be admitted that there is still a big gap compared to EBL, and needs further efforts.

42. Fig 4e would be clearer shown in log coordinates instead of the linear coordinates. And this figure is not complete, e.g. please see Wang S, Nano Futures, 2018, 2(2): 025006. In this work, they realized 7 nm width nano-webs by two-photon lithography.

Response:

We appreciate the comments. We have replaced the original linear coordinates with log coordinates in the revised manuscript (**Fig. 4f**). In addition, the reference mentioned by the reviewer [Nano Futures, 2018, 2(2): 025006.], reported a result of suspended lines (7 nm CD and 33 nm resolution). We have emphasized below **Fig. 4f (highlight in blue, Page 12)** that all the data listed in **Fig. 4f** are the results of lines writing on the substrate, excluding the suspended lines. Suspended lines can achieve higher precision than substrate lie due to the unavoidable shrinkage. Therefore, it is inappropriate to compare the suspended line with the substrate line.

43. There are some repeated figures, and it will be better to reorganize the figures in the manuscript for clarity.

Response:

We appreciate the comments. We have reorganized and formatted the whole figures in our manuscript.

We thank the reviewer again for the chance of revision/resubmission and also for the time and attention given to this manuscript.

Reviewer #3 (Remarks to the Author):

The paper describes the benefit of combining light-confined or matter-confined MPL to achieve higher resolution in resists. The synergistic effect of appropriate quenchers is described which gives some indication in which direction future research could go. The results were mainly achieved by screening quenchers, and TEMPO showed the best quenching effects. Furthermore, a two-step-STED process was identified, that enhances resolution further. It is good that in the conclusion the shortcomings of the new method are presented, that indicate that the method requires more sensitive resists that would allow higher throughput. Overall, the paper is a good overview about the combination of excitation and inhibition by the MPL method, which derives from STED, and therefore helps to understand the underlying effects. However, although the language is mostly very scientific, there is a certain sloppyness that would need to be improved. First, some terms are not well defined, or at least it is assumed that everybody knowing the state of the art understands them, e.g., critical dimension (CD) and line resolution (LR). Second, there are sometimes assumptions that are simply not true, or not precise: The resolutions presented are certainly achievements that go beyond the state of the art of MPL. However, when comparing with conventional lithographies, the authors compare pears and apples, i.e., their results with published results of DUV and EUV resolutions achieved in production. For me this is sloppyness, but it could also be interpreted as overselling results.

Response:

We appreciate the reviewer's positive comments and constructive suggestions. We have added the definition or explanation of some uncommon terms, including CD and LR, in the revised manuscript (**highlight in blue, Page1**). Among them, critical dimension (CD) represents the minimum achievable linewidth of an isolated line and lateral resolution (LR) means the minimum grating period between two separated lines [*Nature Photonics*. 15, 932-938 (2021)]. In addition, a schematic diagram was also given in the Supplementary Information (**Supplementary Fig. 12**) for allowing readers to understand more intuitively.

Supplementary Fig. 12 a schematic diagram of CD and LR.

Our work is focused on using new strategy to improve precision of MPL and reveal the mechanism. There is indeed a big gap between MPL and conventional lithography, which we will never deny. The comparison with conventional lithography is to illustrate the gap between MPL and conventional lithography. There may be some inappropriate expressions in the manuscript that may cause misunderstanding to the readers, and we have modified these expressions. The specific modifications are as follows. Specific modifications have been given below.

44.Abstract: What means cheaper and readily available in this sense? With current resist systems (and pulsed lasers), can we aim for high throughput comparable or even better than conventional lithography? Instead of cheap, use "low cost" and refer to what kind of cost you mean (production with 200 wafers/h or simply a research tool that can be used instead of EBL or another MPL? Since it is in the abstract, it seems an assumption that does not need proof here, but if you claim this in the introduction, a reference or at least the basis for a comparison is needed - and I think that this has to be specified in the introduction. And since the sensitivity will be lower, will there ever be a chance to be high throughput? Higher resolution also means - in a serial process - writing of more lines and thus longer writing time.

Response:

We appreciate the comments. We admit that it is inappropriate using “cheaper and readily available” here. We have changed the expression here using “low cost and more accessible”. As for “Mask-free multi-photon lithography (MPL) enables the fabrication of arbitrary nanostructures low cost and more accessible than conventional lithography”, our original intention is to emphasize two things: First, mask-free, which makes the manufacturing

process of MPL simpler than conventional lithography which needs mask for patterning. In addition, low cost of MPL can also be reflected in minimized material waste, and low-volume production. Second, fabrication of arbitrary nanostructures, it is obvious that for arbitrary nanostructures, especially 3D structure, MPL has greater advantages than traditional photolithography, making it more accessible. For these two points, we specified this and cited the reference [Advanced Functional Materials. 2214211 (2023)] in the introduction as well. So, we do not aim to compare throughput with conventional lithography.

We have admitted that the low sensitivity of the photoresist will affect the speed of writing, leading to a reduction of manufacturing efficiency. As for the impact of resolution on efficiency, this is a general and inevitable issue. These two problems can be solved from two aspects: First, developing high-sensitivity photoresist, such as developing highly sensitive photoinitiators [ACS Applied Polymer Materials. 5, 2956-2963 (2023)]. However, there are currently very few photoinitiators with photo-inhibition properties, so this is still a challenge. Second, developing multi-beam parallel laser direct writing system. Our colleagues are already developing 6-beam [Doi: 10.3788/CJL202249.2202009, in Chinese], 10-beam or even 1,000-beam lithography systems [Opt. Express. 31, 14174-14184 (2023)], which will greatly improve the writing efficiency of MPL.

45. Why only referencing LC-MPL 12, and not STED by Stefan Hell? Describe difference between critical dimension and line resolution, is the latter the half pitch? Or is it the distance between resist lines? The terms are not generally used and should be defined (best would be in a figure).

Response:

We appreciate the comments. STED proposed by Stefan Hell represents a stimulated emission depletion ($S_1 \rightarrow T_1$) by inhibition beam [Opt. Lett. 1994, 19, 780]. In our work, LMC-MPL contains STED, but not limited to traditional STED proposed by Stefan Hell. Through experiment and discussion, we find a two-step-STED mechanism. As shown in **Fig. 3i**, inhibition beam enables the transition of electron not only from S_1 to S_0 (STED- K_{S_1}), but also from S_n to S_1 (STED- K_{S_n}). Among them, STED- K_{S_1} corresponds to the traditional SETD

proposed by Stefan Hell, while STED-K_{Sn} is a new discovery. Therefore, we used LC-MPL to describe the effect of inhibition beam.

As requested by reviewer, we have added the definition of CD and LR in the revised manuscript (**highlight in blue, Page1**). Critical dimension (CD) represents the minimum achievable linewidth of an isolated line and lateral resolution (LR) means the minimum grating period between two separated lines [*Nature Photonics*. 15, 932-938 (2021)]. In addition, a schematic diagram was also given in the Supplementary Information (**Supplementary Fig. 12**) allowing readers to understand the definition more intuitively. So, LR means “an entire pitch” instead of half-pitch in the manuscript.

46.MPL and LC-MPL performance of the two photoresists were performed, respectively." Is there something "better" than a synergistic effect of subsequent LC-MPL and MC-MPL processes?

Response:

We appreciate the comments. What we would like to express here is that we first investigated the performance of two photoresists without inhibition light (MPL), and then we compared the performance of the two photoresists (Pr1 and Pr2) with inhibition light (LC-MPL). LMC-MPL means inhibition light and matter co-confined MPL. So, “LMC-MPL” and “the synergistic of LC-MPL and MC-MPL” refer to the same meaning. “Better” here means that the performance of LMC-MPL is better than using LC-MPL or MC-MPL alone. The expression here is not accurate enough, so we modified the statement in the revised manuscript (**highlight in blue, Page 3**).

47 Reducing the wavelength of excitation laser is an effective method. Yes, this is a general approach, why is it not used here? And right, we probably do not have commercial resists for 0.5x157nm, 0.5x193nm resists?

Response:

We appreciate the comments. First, femtosecond lasers with wavelength less than 500 nm are more expensive on the market. Second, although second harmonic fiber lasers are being used to get a shorter wavelength using a 780 nm laser [*ACS Applied Nano Materials*. 3, 11434-11441 (2020)], this method

has higher requirements for the optical system and will increase the cost as well. Third, the resins especially with photo-inhibition properties used in MPL, are usually polymerized by absorbing multi photons at a longer wavelength. When laser with shorter wavelength is used, linear absorption of single photon easily occurs, causing a reduction in precision and making photo-inhibition impossible as well in MPL. The excitation laser we used here has a wavelength of 525 nm, can not only avoid single photon absorption to some extent, but also ensure the nonlinear interaction between the excitation light and the photoresist. Lasers with a wavelength of 525 nm are also relatively common and cheap on the market currently.

Commercial photoresists (157nm and 193nm resists) are generally positive tone and use photoacid (PAG) as initiator. As far as we know, these commercial photoresist does not have a photo-inhibition property, and it also cannot use quenchers for material confinement. We have also purchased some commercial resins (SU8) that are available in China. In terms of commercial 157nm and 193nm resists, it is almost unavailable in China. As shown in **Fig. R12**, SU8 exhibited relative low precision and had no light inhibition capacity.

Fig. R12 The achieved CD of SU-8 by MPL at various laser power (left). Light confining capability tests of SU-8 (right), corresponding to **Fig. 3b** in the manuscript.

48. "inevitable power fluctuation and focus drift in the system". This means the system used? Well, this could be improved and is therefore not a fundamental problem?

Response:

We appreciate the comments. Yes, this refers to power fluctuation and focus drift of the lithography system we used. Here, we cite the work by Minfei

He et al [PhotoniX. 3, 25 (2022)]. In her paper, although 140 nm resolution was reached, the line-edge-roughness shows very poor (**Fig. R13**). And, they proposed that “we attributed linewidth instability to insufficient mechanical strength, laser power fluctuations, and optical system drift”. So, this claim here quotes the views from the original paper.

[REDACTED]

Fig. R13. The 140 nm resolution results obtained by Minfei He et al. using LC-MPL. [PhotoniX. 3, 25 (2022)].

In our LMC-MPL system, we have really found that power fluctuation and focus drift of the MPL system does have great impact on the writing results. Since MPL is based on the nonlinear effect of a femtosecond laser, extremely small power fluctuations of the laser will be exponentially amplified in the lithography results. As for focus drift, because the voxel size of femtosecond laser is about a few hundred nanometers (**Supplementary Fig. 6a**), the small drift of the focal plane will have a great impact on the accuracy and quality of the photolithography lines on the substrate (**Fig. R14**). In addition, the focal plane will also be affected by environmental vibrations. Thus, it has extremely high requirements for the stability of the lithography system, such as constant temperature and humidity, low noise, low electromagnetic interference, high vibration-resistant ground, high stability power supply system, and high stability system platform. Unfortunately, our self-constructed lithography system is currently in the early stages of scientific research, and it is difficult to fully meet the above requirements.

Fig. R14 Lines at different focal plane positions (step: 30 nm, along Z direction) under the same fabrication parameters (power and writing speed).

In short, we believe that “power fluctuation and focus drift” is an important factor that cannot be ignored currently, but we believe that with the deepening of research and the promotion of applications, this problem will be gradually improved in the future.

49.what is a decent LR? 140 nm is great, depending on the point of view. Better: A LR below 100 nm or comparable to conventional DUV lithography.

Response:

We appreciate the comments. The expression "decent" here is inappropriate. What we want to express is to get better resolution than 140 nm. So, we replaced “decent” by "better" in the revised manuscript. 140 nm is a relatively good resolution in MPL, and the 100 nm achieved by us is almost the best resolution achieved by MPL (**Fig. 4f**) so far. Thus, we have made some progress in the MPL field. Of course, compared with DUV, there is still a long way to go. And we believe that the pursuit of high resolution will never stop.

50.Definition of the diffusion-based proximity effect! Typical resist manufacturers count on the effect of diffusion, otherwise crosslinking and chemically amplified resists would not be possible!

Response:

We appreciate the comments. The definition of “proximity” has been given in the introduction of the revised manuscript (**highlight in blue, Page1**). Simultaneous writing of structures in close spatial proximity generates fabrication artefacts, collectively referred to as “proximity effects”, which strongly limit the accessible structure resolution [*Additive Manufacturing*. **49**, **102491 (2022)**]. And the proximity effect is generally caused by the diffusion and accumulation of free radicals for the photoresist using radical photoinitiators. In MPL, the voxel size is usually hundreds of nanometers in size, and chemical molecule have a faster diffusion in the liquid photoresist. Therefore, the diffusion-base proximity effect is one of the very important factors affecting the resolution in this case. If diffusion is not confined, the

resolution can only reach the previously mentioned resolution of 140 nm at most, even with photo-inhibition [PhotoniX. 3, 25 (2022)].

Chemically amplified photoresists do rely on the diffusion of photoacid to cross-link or deprotect. However, it is worth noting that alkaline substances are also added to above photoresists to constrain the diffusion of photoacid and improve photolithography accuracy [ACS Omega, 2023, 8, 30, 26739-26748], which is similar to the strategy in our work.

51.a continuous diffusion of quencher molecules from other, non-depleted areas.

Response:

We appreciate the comments. In the previous manuscript, the statement about quencher diffusion may not be rigorous enough, and we have completed the revision as suggested by the reviewer (**highlight in blue, Page2**). Explanation in detail: In the exposed area, the photoinitiators will generate large number of free radicals, so the quenchers in the exposed area will be consumed rapidly due to their interaction with the free radicals. This results in a quencher concentration difference between the exposed area and the non-exposed area, thus leading to a continuous diffusion of quencher molecules from other non-depleted areas. [ACS Nano. 6, 2302-2311 (2012)] and [AIP Advances. 5, 127215 (2015)].

52.I guess that LMC-MPL also has limitations in terms of CD or LR, simply different ones?

Response:

We appreciate the comments, this is a good question. Undoubtedly, LMC-MPL also has limitations. As we mentioned in the last paragraph of the **Conclusion** section: “*Nevertheless, the improved lithography precision by LMC-MPL is at the expense of sensitivity of photoresist to some extent. This will not only increase the energy consumption of the manufacturing, but also is harmful to high-speed and large-area lithography. Therefore, developing photoresists with both high sensitivity and precision is one of the future directions in MPL.*” In other words, the exposure efficiency of the present

technique is inadequate, much less comparable to conventional DUV/EUV techniques.

In terms of CD or LR, the improvement through LMC-MPL is clearly limited. Both CD and LR are difficult to further increase due to an upper limit on the efficiency of matter / light confinement. The difference is that LR is more susceptible to the proximity effect because the two lines are too close together, whereas CD is usually an isolated line. As we mentioned in **Response #50**, simultaneous writing of structures in close spatial proximity generates fabrication artefacts, collectively referred to as “proximity effects”, which strongly limit the accessible structure resolution [*Additive Manufacturing*. 49, 102491 (2022)]. And the proximity effect is generally caused by the diffusion and accumulation of free radicals for the photoresist using radical photoinitiators. As a result, LR tends to be greater than twice CD. For example, in our work, CD is 30 nm while LR is only 100 nm.

53."Achieving stable sub-50 nm linewidth and 100 nm resolution simultaneously." What does this mean? Sub-50 nm CD as isolated lines and 100 nm resolution (half pitch) of dense lines? And why do you mention at the same time, does this mean in the same resist, with the same process parameters?

Response:

We appreciate the comments. We have given the definition of CD and LR in the revised manuscript (**highlight in blue, Page1**). Among them, CD represents the minimum achievable linewidth of an isolated line and LR means the minimum grating period between two separated lines [*Nature Photonics*. 15, 932-938 (2021)]. We also gave a schematic diagram in the Supplementary Information (**Supplementary Fig. 12**).

So, it means that the linewidth is less than 50 nm in a 100 nm resolution (one pitch). When using MPL, it is difficult to achieve a linewidth exactly equal to half a pitch, which is somewhat different from traditional mask-based semiconductor lithography. There is a difference in the minimum linewidth that can be achieved when writing separate lines and writing tight lines even with the same exposure dose, due to the influence of the proximity effect. Generally,

the linewidth in tight lines will be larger than the separate line. As shown in the following (**Fig. R15**), the linewidth alone can reach 30 nm, while in experiments with minimum resolution (100 nm), the linewidth is 43 nm greater than 30 nm.

Fig. R15 Separate lines (left) and tight lines (right) writing with same exposure dose.

54. What does "To demonstrate LMC-MPL has a better synergistic effect than using light-confined or matter-confined alone, MPL and LC-MPL performance of the two photoresists were performed, respectively." Is there something "better" than a synergistic effect of subsequent LC-MPL and MC-MPL processes?

Response:

We appreciate the comments. What we would like to express here is that we first investigated the performance of two photoresists without inhibition light (MPL), and then we compared the performance of the two photoresists (Pr1 and Pr2) with inhibition light (LC-MPL). LMC-MPL means inhibition light and matter co-confined MPL. So, "LMC-MPL" and "the synergistic of LC-MPL and MC-MPL" refer to the same meaning. "Better" here means that the performance of LMC-MPL is better than using LC-MPL or MC-MPL alone. The expression here is not accurate enough, so we modified the statement in the revised manuscript (**highlight in blue, Page 3**).

55. The comparison with EUV lithography seems somehow odd to me. Is this a comparison with current production CD and linewidths or such obtained in laboratory? then rather write that this is comparable with CDs and linewidths currently employed in production using DUV and EUV litho. Are there references for DUV and EUV proving the claim? Also for minimum resolution.

Response:

We appreciate the comments. We admit that it is inappropriate to use 30 nm-CD and 100 nm-LR in MPL to compare with EUV lithography which even achieve a limit CD of 13 nm (TWINSCAN NXE:3600D, ASML, **Fig. R16**).

[REDACTED]

Fig. R16 Parameters of EUV lithography system (TWINSCAN NXE:3600D, ASML).

In case of DUV lithography, the latest 193 nm DUV lithography (NXT:2100i, ASML, **Fig. R17**) achieve a 38 nm CD (96 nm LR). Therefore, if only considering a single exposure, the 30 nm CD and 100 nm LR we obtained are comparable to DUV lithography. Of course, multiple exposure has been used in actual manufacturing to further improve the feature size and resolution. For example, the LR of Intel 14nm and 10nm are 52nm and 36nm respectively. Indeed, we cannot achieve this precision currently using MPL.

[REDACTED]

Fig. R17 Parameters of DUV lithography system (NXT:2100i, ASML).

In addition, there are relatively few reports on DUV lithography in the literature recent years and its accuracy is not very high, such as 160 nm CD and 400 nm LR [*Appl. Phys. Lett.* 118, 141103 (2021)], 150 nm CD and 300 nm LR [*Optical and Quantum Electronics.* 44, 521-526 (2012)], 580 nm LR [*ACS Appl. Polym. Mater.* 2022, 4, 4508–4519]. While reports on EUV lithography

can indeed achieve higher precision, such as 22 nm half pitch [ACS Mater. Au 2022, 2, 343–355]. Therefore, compared with the DUV and EUV data obtained in laboratory, we can draw similar conclusions to the above.

The comparison with conventional lithography is rather to show the gap between the two. So, we used "shortened the gap" to replace "comparable" for more appropriate expression. As the reviewer mentioned in **Response #66**, we find that, in the first version of the manuscript, we mentioned EBL and DUV in Introduction, but listed EBL and EUV in **Fig. 4f**, and used DUV and EUV in Conclusion. It's confusing and doesn't correspond to each other. We are very sorry our carelessness caused the error. Based on the above analysis, it can be found that there are few recent references about DUV lithography, and the accuracy obtained by DUV in the literature is not high. Therefore, we deleted all the discussion of DUV in the revised manuscript and only discussed EUV and EBL lithography.

56. Generally use μm , if possible, not um .

Response:

We appreciate the comments. We have corrected this error in the revised manuscript.

57. "is not conductive"? What does this mean? And can you present a factor of sensitivity decrease?

Response:

We appreciate the comments. "be not conductive to" is an uncommon expression, we have replaced it with "be harmful to" in the revised manuscript (**highlight in blue, Page15**). What we want to express here is the reduction of sensitivity of photoresist is harmful to high-speed and large-area lithography. According to **Fig. 1f**, the increasing of the threshold means Pr2 requires a larger exposure dose at the same scan speed. Correspondingly, with the same exposure dose, Pr2 needs a lower scan speed, which will directly affect the efficiency of large-area lithography.

The decrease in Pr2 sensitivity is attributed to the quenching effect of the quencher on the initiator molecules. Other factors that affects sensitivity include

the following aspects: First, photoresist composition including initiators [Chem. Eur. J. 2022, 28, e202104191] and monomers [Additive Manufacturing 51 (2022) 102658]. The multi-photon absorption cross-section and the quantum yield of free radicals will directly affect the efficiency of photoinitiation and thus affect the photosensitivity. Generally, the smaller functionality and lower activity of functional groups of monomers, the worse the photosensitivity of the photoresist. Second, refractive index of photoresist. The better the refractive index match between the photoresist and the objective lens, the smaller the aberration and the better the photosensitivity will be gained. We choose a mixture of TCDA and EBPFDA-OPPEA as the monomer of our photoresist. The refractive index of TCDA and EBPFDA-OPPEA is 1.503 and 1.608 respectively, through adjusting the ratio of the two monomers, we can get a photoresist with a refractive index of 1.518, which exactly matches the refractive index of our objective lens. We explain the issue of monomer selection and refractive index matching of photoresists in detail in our previous work [ACS Applied Materials & Interfaces. 14, 31332-31342 (2022)].

58. What means (0-0-26.65 mW)?

Response:

We appreciate the comments. We are sorry for making this mistake and we have corrected it in the revised manuscript. It should be 0 - 26.65 mW.

59. CRedit?

Response:

We appreciate the comments. "CRedit author statement" represents "Author contributions". This is an error because we did not describe it according to the Nature communication format, we have modified this in the revised manuscript (**highlight in blue, Page 22**).

60. Do the quenchers interact chemically with the initiators? What would be the case if initiators with higher sensitivity would be used?

Response:

We appreciate the comments. There are three quenching paths for quenchers and initiator, so we next discuss whether they fall under the category of chemical reactions separately.

Quenching path 1 is static quenching. Whether the nonlinearity of the Stern-Volmer plots of fluorescence intensity and fluorescence lifetime, or the reduction of the ESR peak of TEMPO due to the presence of ground state DETC, all prove the existence of static quenching. And “In static quenching a complex is formed between the fluorophore and the quencher, and this complex is nonfluorescent.” [Lakowicz, J. R. Principles of Fluorescence Spectroscopy; Plenum Press: New York, 1986]. Unfortunately, how the complexes are formed and the specific molecular structures of complex are not reported in this reference, since the complex is difficult to detect by existing means. In our work, the fluorescence quenching of DETC by TEMPO has been consistent with the above literature, but the formation mechanism of the complex between ground DETC and TEMPO is really difficult for us to investigate at present. As shown in **Supplementary Fig. 5b**, the introduction of TEMPO has almost no effect on the UV-vis absorption spectrum of DETC, indicating that TEMPO does not change the conjugation system of DETC. Therefore, this static quenching is likely to be from some weak interaction. We surmise that there two kinds of weak interaction between them: 1. van der Waals force, which is an ordinary force. For example, the motion of electrons in a TEMPO molecule generates a transient dipole moment, which transiently polarizes neighboring DETC molecules, and the polarized DETC in turn enhances the transient dipole moment of the TEMPO molecule. This force often needs to be manifested when two molecules are in very close proximity, which is consistent with the concentration dependence of TEMPO. 2. shielding effect of TEMPO, TEMPO may act as a polarizing molecule, high concentration TEMPO may shield the DETC molecule, making the two DETC molecules inaccessible and inhibiting the aryl-ring stacking effect, which can accelerate the rapid rotation and vibration of a single DETC molecule, resulting in a high non-radiative decay rate and hence fluorescence burst. Clearly, this process is not a chemical reaction.

Quenching path 2 is dynamic quenching, which is an energy transfer occurring between TEMPO and S1-state DETC. So, no chemical interaction is involved.

Quenching path 3 is active free radical (from DETC) scavenging by TEMPO. A valence bonding interaction is formed between TEMPO and the active free radicals, which can be considered a chemical process [*Fuel Processing Technology*. 171, 350-360 (2018)]. However, the active free radical of the initiator is not the original form of the initiator, but a free radical generated by the initiator. Essentially, the quencher does not interact chemically with original form of DETC.

If initiators with higher sensitivity are used, the photosensitivity of the photoresist will increase theoretically, which is expected. But there may be some problems: First, whether the photoinhibition performance of the initiator exist ? Studies on developing new photoinitiators have never stopped, but few of them have photoinhibition properties [*Chemistry - A European Journal*. 28, e202104191 (2022)]. As far as we know, there are only two relatively mature initiators with photoinhibition properties in MPL, that are DETC and ITX (Isopropyl thioxanthone) [*Laser & Photonics Reviews*. 16, 2100229 (2022)]. So, it is still a challenge to develop highly sensitive and photo-inhibited initiators. Second, it is still questionable whether the quencher (TEMPO) still inhibit polymerization through the same three quenching pathways for the new initiators. New quenchers may need to be developed for new initiators to obtain matter-confinement. This is another new topic and challenge. However, we believe that this is a direction (developing new initiators with higher sensitivity and photo-inhibition property) worthy of study and discussion, and we will continue to study and explore it in future.

61. What if you write two times over the same resist? Larger areas composed of adjacent narrow lines. Would you then switch off the inhibition beam to merge lines?

Response:

We appreciate the comments. If writing twice at the same location on the same photoresist, the photoresist will form wider lines than a single exposure,

due to dose accumulation and diffusion (keeping same dose for each exposure). If the same photoresist is written twice in different locations (similar to the double exposure in semiconductors), it is theoretically possible to achieve higher resolution because the misalignment time can overcome the proximity effect. But this double exposure requires the focal spot of the two exposures to be highly aligned in 3D space. Our system does not have a self-alignment function currently, so we cannot provide the results through double exposure conditions.

As to whether or not to turn off the inhibition beam for larger areas composed of adjacent narrow lines, it depends. If the linewidth of the line (close to 30 nm) or the resolution (close to 100 nm) is very small, the inhibition beam must be turned on. Because with the inhibition beam turned off, this accuracy cannot be achieved. And if this large area of lines does not require high accuracy and resolution, of course the inhibition beam can be switched off to obtain higher efficiency. In practice, we can use software to control turning on and turning off of the inhibition beam at any time.

62. Fig. 2: b and g Gauss? -> Gauss? Better say "Magnetic field strength"

Response:

We appreciate the comments. We have corrected the abscissa titles and units in all ESR spectra.

63. Figure 3a) barely to see details in this image resolution. e) Point C, does it mean Point C?

Response:

We appreciate the comments. We apologize for the fact that after converting from word to pdf, the clarity of the pictures is greatly reduced, making it difficult for the reader to see them clearly. We've replaced the original image with a clearer one in the revised manuscript. Meanwhile, the error in **Fig. 3g** also have been corrected, it should be "Point C".

64. The overall slide seems too busy to me, it contains SEM micrographs, evaluations, and schematics. Why not taking a and b together, then c, d and then e to g and then h and i?

Response:

We appreciate the comments. We have rearranged the ordering of the images in **Fig. 3**, making them horizontally ordered and clearly to see.

65. Showing the 3D Nezza figures is nice (the resolution in the presented version for review is too low to see major difference), but there is one shortcoming: The "writing element" is a voxel, that typically is an extended ellipsoid that is longer in z than in xy. Can the authors provide the real size of the voxel?

Response:

We appreciate the comments. First, we've replaced the original image with a clearer one (**Fig. 1e** and **Fig. 3c-d**). Second, as shown in **Supplementary Fig. 7** and **Supplementary Fig. 8**, we further supplemented the experiments of 3D woodpile structure for MPL, MC-MPL and LMC-MPL. By taking SEM photos of tilted photonic crystals (xy plane is tilted 45°), we can get the real voxel size of MPL ($a=260$ nm, $b=806$ nm), MC-MPL ($a=78$ nm, $b=250$ nm) and LMC-MPL ($a=41$ nm, $b=148$ nm) with the same exposure dose (**Supplementary Fig. 7**).

Supplementary Fig. 7 (a) Schematic of a 3D woodpile structure with lateral period d and axial period $\sqrt{2}d$. (b) Schematic diagram of voxels, where a represents the size of the voxel along the y direction, b represents the size of the voxel along the z direction). (c) SEM morphology of a woodpile structure ($d=300$ nm) made by MPL, MC-MPL and LMC-MPL with 1.12 mW excitation beam power, 9.65 mW inhibition beam power and $50 \mu\text{m s}^{-1}$ writing speed. The images above (A, B and C) are taken perpendicular to the xy plane (scale bar: 300 nm), and images below (D, E and F) are taken with the xy plane tilted 45° (scale bar: $1 \mu\text{m}$). At a period of $a=300$ nm, the lines of woodpile structure obtained by MPL cannot be separated, so the size of the voxel cannot be measured. Therefore, while maintaining the same exposure dose (P_{ex} and scan speed), the period of the MPL is expanded to $a=700$ nm (inset of D image), thereby obtaining the voxel size in the MPL mode.

Supplementary Fig. 8 SEM morphology of woodpile structures with different period (d) made by MPL, MC-MPL and LMC-MPL (1.10 mW excitation beam power, 9.65 mW inhibition beam power and $50 \mu\text{m s}^{-1}$ writing speed). Scale bar: 300 nm.

66.Fig. 4e: Is LMC-MPL then better than or comparable to EUV, as mentioned

in the summary/conclusion 383 "By using LMC-MPL, we improve the critical dimension and lateral resolution to 30 nm and 100 nm, respectively, which are comparable to DUV and EUV lithography."? OK, with some tolerance, but the actual resolution of EUV achieved in labs is higher (7 nm). I do not see this. Where is DUV?

Response:

We appreciate the comments. We admit that it is inappropriate to use 30 nm-CD and 100 nm-LR to compare with the accuracy of EUV lithography and DUV lithography. So, we used "shortened the gap" to replace "comparable" for more accurate expression.

"Actual resolution of EUV achieved in labs is higher (7 nm)", the 7nm EUV currently mentioned mainly refers to TSMC 7 nm and Samsung 7 nm. In fact, the Minimum Metal Pitch (LR) that "TSMC 7nm" and "Samsung 7nm" can achieve are 40 nm and 36 nm respectively instead of real 7 nm. The EUV lithography result given in the **Fig. 4f** is 22 nm CD and 40 nm LR, which is comparable to "Samsung EUV 7nm". Furthermore, we also cite this reference again here.

In case of DUV lithography, the latest 193 nm DUV lithography (NXT:2100i, ASML, **Fig. R17**) achieve a 38 nm CD (96 nm LR). Therefore, if only considering a single exposure, the 30 nm CD and 100 nm LR we obtained are comparable to DUV lithography. Of course, multiple exposure has been used in actual manufacturing to further improve accuracy and resolution of DUV. For example, the LR of Intel 14nm and Intel 10 nm are 52 nm and 36nm respectively. We cannot achieve this precision using MPL currently. In addition, there are relatively few reports on DUV lithography in the literature recent years and its accuracy is not very high, such as 160 nm CD and 400 nm LR [*Appl. Phys. Lett.* 118, 141103 (2021)], 150 nm CD and 300 nm LR [*Optical and Quantum Electronics.* 44, 521-526 (2012)], 580 nm LR [*ACS Appl. Polym. Mater.* 2022, 4, 4508-4519].

As the reviewer mentioned, we find that, in the first version of the manuscript, we mentioned EBL and DUV in Introduction, but listed EBL and EUV in **Fig. 4f**, and used DUV and EUV in Conclusion. It's confusing and doesn't correspond to each other. We are very sorry our carelessness caused

the error. Based on the above analysis, it can be found that there are few recent references about DUV lithography, and the accuracy obtained by DUV in the literature is not high. Therefore, we deleted all the discussion of DUV in the revised manuscript and only discussed EUV and EBL lithography.

67.References: Are the first relevant publications [1,2] really from 2018 and 2021? There must be earlier references on MPL.

Response:

We appreciate the comments. There are indeed earlier references on MPL than reference [1,2] in our manuscript, we have re-cited the first publication about two-photon lithography by Maruo et al. in 1997 [Opt. Lett. 22, 132-134 (1997)].

We thank the reviewer again for the chance of revision/resubmission and also for the time and attention given to this manuscript.

REVIEWER COMMENTS

Reviewer #1 (Remarks to the Author):

Comment 1

The authors' reply to the reviewer's original comment #1 is answered in a piecewise fashion. The authors replies are highlighted in blue.

We appreciate the comments. The focus of our work is mainly on the proposal of a new strategy (LMC-MPL) to realize a high-precision and high-resolution lithography. And it is worth emphasizing that the acquisition and transfer of high precision patterns on silicon substrates is the first step for producing diverse micro devices, including semiconductor manufacturing. The experiment of on-silicon gratings certified the possibility of applying our strategy in above fields. That's why we put this part into the previous manuscript. Of course, that's not the point of this work. Therefore, as request by the reviewer, we have put this part in the Supplementary Information (Supplementary Fig. 19).

OK

In the manuscript, we proposed that the photochemistry of DETC may generate radicals through two paths ($S_1 \rightarrow T_1$ and $S_n \rightarrow T_n$) according to the experiments, and the DFT calculations certified the probability of this energy level transition. In fact, we are not the first to make this point. Martin Wegener et al once proposed that "when using DETC without a co-initiator, polymerization is initiated via two competing pathways, one of which obeys an $N = 3$ scaling and the other follows an $N = 4$ scaling. While the first one can be depleted (as it passes through S_1 and T_1), the other one cannot be interrupted, as it passes neither the relaxed S_1 state nor the T_1 state (Fig. R1)." [Advanced Optical Materials. 3, 221-232 (2015)]. This is exactly consistent with our experiments and mechanistic explanation. But, Martin Wegener et al believe that the two polymerization paths of DETC are achieved by three photon absorption and four photon absorption because they observed that a different nonlinearity absorption exponent ($N=3$ and $N=4$) versus repetition rate. Martin Wegener et al also got a result of $N>3$ for PETA+DETC in a more recent paper (Fig. R2) [Advanced Optical Materials. 7, 1901040 (2019)], but this time they attributed $N=4$ to the absorption of PETA monomer. Therefore, based on the above analysis and our own results (Fig. 1g), we are more inclined to think that all the absorption of photon for DETC are achieved by a three-photon absorption process. And subsequently it has two paths to go. One is it may transition from S_n to T_n state (ISC, $S_n \rightarrow T_n$) and generate free radicals at T_n state, and the other is it may return to S_1 state through vibrational relaxation (VR), then transition to T_1 state (ISC, $S_1 \rightarrow T_1$), and finally generate free radicals at T_1 state. We also cited these two references in the revised manuscript and added a brief discussion (highlight in blue, Page10).

The reviewer acknowledges the expanded discussion and the authors' cautious statement "we think there is only a three-photon absorption".

As for the photochemical experiments of DETC, we have further tested ESR spectra of TEMPO and TEMPO+DETC before and after UV irradiation (placed in the dark for 0 min, 20 min or 60 min) in Fig. 2c. Additionally, we also supplemented the fluorescence decays spectra of the photoresists (DETC) with

different concentration (0-40 mM) of TEMPO in Fig. 2f and the Stern-Volmer plots of fluorescence lifetime of DETC in Fig. 2g. The linear change of the lifetime indicates that dynamic quenching dominates at lower concentration of TEMPO (Fig. 2g). In the case of higher concentrations of TEMPO, the two processes (static and dynamic quenching) may be competitive, which results in a nonlinear relationship of lifetime and fluorescence. The combination of static and dynamic quenching almost completely quenched the fluorescence of Pr2 (Supplementary Fig. 5a and Fig. 2d). These further discussions about the photochemical process have been discussed in the revised manuscript (highlight in blue, Page7). The reviewer acknowledges these changes. These changes are discussed in comments 17ff.

DETC (7-diethylamino-3-thenoylcoumarin) was chosen as the photoinitiator in our photoresist because the initiator must not only have photoinhibition properties but also be wavelength-matched to our lasers. Up to now, there are only few relatively mature initiators with photoinhibition properties in MPL, such as DETC and ITX (Isopropyl thioxanthone) [Laser & Photonics Reviews. 16, 2100229 (2022)]. So, we don't have much choice for initiators with photoinhibition property. As for why we didn't choose ITX as another research object, because the photophysical and photochemical mechanism of ITX and DETC are not exactly same, such as their different performance in the pump-probe experiments (Fig. R3) [Opt. Lett. 36, 3188-3190 (2011)]. Therefore, the study on the interaction between DETC and TEMPO in this paper has its own uniqueness and the research on other photoinitiators is of little significance to the discussion and research based on DETC in this paper. That is why our entire work focuses on DETC and does not discuss other photoinitiators. Of course, we believe that using quenchers as a tool to study the photophysical and photochemical mechanisms of other photoinitiators, such as ITX, would be another feasible and interesting work.

In the reviewer's opinion, additional experiments using ITX or another depletable photoinitiator are not required. The reviewer's original comment aimed at underlining the complexity of DETC's photochemistry and the urgent need for a dedicated DETC review paper summarizing and discussing DETC's photochemistry in detail.

The fluorescence and ESR experiments were indeed performed ex-situ which has been emphasized in the revised manuscript (highlight in blue, Page7) and Methods (Photo-Physical characterization) section. OK, this experimental detail is of prime importance.

In fact, we had initially hoped to conduct an in situ test as well, but were not successful. We have not found commercial instruments for us to perform the in-situ fluorescence and ESR experiments, since such experiments are extremely rare and demanding. We have also tried our best to build our own instruments for in-situ fluorescence testing within the recent three months, but we failed. We found that not only did it take a long time, but it was so expensive that we couldn't afford it.

The reviewer appreciates the effort.

As for in-situ ESR test, it is theoretically not feasible because the laser beam will be focused into a point inside the photoresist and the radical distribution generated at this point cannot be tested by the current ESR instrument. In addition, although the ex-situ fluorescence and ESR tests cannot take the influence of oxygen (diffusion lengths and timescales) into account, this does not negate our use of ex-situ tests to explore the quenching mechanism of TEMPO. It can only be said that our ex-situ test cannot take into all the factors during the in-situ lithography, and it can indeed partially reflect the influence of

TEMPO on the fluorescence and ESR signal of DETC. And these effects must be taken into account in the analysis of the in-situ MPL.

The reviewer acknowledges that the authors disclosed that the experiments were conducted ex-situ in their revised manuscript.

Comment 2

The authors demonstrate that woodpile photonic crystals with rod spacings below 200 nm can be fabricated using LMC-MPL. Unfortunately, they keep the laser power constant when comparing LMC-MPL with MC-MPL and MPL. This is an unfair comparison, and this has been addressed in the original review comment #9. The authors should fabricate woodpiles using the threshold laser-power for each photoresist. The reviewer acknowledges that this requires effort, but the results are vital for the conclusions drawn in the manuscript.

Comment 3

The authors have expanded their methods section and included more detailed information about the setup. In their response to the reviewer's comments, they specify an objective lens numerical aperture of 1.4, whereas in the main text they specify an objective lens numerical aperture of 1.45. The authors should clarify which one is correct. Furthermore, the authors should be more precise and indicate the *full* model name of the objective lens in the main text.

In their methods section on the photo-physical characterization, the authors write "interference factors". To me, it is unclear what interference factors the authors are referring to. Please specify or remove.

Comment 4

In their revised manuscript, the authors have mentioned that the mechanistic TEMPO investigation was performed in-situ by writing "All the above conclusions are based on ex-situ tests, while MPL (in-situ) will be limited by space and time (such as quencher diffusion) as well." I don't understand the second part of the sentence, starting with "while MPL". The authors *must state clearly and concisely* what they mean.

I fully agree with the authors, that not all experiments can be conducted in-situ with state-of-the-art equipment. Nevertheless, it is important to make the reader aware that ex-situ experiments have a limited validity.

Comment 5

As requested, the authors have specified that the lithography experiments were conducted under ambient atmosphere in their revised methods section. The authors should likewise specify in the photo-physical-characterization section and EPR section that these experiments were also conducted under ambient atmosphere, i.e., normal air.

Comment 6

OK

Comment 7

References for two claims made in the original paper were requested. The reference for the first claim is OK.

Concerning the second claim, “It is worth noting that the quencher will be consumed rapidly in the exposed area, leading to a continuous diffusion of the quencher, which will further suppress the proximity effect.”, the authors now provide two references. In the first reference, “ACS Nano. 6, 2302-2311 (2012)”, 2-(dimethylamino)ethyl methacrylate is used as quencher. In contrast, the authors of the present paper refer to a quencher of the hindered-amine light-stabilizer group. It is highly questionable if the two quenchers behave comparably.

The second reference, “AIP Advances. 5, 127215 (2015)”, assumes that the quencher is consumed rapidly. However, no proof thereof is provided.

It is known that hindered-amine light stabilizers, like TEMPO or BTPOS, can recover within the Denisov-cycle. As of now, it is unclear if this is also the case in multi-photon 3D printing. Hence, the authors should rephrase their statement and clarify that the quencher consumption is an assumption the authors tacitly make.

Comment 8

In the original statement, “More importantly, it seems impossible to realize a decent LR since the proximity effect is still there.”, the authors have replaced the word “decent” by “better”. “Better” is a comparative word. It is not entirely clear to me, with what the authors are comparing the LR. I assume that the authors want to say that the LR in Minfei He et al.’s work is still limited by the proximity effect. Please clarify.

Comment 9

OK

Comment 10

The reviewer requested to perform linewidth measurements for the six different photoresist at their respective threshold laser power instead of keeping the laser power constant for all six photoresists. The authors now have added results of this experiment and show corresponding data in the Supplementary Fig. 2. The authors should add the laser powers that they have used for the respective lines. This is crucial since, unfortunately, the laser threshold laser powers at a scan speed of 50 $\mu\text{m/s}$ cannot be extracted from Supplementary Figure 1. Moreover, the figure title is very sloppy. What is an “extreme accuracy comparison”?

Furthermore, the authors have added the sentence “The inhibition capability of quenchers is inversely proportional to the relative molecular weight of the quenchers.” to their manuscript. The authors should add a literature reference to this statement.

Comment 11

OK

Comment 12

As requested, the authors have added some definition for “matter confining capability”. The authors now write in their revised manuscript, that matter confining capability “refers to the ability of quenchers to inhibition polymerization at the edge of the focal spot by chemical and physical quenching process, and thus confine the polymerization zone into a smaller space”. Unfortunately, this sentence is grammatically incorrect and should be revised. Furthermore, the term “edge of the focal spot” is

technically wrong – there is no edge in the focal spot. I recommend choosing a different term, e.g., “... in the tails of the focused laser beam”.

As requested, the authors have added some definition for their term “light confining capability”. They link this definition to the results shown in Fig. 3b. Typically Interestingly, some artefacts can be observed in the printed line crossings. Unfortunately, the authors do not show an SEM image of the same pattern printed with PR1 (the quencher-free photoresist). Without this SEM image, the reader is unable to judge whether this line-crossing artifact is caused by the introduction of quenchers. The authors should add this image to the Supplementary Figures.

Comment 13

As requested, the authors have added easy-to-read writing speed labels to Supplementary Fig. 4. However, in the caption of Supplementary Figure 4, the authors write “the laser power is controlled by the voltage”. What voltage are the authors referring to? The reviewer would highly appreciate a more concise language.

Comment 14

OK

Comment 15

OK

Comment 16

OK

Comment 17

As requested, the authors have added data for different quencher concentrations in their Stern-Volmer plots. In their manuscript, the authors still claim that “The non-linear plot at higher concentration of TEMPO in Fig. 2e and Fig. 2g, directly proves the simultaneous existence of dynamic and static quenching.” I disagree with this judgement. A nonlinear relationship between the quencher-concentration and the fluorescence lifetime is not evident from Fig. 2g. Hence, the existence of static quenching cannot be concluded. The authors should rephrase this statement accordingly.

Comment 18

Based on their observations in the Stern-Volmer experiments and the ESR experiments, the authors conclude that ground-state complexation exists for the DETC+TEMPO photoresist. As explained in Comment 17, the reviewer does not agree that the Stern-Volmer experiments indicate static quenching. However, lacking expertise in the field, the reviewer cannot judge if Fig. 2c proves ground-state complexation between DETC and TEMPO.

The reviewer appreciates the authors’ interpretation of the observed effects in the authors’ rebuttal letter.

Comment 19

OK

Comment 20

OK

Comment 21

OK

Comment 22

OK. A comment if the focal spot was characterized using gold beads or fluorescent beads would be appreciated.

Comment 23

OK

Comment 24

As requested, the authors have added an absorption spectrum of TEMPO. Unfortunately, the authors do not write in what solvent they have measured the absorption spectrum, neither do they specify the TEMPO concentration. Unfortunately, the extinction coefficient cannot be extracted from the plotted data. Interestingly, no absorption peak can be observed in the TEMPO data. However, it is well known that TEMPO solutions have a faint red color, which is indicative of some absorption band. Furthermore, the y-label of Supplementary Fig. 5b reads “intensity (a.u.)”. Hopefully, the authors did not plot the transmitted intensity but the optical density. More scientific vigor would be highly appreciated.

Comment 25

OK

Comment 26

OK

Comment 27

OK

Further Comments

The authors introduce the abbreviation “LER” for line-edge roughness. However, it seems as if the abbreviation is not used anywhere in the manuscript. The abbreviation should be removed.

Fig. 3a: Labels are cropped.

Fig. 3b shows a “loop pattern” used to test STED-MPL. Unfortunately, the schematic is mirrored with respect to the printed pattern. Moreover, the direction of printing is not shown, which is an essential information to judge the pattern. The authors MUST fix both issues.

Reviewer #2 (Remarks to the Author):

The authors have responded adequately to all our questions. The resolution achieved in the woodpile structures are very impressive.

The authors might want to be aware of work related to 3d printing at very high/ultimate resolutions here: DOI 10.1088/2399-1984/aabb94 and <https://doi.org/10.1021/acsnano.2c01999>

Reviewer #3 (Remarks to the Author):

I appreciate the explanations in the rebuttal letter and the amendments in the revised version of the manuscript. Most of my concerns have been satisfactorily addressed. Two comments:

To point 66: In fact EUV has already achieved sub-10 nm resolution (in the lab) in resist, however not with industrial projection schemes, but with interference lithography using 13.5 nm wavelength from a synchrotron. This is the resolution in resist and not in metal, as the author point out when referring to minimum Metal Pitch (LR) of "TSMC 7nm" and "Samsung 7nm". Similarly, the results presented by the authors are in resist and not in metal. So, there is still a difference and I wonder, whether this gap can be closed easily when using the new method.

In general, the extensive use supplementary information enhances the readability of the manuscript, but the reader should not need to read the supplementary information to understand the achievements of the paper. I understand that the authors still want to use the Nezha figure to illustrate the resolution, however, in Fig. 3, the difference between the figures can barely be seen - and I do not think that the size of these figures should be expanded. I would prefer to see the woodpile structure in Figure 7 from the supplementary in the main manuscript (e.e., as an additional figure), because it shows much better the resolution enhancements possible with the new technique.

Reviewer #4 (Remarks to the Author):

My primary concern is Fig.2c and all the explanations around it.

Static quenching path #1.

An ESR spectrum is characterized by lineshape and integral intensity. The latter is proportional to the number of electron spins and can be measured by double integration of the spectrum. Spin interaction may change the ESR line shape, which is often the case, but not the integral intensity, as the number of spins (radicals in this case) remains constant. If, for example, an ESR line gets broader due to a change in viscosity, the peak intensity will decrease, but the double integral will not.

A quantitative comparison of two samples with respect to the radical concentration requires an independent ESR reference. It is often a stable solid sample in a glass tube placed in the ESR resonator in a fixed position. Experimenters also use a double cavity for that purpose if available. Integral intensities of the compared spectra are measured with respect to the intensity of the EPR spectrum of the reference.

Instead of performing double integration (adds low-frequency noise), it is advised to perform EPR line

fitting. One of the fitting parameters is the integral intensity. The TEMPO spectrum is well-known and can be simulated using free software, including the commonly used <https://www.easyspin.org>. If a formation of nitroxide-photoinitiator complex happens, line fitting may reveal this interaction as a deviation from the expected at a given concentration TEMPO lineshape. There is chance here to gain an additional useful information.

The variation in the peak signal value in Fig.2c can be explained by changes in the viscosity and electric properties of the sample. The latter may affect the resonator quality factor and the detected signal intensity as a result.

The slow [up to 60 mins] signal recovery may be a result of the sample heating & cooling due to light irradiation. All other radical-related processes should occur on a much faster time scale.

Also,

I would recommend using colors with better contrast in the figure.

Please provide concentrations [mM] of both TEMPO and DETC in the figure captions; the reader can compare the TEMPO / DETC ratio when it comes to the complex formation

Quenching path #3

My experience with another ESR probe type is that it gets destroyed during photopolymerization due to the reaction with the excited photoinitiator. I never tested TEMPO in that regard. However, I would recommend a proper ESR measurement [see above] of the decline of TEMPO concentration as a function of UV exposure time.

Point-to-point Response to Reviewers

Reviewer #1 (Remarks to the Author):

Response:

We greatly appreciate the reviewer's further comments, which are very valuable. The authors' response to the reviewer's further comment #1 is still answered in a piecewise fashion. The new replies (Round 2) are highlighted in blue, and the original replies (Round 1) are highlighted in orange.

Comment 1

Original response: We appreciate the comments. The focus of our work is mainly on the proposal of a new strategy (LMC-MPL) to realize a high-precision and high-resolution lithography. And it is worth emphasizing that the acquisition and transfer of high precision patterns on silicon substrates is the first step for producing diverse micro devices, including semiconductor manufacturing. The experiment of on-silicon gratings certified the possibility of applying our strategy in above fields. That's why we put this part into the previous manuscript. Of course, that's not the point of this work. Therefore, as request by the reviewer, we have put this part in the Supplementary Information (Supplementary Fig. 19).

Reviewer's further comments:

OK

Response:

We appreciate the reviewer's recognition.

Original response: In the manuscript, we proposed that the photochemistry of DETC may generate radicals through two paths ($S_1 \rightarrow T_1$ and $S_n \rightarrow T_n$) according to the experiments, and the DFT calculations certified the probability of this energy level transition. In fact, we are not the first to make this point. Martin Wegener et al once proposed that "when using DETC without a coinitiator, polymerization is initiated via two competing pathways, one of which obeys an $N = 3$ scaling and the other follows an $N = 4$ scaling. While the first one can be depleted (as it passes through S_1 and T_1), the other one cannot be interrupted, as it passes neither the relaxed S_1 state nor the T_1 state (Fig. R1)." [Advanced Optical Materials. 3, 221-232 (2015)]. This is exactly consistent with our experiments and mechanistic explanation. But, Martin Wegener et al believe that the two polymerization paths of DETC are achieved by three photon absorption and four photon absorption because they observed that a different

nonlinearity absorption exponent ($N=3$ and $N=4$) versus repetition rate. Martin Wegener et al also got a result of $N>3$ for PETA+DETC in a more recent paper (Fig. R2) [Advanced Optical Materials. 7, 1901040 (2019)], but this time they attributed $N=4$ to the absorption of PETA monomer. Therefore, based on the above analysis and our own results (Fig. 1g), we are more inclined to think that all the absorption of photon for DETC are achieved by a three-photon absorption process. And subsequently it has two paths to go. One is it may transition from S_n to T_n state (ISC, $S_n \rightarrow T_n$) and generate free radicals at T_n state, and the other is it may return to S_1 state through vibrational relaxation (VR), then transition to T_1 state (ISC, $S_1 \rightarrow T_1$), and finally generate free radicals at T_1 state. We also cited these two references in the revised manuscript and added a brief discussion (highlight in blue, Page10).

Reviewer's further comments:

The reviewer acknowledges the expanded discussion and the authors' cautious statement "we think there is only a three-photon absorption".

Response:

We appreciate the reviewer's recognition. The reviewer's earlier comments were very meaningful and made us think very deeply about this aspect.

Original response: As for the photochemical experiments of DETC, we have further tested ESR spectra of TEMPO and TEMPO+DETC before and after UV irradiation (placed in the dark for 0 min, 20 min or 60 min) in Fig. 2c. Additionally, we also supplemented the fluorescence decays spectra of the photoresists (DETC) with different concentration (0-40 mM) of TEMPO in Fig. 2f and the Stern-Volmer plots of fluorescence lifetime of DETC in Fig. 2g. The linear change of the lifetime indicates that dynamic quenching dominates at lower concentration of TEMPO (Fig. 2g). In the case of higher concentrations of TEMPO, the two processes (static and dynamic quenching) may be competitive, which results in a nonlinear relationship of lifetime and fluorescence. The combination of static and dynamic quenching almost completely quenched the fluorescence of Pr2 (Supplementary Fig. 5a and Fig. 2d). These further discussions about the photochemical process have been discussed in the revised manuscript (highlight in blue, Page7).

Reviewer's further comments:

The reviewer acknowledges these changes. These changes are discussed in comments 17.

Response:

We appreciate the comments. The further replies are provided in comments 17, as the reviewer mentioned.

Original response: DETC (7-diethylamino-3-thenoylcoumarin) was chosen as the photoinitiator in our photoresist because the initiator must not only have photoinhibition properties but also be wavelength-matched to our lasers. Up to now, there are only few relatively mature initiators with photoinhibition properties in MPL, such as DETC and ITX (Isopropyl thioxanthone) [Laser & Photonics Reviews. 16, 2100229 (2022)]. So, we don't have much choice for initiators with photoinhibition property. As for why we didn't choose ITX as another research object, because the photophysical and photochemical mechanism of ITX and DETC are not exactly same, such as their different performance in the pump-probe experiments (Fig. R3) [Opt. Lett. 36, 3188-3190 (2011)]. Therefore, the study on the interaction between DETC and TEMPO in this paper has its own uniqueness and the research on other photoinitiators is of little significance to the discussion and research based on DETC in this paper. That is why our entire work focuses on DETC and does not discuss other photoinitiators. Of course, we believe that using quenchers as a tool to study the photophysical and photochemical mechanisms of other photoinitiators, such as ITX, would be another feasible and interesting work.

Reviewer's further comments:

In the reviewer's opinion, additional experiments using ITX or another depletable photoinitiator are not required. The reviewer's original comment aimed at underlining the complexity of DETC's photochemistry and the urgent need for a dedicated DETC review paper summarizing and discussing DETC's photochemistry in detail.

Response:

We appreciate the comments. We cannot agree more that the DETC's photochemistry/photophysics are very complex, especially in the field of MPL. New mechanisms and understandings of DETC are being developed, and until now there has been a lack of a dedicated DETC review paper summarizing and discussing DETC's photochemistry in detail. In our opinion, there are two important research papers about DETC that can be consulted for the readers: 1. Exploring the Mechanisms in STED-Enhanced Direct Laser Writing, Adv. Optical Mater. 2014, DOI: 10.1002/adom.201400413; 2. Polymerization

Inhibition by Triplet State Absorption for Nanometer-Scale Lithography, Adv. Mater. 2013, DOI: 10.1002/adma.201204141. However, in these two papers, they had proposed very different mechanisms. And a brief overview had been given in our manuscript.

In the future, we will be happy to write a dedicated review paper summarizing and discussing DETC's photochemistry/photophysics in detail for community.

Thanks to the reviewers for the inspiration.

Original response: The fluorescence and ESR experiments were indeed performed ex-situ which has been emphasized in the revised manuscript (highlight in blue, Page7) and Methods (PhotoPhysical characterization) section.

Reviewer's further comments:

OK, this experimental detail is of prime importance.

Response:

We appreciate the comments. We apologize that this experimental detail was missing from the initial manuscript.

Original response: In fact, we had initially hoped to conduct an in situ test as well, but were not successful. We have not found commercial instruments for us to perform the in-situ fluorescence and ESR experiments, since such experiments are extremely rare and demanding. We have also tried our best to build our own instruments for in-situ fluorescence testing within the recent three months, but we failed. We found that not only did it take a long time, but it was so expensive that we couldn't afford it.

Reviewer's further comments:

The reviewer appreciates the effort.

Response:

We appreciate the comments. It's our responsibility to make that effort.

Original response: As for in-situ ESR test, it is theoretically not feasible because the laser beam will be focused into a point inside the photoresist and the radical distribution generated at this point cannot be tested by the current ESR instrument. In addition, although the ex-situ fluorescence and ESR tests cannot take the influence of oxygen (diffusion lengths and timescales) into account, this does not negate our use of ex-situ

tests to explore the quenching mechanism of TEMPO. It can only be said that our ex-situ test cannot take into all the factors during the in situ lithography, and it can indeed partially reflect the influence of TEMPO on the fluorescence and ESR signal of DETC. And these effects must be taken into account in the analysis of the in-situ MPL.

Reviewer's further comments:

The reviewer acknowledges that the authors disclosed that the experiments were conducted ex-situ in their revised manuscript.

Response:

We appreciate the comments. We apologize that this experimental detail was missing from the initial manuscript.

Comment 2

The authors demonstrate that woodpile photonic crystals with rod spacings below 200 nm can be fabricated using LMC-MPL. Unfortunately, they keep the laser power constant when comparing LMC-MPL with MC-MPL and MPL. This is an unfair comparison, and this has been addressed in the original review comment #9. The authors should fabricate woodpiles using the threshold laser-power for each photoresist. The reviewer acknowledges that this requires effort, but the results are vital for the conclusions drawn in the manuscript.

Response:

We appreciate the comments. As the reviewer requested, we have conducted the extra experiments and added the results in the revised manuscript (Supplementary Fig. 7b). As the result shows, the respective minimum period of woodpile structures achieved by MC-MPL and MPL are 225 nm and 300 nm at their threshold excitation laser power, which are larger than LMC-MPL (200 nm). Therefore, it also proves the advancement of our strategy. Besides, relevant discussion also has been modified in the revised manuscript (highlighted in blue, Page 8).

Comment 3

The authors have expanded their methods section and included more detailed information about the setup. In their response to the reviewer's comments, they specify an objective lens numerical aperture of 1.4, whereas in the main text they specify an objective lens numerical aperture of 1.45. The authors should clarify which one is correct. Furthermore, the authors should be more precise and indicate the full model name of the objective lens in the main text. In their methods section on the photo-physical characterization, the authors write "interference factors". To me, it is unclear what interference factors the authors are referring to. Please specify or remove.

We appreciate the comments. We apologize for making this mistake. We have carefully confirmed that the numerical aperture of the objective lens should be 1.45 (Fig R1), and meanwhile we have provided the full model name of the objective lens (Olympus, UPlanXApo, 100x, NA=1.45), in the revised manuscript (highlighted in blue below Fig.1, Page3).

Fig. R1 Photo of the objective lens we used.

In terms of "interference factors", it refers to the effects of solvents on photophysical properties of the analytes. It is known that solvents can have a great impact on the photophysical properties of molecule. Sensors and Actuators B 245 (2017) 845–852. Because DETC and TEMPO are dissolved in the monomer for lithography, we chose the monomer (87.5 wt% TCDA + 12.5 wt% EBPFDA- OPPEA) as the solvent for photo-physical characterization as well. To avoid the confusion or misunderstanding from readers, we have replaced "To eliminate interference factors as much as possible" by "To eliminate the influence of solvents" in the revised manuscript (highlighted in blue, Page15).

Comment 4

In their revised manuscript, the authors have mentioned that the mechanistic TEMPO investigation was performed in-situ by writing “All the above conclusions are based on ex-situ tests, while MPL (in-situ) will be limited by space and time (such as quencher diffusion) as well.” I don’t understand the second part of the sentence, starting with “while MPL”. The authors must state clearly and concisely what they mean. I fully agree with the authors, that not all experiments can be conducted in-situ with state-of the-art equipment. Nevertheless, it is important to make the reader aware that ex-situ experiments have a limited validity.

Response:

We appreciate the comments. In terms of sentence “All the above conclusions are based on ex-situ tests, while MPL (in-situ) will be limited by space and time (such as quencher diffusion) as well.” What we want to express is all the above tests are performed ex-situ, but MPL is actually an in-situ process which will also be affected by oxygen, diffusion lengths, timescales etc. So, these ex-situ experiments have a limited validity.

To make our statement more accurate and clearer, we have modified the expression here, as follow: “*All the above conclusions are based on ex-situ tests. Therefore, it is undeniable that they have a limited validity in exploring the mechanism for in-situ MPL*”. (highlighted in blue, Page 8)

Comment 5

As requested, the authors have specified that the lithography experiments were conducted under ambient atmosphere in their revised methods section. The authors should likewise specify in the photo-physical-characterization section and EPR section that these experiments were also conducted under ambient atmosphere, i.e., normal air.

Response:

We appreciate the comments. As requested by reviewer, we have specified that all the photo-physical-characterization and EPR experiments are conducted in air. (highlighted in blue, Page 15 and Page 16)

Comment 6

OK

Response:

We appreciate the reviewer's recognition.

Comment 7

References for two claims made in the original paper were requested. The reference for the first claim is OK. Concerning the second claim, "It is worth noting that the quencher will be consumed rapidly in the exposed area, leading to a continuous diffusion of the quencher, which will further suppress the proximity effect.", the authors now provide two references. In the first reference, "ACS Nano. 6, 2302-2311 (2012)", 2-(dimethylamino)ethyl methacrylate is used as quencher. In contrast, the authors of the present paper refer to a quencher of the hindered-amine light stabilizer group. It is highly questionable if the two quenchers behave comparably. The second reference, "AIP Advances. 5, 127215 (2015)", assumes that the quencher is consumed rapidly. However, no proof there of is provided. It is known that hindered-amine light stabilizers, like TEMPO or BTPOS, can recover within the Denisov-cycle. As of now, it is unclear if this is also the case in multi-photon 3D printing. Hence, the authors should rephrase their statement and clarify that the quencher consumption is an assumption the authors tacitly make.

Response:

We appreciate the comments. It is a good question. As mentioned by reviewer, in "ACS Nano. 6, 2302-2311 (2012)", 2-(dimethylamino)ethyl methacrylate is used as quencher which is different from that we used. However, regardless of the quencher used, its diffusion in the photoresist is inevitable, which is a physical process independent of the quenching mechanism. Thus, we think it is no problem to cited this reference to claim "a continuous diffusion

of the quencher". For the second reference, "AIP Advances. 5, 127215 (2015)", it has the same author as the first reference "ACS Nano. 6, 2302-2311 (2012)". The second reference is a simulation study based on experiments conducted in the first reference. Theoretically, once a free radical is created, the quencher will inevitably bind to the free radical and be consumed. However, it is true that we cannot provide experimental results to demonstrate the process of free radical consumption. Therefore, in the revised manuscript, as requested by reviewer, we have rephrased statement and clarified that the quencher consumption is an assumption (highlighted in blue, Page 2). The sentence was modified to "*Notably, if the quencher can be consumed rapidly in the exposed area...*"

In terms of TEMPO/BTPOS or "Denisov-cycle", which as far as we know is often used in the field of reversible reactive polymerization of polymers, TEMPO will form a dormant species with the generated free radicals and release free radicals again at a certain temperature (generally above room temperature). In addition, this process ("Denisov-cycle") generally occurs on much longer time scales, whereas in MPL lithography the time scales are very short, typically on the order of fs to μ s. Therefore, we believe that there is no need to consider this cycle in MPL.

Comment 8

In the original statement, "More importantly, it seems impossible to realize a decent LR since the proximity effect is still there.", the authors have replaced the word "decent" by "better". "Better" is a comparative word. It is not entirely clear to me, with what the authors are comparing the LR. I assume that the authors want to say that the LR in Minfei He et al.'s work is still limited by the proximity effect. Please clarify.

Response:

We appreciate the comments. The reviewer's guess was correct. We do want to say that the LR in Minfei He et al.'s work is still limited by the proximity effect. Therefore, we've further modified the sentence (as shown in italics below) in the revised manuscript. (highlighted in blue, Page 2)

The sentence was modified to *“More importantly, the LR in Minfei He et al.’s work is still limited by the proximity effect.”*

Comment 9

OK

Response:

We appreciate the reviewer’s recognition.

Comment 10

The reviewer requested to perform linewidth measurements for the six different photoresist at their respective threshold laser power instead of keeping the laser power constant for all six photoresists. The authors now have added results of this experiment and show corresponding data in the Supplementary Fig. 2. The authors should add the laser powers that they have used for the respective lines. This is crucial since, unfortunately, the laser threshold laser powers at a scan speed of 50 $\mu\text{m/s}$ cannot be extracted from Supplementary Figure 1. Moreover, the figure title is very sloppy. What is an “extreme accuracy comparison”? Furthermore, the authors have added the sentence “The inhibition capability of quenchers is inversely proportional to the relative molecular weight of the quenchers.” to their manuscript. The authors should add a literature reference to this statement.

Response:

We appreciate the comments. As requested, we have provided the respective laser powers in the Supplementary Fig. 2. Meanwhile, “extreme accuracy comparison” has been replaced by “Comparison of minimum critical dimension” in the Supplementary Fig. 2, and they represent the minimum linewidths that the six photoresists can achieve at each threshold laser power.

In terms of statement of “The inhibition capability of quenchers is inversely proportional to the relative molecular weight of the quenchers”, this statement is derived from our experimental results in Fig. 1d, not from any reference cited. In Fig. 1d, we used five quenchers whose molecular weights are gradually increasing, while their ability to narrow the linewidth at the same laser power is

gradually decreasing. From this we make this statement (“Fig. 1d shows that the inhibition capability of quenchers is inversely proportional to the relative molecular weight of the quenchers”), and further give possible reasons: “An increase in molecular weight might lead to a decrease in the active site freedom and molecular diffusion coefficient, thus causing inefficient kinetic processes of quenching”. we have modified the statement in the revised manuscript. (highlighted in blue, Page 4)

Meanwhile we have also found relevant literatures to support the possible reasons we gived, and included them in the revised manuscript. (highlighted in blue, Page 4). Chemosphere. 119, 498-503 (2015). Journal of Electroanalytical Chemistry. 681, 121-126 (2012)

Comment 11

OK

Response:

We appreciate the reviewer’s recognition.

Comment 12

As requested, the authors have added some definition for “matter confining capability”. The authors now write in their revised manuscript, that matter confining capability “refers to the ability of quenchers to inhibition polymerization at the edge of the focal spot by chemical and physical quenching process, and thus confine the polymerization zone into a smaller space”. Unfortunately, this sentence is grammatically incorrect and should be revised. Furthermore, the term “edge of the focal spot” is technically wrong – there is no edge in the focal spot. I recommend choosing a different term, e.g., “... in the tails of the focused laser beam”. As requested, the authors have added some definition for their term “light confining capability”. They link this definition to the results shown in Fig. 3b. Typically Interestingly, some artifact can be observed in the printed line crossings. Unfortunately, the authors do not show an SEM image of the same pattern printed with PR1 (the quencher-free photoresist). Without this SEM image, the reader is unable to judge whether this line crossing

artifact is caused by the introduction of quenchers. The authors should add this image to the Supplementary Figures.

Response:

We appreciate the comments and kind remind. We have fixed the grammatical and technical errors mentioned by the reviewer, and the revised sentence is as follows “*The matter confining capability of various quenchers, which refers to their ability to inhibit polymerization in the tails of the focused laser beam by chemical and physical quenching processes and thus confine the polymerization zone into a smaller space, was first investigated.*” (highlighted in blue, Page 3)

Fig. R2 Light confining capability tests: a standard pattern, where the excitation beam always keeps on and the inhibition beam was only switched on at the green area, and the resultant SEM images for **Pr1** (the quencher-free photoresist) and **Pr2**. Scale bar: 3 μm .

In fact, we had provided the SEM image of the same pattern printed with PR1 (the quencher-free photoresist), as shown in the bottom right of Fig. 3b and Fig. R2. It can be seen that this line crossing artifact exists with or without a quencher. That is, the artifact is not caused by the introduction of quenchers. In our opinion, due to the Gaussian distribution of the femtosecond laser energy, the artifact is caused by the residual polymerization of photoresist, after all, light confining capability is not strong enough to completely eliminate the DETC in the excited state through the STED path. Nevertheless, this does not affect our conclusions.

Comment 13

As requested, the authors have added easy-to-read writing speed labels to Supplementary Fig. 4. However, in the caption of Supplementary Figure 4, the authors write “the laser power is controlled by the voltage”. What voltage are the authors referring to? The reviewer would highly appreciate a more concise language.

Response:

We appreciate the comments. In the lithography system (as shown in Supplementary Fig. 6), the laser power is controlled by the voltage of acoustic optical modulator. Since the reviewer wonder “..it is unclear in what steps the laser power has been increased..” in the original comment 13, so we provided the plot in Supplementary Fig. 4c to give a clear instruction. We have changed this sentence in the revised manuscript as follows: "The laser power is controlled by the voltage of the acousto-optic modulator" (highlight blue below Supplementary Fig. 4).

Comment 14

OK

Response:

We appreciate the reviewer’s recognition.

Comment 15

OK

Response:

We appreciate the reviewer’s recognition.

Comment 16

OK

Response:

We appreciate the reviewer’s recognition.

Comment 17

As requested, the authors have added data for different quencher concentrations in their Stern-Volmer plots. In their manuscript, the authors still claim that “The non-linear plot at higher concentration of TEMPO in Fig. 2e and Fig. 2g, directly proves the simultaneous existence of dynamic and static quenching.” I disagree with this judgement. A nonlinear relationship between the quencher-concentration and the fluorescence lifetime is not evident from Fig. 2g. Hence, the existence of static quenching cannot be concluded. The authors should rephrase this statement accordingly.

Response:

We appreciate the comments. First, Stern-Volmer usually is used to study the relationship between fluorescence intensity and the concentration of quenchers [$\frac{I_0}{I} = 1 + K_{SV}Q$]. Therefore, we first select Fig 2e to observe the linear relationship of Stern-Volmer. Second, a linear Stern-Volmer relationship may be observed if either a static (ground state interaction) or dynamic (excited state interaction) quenching process is dominant. *J. Am. Chem. Soc.* 125, 3821-3830 (2003). Chapter 8, Principles of Fluorescence Spectroscopy.

In Fig. 2e, the Stern-Volmer plot shows a linear relationship at low concentration (less than 20 mM) and a non-linear relationship at high concentration. Therefore, one kind of quenching path dominates at low concentration, while, at high concentrations, simultaneous existence of two quenching paths. *J. Am. Chem. Soc.* 125, 3821-3830 (2003). That is, the results in Fig. 2e are sufficient for us to justify this conclusion (*co-existence of dynamic and static quenching*).

As requested by the other reviewer (Round 1), we added the Stern-Volmer plot of fluorescence lifetime (Fig 2f-g) to figure out which quenching path dominates at low concentrations. Obviously, the fluorescence lifetime of DETC decreases linearly at low concentrations. This demonstrates that dynamic quenching dominates at low concentration. As for the situation at high concentration of the quencher, the slope of the curve in Fig. 2g at high concentration decreases slightly (or even if we assume it remains the same), indicating the dynamic quenching ability of the quencher decreases (or remains the same). However, in Fig. 2e, the slope of the curve (K_{SV}) increases at high

concentration, which indicates that the overall quenching ability of the quencher is enhanced, i.e., static quenching must exist. Therefore, we conclude that static quenching exists mainly based on the nonlinearity phenomenon in Fig 2e instead of Fig. 2g.

In summary, the existence of static quenching (ground state interaction) can be concluded. However, we are indeed unable to determine whether the static quenching is due to the formation of a ground state complex (DETC-TEMPO). Because it is difficult to characterize the formation of ground state complexes in a direct way, we gave this conclusion just based on the reference that "static process, by the quenching of a bound complex" *J. Am. Chem. Soc.* 125, 3821-3830 (2003). Chapter 8, Principles of Fluorescence Spectroscopy.

Therefore, the expression here is indeed not rigorous enough, so we have modified this part. We replaced the expression "ground state complex" with "ground state interaction", and supplemented the above explanation to the revised manuscript (highlighted in blue Page7).

Comment 18

Based on their observations in the Stern-Volmer experiments and the ESR experiments, the authors conclude that ground-state complexation exists for the DETC+TEMPO photoresist. As explained in Comment 17, the reviewer does not agree that the Stern-Volmer experiments indicate static quenching. However, lacking expertise in the field, the reviewer cannot judge if Fig. 2c proofs ground state complexation between DETC and TEMPO. The reviewer appreciates the authors' interpretation of the observed effects in the authors' rebuttal letter.

Response:

We appreciate the comments. As mentioned in comment 17, we have fully explained why according to Fig. 2e and Fig. 2g, we can conclude that static quenching and dynamic quenching coexist. And we replaced the expression "ground state complex" with "ground state interaction", because it is difficult to provide direct evidence for the ground state complex.

As for the ESR experiments, it is used to assist in proving there is indeed interaction between ground state DETC and TEMPO. As the fourth reviewer

said “An ESR spectrum is characterized by line shape and integral intensity. The latter is proportional to the number of electron spins and can be measured by double integration of the spectrum.” We have given the number of electron spins (Supplementary Fig. 5a) by double integration of the spectrum (Fig. 2c). It can be seen that DETC will cause the ESR signal of TEMPO to decrease, and the decrease in the number of electron spin in the solution (Supplementary Fig. 5a), and thus indicates that there must be some interaction between the ground state DETC and TEMPO. A more detailed discussion and explanation can be seen in **Comments 32-27**.

We also added relevant information to the revised manuscript. (highlight blue Page 7)

Comment 19

OK

Response:

We appreciate the reviewer’s recognition.

Comment 20

OK

Response:

We appreciate the reviewer’s recognition.

Comment 21

OK

Response:

We appreciate the reviewer’s recognition.

Comment 22

OK. A comment if the focal spot was characterized using gold beads or fluorescent beads would be appreciated.

Response:

We appreciate the comments. In Fig.R7 (in original response to comment 22), the focal spot was indeed characterized using gold beads.

Comment 23

OK

Response:

We appreciate the reviewer's recognition.

Comment 24

As requested, the authors have added an absorption spectrum of TEMPO. Unfortunately, the authors do not write in what solvent they have measured the absorption spectrum, neither do they specify the TEMPO concentration. Unfortunately, the extinction coefficient cannot be extracted from the plotted data. Interestingly, no absorption peak can be observed in the TEMPO data. However, it is well known that TEMPO solutions have a faint red color, which is indicative of some absorption band. Furthermore, the y-label of Supplementary Fig. 5b reads "intensity (a.u.)". Hopefully, the authors did not plot the transmitted intensity but the optical density. More scientific vigor would be highly appreciated.

Response:

We appreciate the comments. We have given the solvent and the concentrations of the analytes below Supplementary Fig. 5. "UV-vis absorption spectra of DETC (3.1×10^{-7} M), TEMPO (4.1×10^{-7} M) and DETC (3.1×10^{-7} M) +TEMPO (4.1×10^{-7} M). All the tested substances were dissolved in the monomer of the photoresist (87.5 wt% TCDA + 12.5 wt% EBPFDA- OPPEA) for testing, and the ratio of DETC and TEMPO is consistent with Pr2."

As for "no absorption peak of TEMPO data", we feel sorry for making a mistake here. In the original absorption spectrum, Pr1 (DETC in monomer) and Pr2 (DETC+TEMPO in monomer) are not actual photoresists, but the samples diluted with monomers (about 100,000 times). Because if the concentration of DETC remains consistent with Pr1 or Pr2, it will exceed the absorbance range of UV-vis absorption spectrum. Therefore, for TEMPO, we also first prepared a

TEMPO solution with the same concentrations as that in Pr2, and then diluted it 100,000 times for testing. All these tests (including the UV-vis absorption spectrum of TEMPO) were all done before we first submitted the article, so we forgot to label the real test concentrations because it's been too long. Since the absorption of TEMPO is not very strong, it has almost no absorption (350-550 nm) after being diluted 100,000 times. As shown in Fig. R3, we tested the absorption spectra of TEMPO at different concentrations. TEMPO at a concentration of 4.1×10^{-2} M (the concentration is consistent with Pr2) has an absorption peak at 464 nm, but when it is diluted to 4.1×10^{-4} M, the absorption peak is already very weak. When diluted to 4.1×10^{-7} M, the absorption peak almost disappears. As shown in Fig. R4, it can be seen that after diluting 100 times, the TEMPO sample almost has no color, but the color of the DETC and DETC+TEMPO is still very obvious.

We have modified the y-label in Supplementary Fig. 5c, "Absorbance (a.u.)". We also given the scale value of y-axis. It is indeed absorption intensity instead of transmission intensity. Additionally, we have modified the statement about TEMPO absorption in the revised manuscript (highlight blue Page 8).

Fig. R3 UV-vis absorption spectra of TEMPO at different concentrations in monomer.

Fig. R4 Left: samples consistent with the concentration of Pr2. Right: Sample diluted 100 times using the monomer.

Comment 25

OK

Response:

We appreciate the reviewer's recognition.

Comment 26

OK

Response:

We appreciate the reviewer's recognition.

Comment 27

OK

Response:

We appreciate the reviewer's recognition.

Comment 28 (Further Comments)

The authors introduce the abbreviation "LER" for line-edge roughness. However, it seems as if the abbreviation is not used anywhere in the manuscript. The abbreviation should be removed. Fig. 3a: Labels are cropped. Fig. 3b shows a "loop pattern" used to test STED-MPL. Unfortunately, the schematic is mirrored with respect to the printed pattern. Moreover, the direction of printing

is not shown, which is an essential information to judge the pattern. The authors must fix both issues.

Response:

We appreciate the comments. As requested, LER has been removed in the revised manuscript (highlight blue Page 2). We have modified the labels in Fig. 3a. We have corrected the “loop pattern” in Fig. 3b, and the direction of printing also has been added (Fig. 3b and Fig. R2).

Reviewer #2 (Remarks to the Author):**Comment 29**

The authors have responded adequately to all our questions. The resolution achieved in the woodpile structures are very impressive. The authors might want to be aware of work related to 3d printing at very high/ultimate resolutions here: DOI 10.1088/2399-1984/aabb94 and <https://doi.org/10.1021/acsnano.2c01999>

Response:

We appreciate the reviewer's recognition. We have carefully read the literature provided by the reviewers, it is very interesting and impressive. It has been cited in the introduction section in our manuscript, and we might refer to it for some extension work in future.

Reviewer #3 (Remarks to the Author):

I appreciate the explanations in the rebuttal letter and the amendments in the revised version of the manuscript. Most of my concerns have been satisfactorily addressed. Two comments:

Response:

We appreciate the reviewer's recognition.

Comment 30

To point 66: In fact EUV has already achieved sub-10 nm resolution (in the lab) in resist, however not with industrial projection schemes, but with interference lithography using 13.5 nm wavelength from a synchrotron. This is the resolution in resist and not in metal, as the author point out when referring to minimum Metal Pitch (LR) of "TSMC 7nm" and "Samsung 7nm". Similarly, the results presented by the authors are in resist and not in metal. So, there is still a difference and I wonder, whether this gap can be closed easily when using the new method.

Response:

We are very grateful to the reviewer for patient explanations and comments. As mentioned in our manuscript, 100 nm resolution is almost the best resolution currently available in the field of MPL. We fully acknowledge that there is still a very big gap between MPL and EUV lithography. If we want to achieve sub-10 nm resolution on substrate through LMC-MPL, this is by no means "easy". As we mentioned in the previous answer, the resolution we achieved is in resist instead of in metal. We believe that if some process methods, such as double exposure, are used, it is possible to achieve higher resolutions in metal for MPL. Moreover, with the continuous efforts of researchers and the use of various new methods, we believe that the writing accuracy of MPL will continue to improve in the future.

Comment 31

In general, the extensive use supplementary information enhances the readability of the manuscript, but the reader should not need to read the

supplementary information to understand the achievements of the paper. I understand that the authors still want to use the *Nezha* figure to illustrate the resolution, however, in Fig. 3, the difference between the figures can barely be seen - and I do not think that the size of these figures should be expanded. I would prefer to see the woodpile structure in Figure 7 from the supplementary in the main manuscript (e.e., as an additional figure), because it shows much better the resolution enhancements possible with the new technique.

Response:

We appreciate the reviewer's comments. At the request of the reviewer, we have put the woodpile structure into the revised manuscript (Fig. 3c and Fig. 3d), and the other pictures (*Nezha*) was moved to the Supplementary Information (Supplementary Fig. 8). We also adjusted the description order of this part in the manuscript.

Reviewer #4 (Remarks to the Author):

My primary concern is Fig.2c and all the explanations around it.

Comment 32

Static quenching path #1.

An ESR spectrum is characterized by line shape and integral intensity. The latter is proportional to the number of electron spins and can be measured by double integration of the spectrum. Spin interaction may change the ESR line shape, which is often the case, but not the integral intensity, as the number of spins (radicals in this case) remains constant. If, for example, an ESR line gets broader due to a change in viscosity, the peak intensity will decrease, but the double integral will not.

Response:

We appreciate the reviewer's professional comments. In order to better understand ESR, we have specifically consulted with experts in the field. It is true that the integral intensity is generally determined only by the numbers of electron spins in the sample, while the line shape is generally related to the kind of spin electron.

All our EPR tests are performed in a solvent (CH_2Cl_2) and there is no photoresist polymerization during exposure, so the viscosity of the sample should remain almost constant during our test. Meanwhile, in our ESR results, both peak intensity and width increase or decrease simultaneously.

A quantitative comparison of two samples with respect to the radical concentration requires an independent ESR reference. It is often a stable solid sample in a glass tube placed in the ESR resonator in a fixed position. Experimenters also use a double cavity for that purpose if available. Integral intensities of the compared spectra are measured with respect to the intensity of the EPR spectrum of the reference.

Response:

We appreciate the reviewer's professional comments. As mentioned by reviewer, a standard sample should be used for absolute quantitative analysis of electron spins. But this is non-essential for us, we don't need to know what

the concentration of spin electrons in the sample is, we are only concerned with the trend of its concentration. This is enough to give us proof for our mechanistic analysis.

Comment 33

Instead of performing double integration (adds low-frequency noise), it is advised to perform EPR line fitting. One of the fitting parameters is the integral intensity. The TEMPO spectrum is well-known and can be simulated using free software, including the commonly used <https://www.easyspin.org>. If a formation of nitroxide-photoinitiator complex happens, line fitting may reveal this interaction as a deviation from the expected at a given concentration TEMPO lineshape. There is chance here to gain an additional useful information.

Response:

We appreciate the comments. At the request of the reviewer, we performed the EPR line fitting for TEMPO and TEMPO+DETC before UV in Fig. R5. Judging from the fitted spectrum, there is no obvious change in the peak shape of TEMPO after the addition of DETC, which is also the same in the original spectrum (Fig 2c). On the one hand, the ESR spectrum after fitting may be shielded from some weak but effective signals/information; on the other hand, the ESR signal of TEMPO itself is very strong, so the spectrum does not have many interfering signals. Therefore, we chose plot using unfitted raw data in our manuscript.

Fig. R5 Fitted ESR spectra of TEMPO before UV and TEMPO+DETC before UV (molar ratio = 1 : 5) in dichloromethane.

Comment 34

The variation in the peak signal value in Fig. 2c can be explained by changes in the viscosity and electric properties of the sample. The latter may affect the resonator quality factor and the detected signal intensity as a result.

Response:

We appreciate the reviewer's professional comments. The reviewer proposed that "the variation in the peak signal value in Fig. 2c can be explained by changes in the viscosity and electric properties of the sample". TEMPO and TEMPO+DETC is dissolved in CH₂Cl₂ for the ESR test (marked below Fig 2 Page 6). Theoretically, the addition of DETC (1 mM/L) is unlikely to cause obvious changes in the viscosity and electrical properties of the solution. I am not sure whether the reviewer mistakenly think the TEMPO or DETC was dissolved in the photoresist. Furthermore, the reviewer may mistakenly believe the polymerization of the sample will change the viscosity and electric properties of the sample.

Comment 35

The slow [up to 60 mins] signal recovery may be a result of the sample heating & cooling due to light irradiation. All other radical-related processes should occur on a much faster time scale.

Response:

We appreciate the comments. In our original manuscript (Round 1), we had not studied whether the ESR signal can recover, after turning off the UV lamp. At the request of a reviewer, we supplemented the experiments. After finding that the signal of TEMPO does recover after removing the UV light, we propose possible reasonable explanations (a radical-related processes). And we did ignore the effect of temperature.

Fig. R6 shows our test conditions: an optical fiber very close to the sample is used to receive the signal, and the UV lamp irradiation time is 10 minutes (it

may indeed cause temperature changes). Therefore, we agree with some of the reviewer's points, temperature is indeed should be taken into account considering the time scale.

More importantly, the ESR signal of TEMPO remains unchanged before and after UV irradiation (Fig. 2c), which illustrate that whether it is the light itself or the temperature changes caused by UV irradiation, will not directly affect the intensity of TEMPO signal. But, when DETC was introduced, things have changed. The signal of TEMPO+DETC is enhanced under UV, and there is also a recovery process. All these proved that not the UV but the DETC causing the change of the ESR signal.

Thus, we believe that the excitation of DETC by the UV light can weaken the static interaction between DETC and TEMPO, resulting in the release of free TEMPO, which causes an increase in ESR intensity. And in the early stages of turning off the UV lamps, the excited DETC returns to the ground state, so the intensity of the ESR peak will recover.

Besides, it is also probably that temperature changes caused by UV irradiation may also have an impact on the interaction between ground state DETC and TEMPO, especially on longer time scales (up to 60 min). Under UV irradiation, temperature rises and the ESR signal of TEMPO+DETC increases. After turning off the UV lamp, the temperature drops and the signal recovers.

We have re-discussed the question in the revised manuscript, and the effect of temperature have been added.(highlighted in blue, Page7)

Fig. R6 Left: sample preparation; Right: the device for ESR.

Comment 36

Also, I would recommend using colors with better contrast in the figure. Please provide concentrations [mM] of both TEMPO and DETC in the figure captions; the reader can compare the TEMPO / DETC ratio when it comes to the complex formation

Response:

We appreciate the comments. We have changed the colors in Fig 2c making it clearer. We have provided the concentrations of TEMPO (0.2 mM) and DETC (1 mM) in CH₂Cl₂ in the Method Section (highlighted in blue, Page 16).

Comment 37

Quenching path #3

My experience with another ESR probe type is that it gets destroyed during photopolymerization due to the reaction with the excited photoinitiator. I never tested TEMPO in that regard. However, I would recommend a proper ESR measurement [see above] of the decline of TEMPO concentration as a function of UV exposure time.

Response:

We appreciate the comments. Quenching path 3 means active free radical generated by the photoinitiator scavenging by nitroxide radicals in TEMPO. It is a well-known property of TEMPO, which is supported by many literatures, such as *Fuel Process. Technol.* 171, 350-360 (2018). So, we did not use ESR to prove the existence of quenching path 3 in the manuscript. Our original intention is to explain: The quenching path 3 will consume TEMPO through free radicals scavenging. So, in theory, quenching path 3 will lead to a decrease in the ESR spectrum of TEMPO+DETC under UV irradiation. However, in fact, the ESR signal of TEMPO+DETC in Fig. 2 is enhanced after UV irradiation, possible reasons have been proposed in **Comments 35**. So, the enhancement of ESR caused by UV irradiation is greater than the weakening capacity of quenching path 3.

In addition, we are grateful to the reviewers for providing the case. One thing I am not sure about is whether the ESR probe used by the reviewer is destroyed due to "polymerization" or caused by photo excitation. Because the solvent chosen for our ESR test is CH₂Cl₂, only photo excitation will occur during UV irradiation for ESR test, and no polymerization happens.

We also provide ESR test results of TEMPO and DETC with different ratios. As shown in Fig. R7, changing the ratio of TEMPO and DETC does not affect the trend of the ESR signal (Fig. R7b changes more obviously). So, the same conclusion can be drawn as for that in **Comments 35**.

Fig. R7 ESR spectra of TEMPO and TEMPO+DETC in dichloromethane before UV irradiation and after UV irradiation (placed in the dark for 0 min, 20 min or 60 min). The irradiation duration is 10 min by a 365 nm light. **a** The molar ratio of TEMPO and DETC is 1 : 5. **b** The molar ratio of TEMPO and DETC is 1 : 10.

REVIEWER COMMENTS

Reviewer #1 (Remarks to the Author):

The reviewer acknowledges that all his comments were addressed. The author's efforts are appreciated. The reviewer has no further comments.

Reviewer #3 (Remarks to the Author):

I have looked through the new report and I have seen that my two comments were considered, i.e., there is a response to the first one (but no change in the manuscript) and there seems to be a change of figures that I recommended in my second comment. I see that this has been changed in Figure 3. So from my point of view, this is a sufficient and valuable modification.

Reviewer #4 (Remarks to the Author):

I value the diligence demonstrated by the authors in responding to my feedback. Nevertheless, certain methodological approaches of the ESR experiments remain inadequately addressed.

1. The authors agreed with me that the integral intensity of ESR spectra is proportional to the number of electron spins. In the figure 2 (>>>), the electron spin is clearly present in the 'static complex'. Therefore, the number of spins does not change. Therefore, the integral intensity of ESR spectra should not change either.

I suggest an alternative redox hypothesis (below). Irradiation causes electron transfer and spin reactivation:

2. My suggestion to utilize a reference sample was not for absolute spin counting, but rather to facilitate the comparison of intensities across different samples. ESR spectroscopy employs such a reference for the normalization of relative intensities, which are influenced not only by the number of spins but also by other factors, including cavity loading and quality factor. In a stable experiment when measuring the same sample, this procedure could, in principle, be disregarded.

3. I proposed the application of ESR line fitting to enhance the precision of intensity measurements. This approach appears to be challenging, as the experimental TEMPO spectrum (illustrated in the right portion of the figure below) does not align well with the anticipated ESR lineshape (depicted in the left portion of the figure below). Notably, there is a discrepancy with a five-line spectrum observed, as opposed to the expected three-line spectrum.

Is it possible that the TEMPO solution was not pure?

Point-to-point Response to Reviewers

Reviewer #1 (Remarks to the Author):

The reviewer acknowledges that all his comments were addressed. The author's efforts are appreciated. The reviewer has no further comments.

Response:

We appreciate the reviewer's recognition.

Reviewer #3 (Remarks to the Author):

I have looked through the new report and I have seen that my two comments were considered, i.e., there is a response to the first one (but no change in the manuscript) and there seems to be a change of figures that I recommended in my second comment. I see that this has been changed in Figure 3. So from my point of view, this is a sufficient and valuable modification.

Response:

We appreciate the reviewer's recognition. In response to the reviewer's first comment, we have responded adequately in our previous reply letter, and no changes are required in the body of the manuscript.

Reviewer #4 (Remarks to the Author):

I value the diligence demonstrated by the authors in responding to my feedback. Nevertheless, certain methodological approaches of the ESR experiments remain inadequately addressed.

Response:

We greatly appreciate the reviewer's further comments, and also for the time and attention given to this manuscript.

1. The authors agreed with me that the integral intensity of ESR spectra is proportional to the number of electron spins. In the figure 2, the electron spin is clearly present in the 'static complex'. Therefore, the number of spins does not change. Therefore, the integral intensity of ESR spectra should not change either. I suggest an alternative redox hypothesis (see supplementary upload).

Irradiation causes electron transfer and spin reactivation (see supplementary upload).

Response:

We greatly appreciate the reviewer's valuable comments. In Figure 2c, the ESR signal of TEMPO remains unchanged before and after UV irradiation, which illustrates that whether it is the light itself or the temperature changes caused by the light, will not directly affect the number of spins for TEMPO (Supplementary Fig. 5a). Only when DETC and TEMPO exist at the same time, the ESR signal of TEMPO+DETC will change after UV irradiation, which is one of the evidences supporting our "static quenching" hypothesis. However, the schematic diagram in Figure 2c does not represent the real chemical structure of "static complex". As mentioned in our manuscript and previous responses to other reviewers' comments, we have emphasized that the real structure of "static complex" or the specific interaction between the ground state DETC and TEMPO, are hard to characterize and given by us now. Additionally, we have replaced the expression "ground state complex" with "ground state interaction" in the previous version of the manuscript and response. But, we forget to modify the annotations, "Static complex", in Figure 2. It may lead the reviewer to misunderstand and we feel sorry for that. We have replaced "static complex" in Figure 2c, using "static quenching".

Again, we greatly appreciate the reviewer for proving a redox hypothesis to us. For DETC+TEMPO, DETC is a photosensitive matter. Just as the reviewer proposed, after DETC absorbing light, intramolecular electron transfer is likely to occur. When DETC contact with TEMPO, an electron transfer between molecules may occur, causing an electron transfer and spin

reactivation of TEMPO and leading to a change of ESR signal of TEMPO. The explanation was indeed more believable compared with “radical hypothesis”, so we have removed the discussion about “radical hypothesis” and added “redox hypothesis” in the revised manuscript (**Highlighted in blue, Page 7**), as follow:

“Additionally, a redox hypothesis seems to give a better explanation. DETC is photosensitive and is likely to undergo intermolecular electron transfer with TEMPO after absorbing photons, causing an spin reactivation of TEMPO [Reference 1, as below], and thereby leading to a change of ESR signal after UV irradiation.”

Nevertheless, since we do not have absolute evidence for the “redox hypothesis”, so we have not modified the schematic diagram of "ground state interaction" in Fig. 2a (quenching path 1).

Reference 1. ACS Applied Polymer Materials, 2019, 1, 2282-2290.

2. My suggestion to utilize a reference sample was not for absolute spin counting, but rather to facilitate the comparison of intensities across different samples. ESR spectroscopy employs such a reference for the normalization of relative intensities, which are influenced not only by the number of spins but also by other factors, including cavity loading and quality factor. In a stable experiment when measuring the same sample, this procedure could, in principle, be disregarded.

Response:

We greatly appreciate the reviewer’s comments. ESR is the premiere analytical technique for the quantitation of the number of unpaired spin species in a sample. Double Integral (DI) of the ESR signal is proportional to the number of unpaired electron spins, which can be reflected by an Equation shown in Fig. R1.

$$\begin{aligned}
 DI &= \int_{b_1}^{b_2} dB \int_{b_1}^B A(B') dB' \\
 &= c [G_R C_t n] \frac{P^{1/2} B_m Q n_B S(S+1) n_S}{f(B_1, B_m)} \quad (1)
 \end{aligned}$$

Thus, DI is proportional to the number of spins n_S , where the calibration constant c combines various spectrometer-dependent factors and the other constants and their units are:

- G_R = receiver gain,
- C_t = conversion time [s],
- n = number of scans acquired,
- P = microwave power [W],
- B_1 = microwave field [G],
- B_m = modulation field [G],
- Q = quality factor of resonator,
- n_B = Boltzmann factor for temperature dependence,
- S = total electron spin,
- n_S = number of electron spins,
- $f(B_1, B_m)$ = spatial distribution of the microwave and modulation fields as experienced by the sample.

Fig. R1 The equation of double integral of ESR signal.

According to the equation, three factors (G_R , C_t and n) can be directly accounted for by processing of the acquired ESR signal. By digitally correcting the measured signal, a normalized ESR spectrum is acquired which facilitates comparison of the ESR spectra acquired with different receiver settings. This normalized acquisition mode is currently implemented in the ELEXSYS line of spectrometers. The remaining terms must be included explicitly for determining the number of spins (n_S), or they must be accounted for comparison with the spectrum of a known standard. Thus, according to the equation, there are three routes to determinate n_S for a given sample [**Book: GR Eaton, SS Eaton, DP Barr, RT Weber, Quantitative EPR, 2010, Chapter 11, Page 112**]:

(1) Relative Spin Quantitation: measurement against an amplitude reference

(2) Absolute Spin Quantitation (Concentration Standard): measurement against a standard of known concentration

(3) Absolute Spin Quantitation (Resonator Calibration): calculation of n_S after determining the factor c .

As the reviewer proposed, ESR test often use concentration or absolute spin standards to avoid the unnecessary introduction of errors in the spin

determination, in the past. However, the ESR spectroscopy we used is a Bruker EMX plus-6/1 spectrometer which do not need to prepare standards, and errors due to sample volume or position are dramatically reduced. The three factors important in this method are the Q-value of the resonator, the constant c , and the distribution $f(B_1, B_m)$ of the microwave and modulation fields within the resonator. The Q-value for the resonator can be readily measured before spectrum acquisition and stored with the experimental data. The determination of the constant c and the distribution function f need only be done once for a given resonator and can be applied to all subsequent measurements with that resonator. Additionally, except for the change in the components of the sample, TEMPO and TEMPO+DETC, all the other test conditions (temperature, irradiation, solvent) were kept consistent in our ESR experiments.

Above all, we believe that the experimental data we provide is reliable.

3. I proposed the application of ESR line fitting to enhance the precision of intensity measurements. This approach appears to be challenging, as the experimental TEMPO spectrum (illustrated in the right portion of the figure below, see uploaded supplementary file) does not align well with the anticipated ESR lineshape (depicted in the left portion of the figure below). Notably, there is a discrepancy with a five-line spectrum observed, as opposed to the expected three-line spectrum. Is it possible that the TEMPO solution was not pure?

Response:

We appreciate the comments. We have further consulted with experts in the field of ESR. At the request of the reviewer, we performed the EPR line fitting for TEMPO and TEMPO+DETC before UV, and it is indeed challenging

to use this method to gain an additional useful information. As for the discrepancy between the fitting one and the experimental one, ESR experts believe that the experiment one is not a five-line spectrum, and that the two small bumps in spectrum may be due to baseline drift. And once we performed the fitting using the simulation data of TEMPO in the software library, the baseline drift was eliminated, which caused the discrepancy between the two figures.

REVIEWERS' COMMENTS

Reviewer #4 (Remarks to the Author):

I appreciate the addition of the proposed by me redox hypothesis.

I'm not 100 % satisfied with the explanations related to EPR line distortion and fitting. For example, 'bumps' in the spectrum are suggested to be a result of 'baseline shift'. That means the instrument may be somewhat unstable.

However, at this point, further discussions related EPR experiments will not be productive. These experiments constitute a minor part of an impressive work, so I suggest, that the manuscript can be published.

Point-to-point Response to Reviewers

Reviewer #4 (Remarks to the Author):

I appreciate the addition of the proposed by me redox hypothesis. I'm not 100 % satisfied with the explanations related to EPR line distortion and fitting. For example, 'bumps' in the spectrum are suggested to be a result of 'baseline shift'. That means the instrument may be somewhat unstable. However, at this point, further discussions related EPR experiments will not be productive. These experiments constitute a minor part of an impressive work, so I suggest, that the manuscript can be published.

Response:

We are grateful to the reviewer for the positive comments and recognition of our work. As stated by the reviewer, this so-called 'baseline shift' has no bearing on the findings of this manuscript. In fact, in our last response, we had also consulted with experts in the field of EPR, and this "baseline drift" did not affect our analysis or judgment.